# ANARCHIC FEDERATED LEARNING

## ABSTRACT

Present-day federated learning (FL) systems deployed over edge networks consists of a large number of workers with high degrees of heterogeneity in data and/or computing capabilities, which call for flexible worker participation in terms of timing, effort, data heterogeneity, etc. To achieve these goals, in this work, we propose a new FL paradigm called "Anarchic Federated Learning" (AFL). In stark contrast to conventional FL models, each worker in AFL has complete freedom to choose i) when to participate in FL, and ii) the number of local steps to perform in each round based on its current situation (e.g., battery level, communication channels, privacy concerns). However, AFL also introduces significant challenges in algorithmic design because the server needs to handle the chaotic worker behaviors. Toward this end, we propose two Anarchic Federated Averaging (AFA) algorithms with two-sided learning rates for both cross-device and cross-silo settings, which are named AFA-CD and AFA-CS, respectively. Somewhat surprisingly, even with general worker information arrival processes, we show that both AFL algorithms achieve the same convergence rate order as the state-of-the-art algorithms for conventional FL. Moreover, they retain the highly desirable *linear speedup effect* in the new AFL paradigm. We validate the proposed algorithms with extensive experiments on real-world datasets.

## 1 INTRODUCTION

Federated Learning (FL) has recently emerged as an important distributed learning framework that leverages numerous workers to collaboratively learn a joint model (Li et al., 2019a; Yang et al., 2019; Kairouz et al., 2019). Since its inception, FL algorithms have become increasingly powerful and have been able to handle various heterogeneity in data, network environments, worker computing capabilities, etc. Moreover, most of the prevailing FL algorithms (e.g., FedAvg (McMahan et al., 2016) and its variants (Li et al., 2018; Zhang et al., 2020b; Karimireddy et al., 2020b;a; Acar et al., 2021)) enjoy so-called *"linear speedup effect,"* i.e., the convergence time of an FL algorithm decreases linearly as the number of workers increases (Stich, 2018; Yu et al., 2019; Wang & Joshi, 2018; Khaled et al., 2019; Karimireddy et al., 2020b; Yang et al., 2021; Qu et al., 2020).

To achieve these salient features, most of the existing FL algorithms have adopted a *server-centric* approach, i.e., the worker behaviors are tightly "dictated" by the server. For example, the server in these FL algorithms can i) determine either all or a subset of workers to participate in each round of FL update; ii) fully control the timing for synchronization and whether to accept/reject information sent from the workers; iii) precisely specify the algorithmic operations (e.g., the number of local steps performed at each worker before communicating with the server), etc.

Despite achieving strong performance guarantees, such a server-centric approach introduces several limitations. Specifically, these server-centric FL algorithms often implicitly rely on the following assumptions: (1) each worker is available for training upon the server's request and throughout a complete round; (2) all participating workers are willing to execute the same number of local updates and communicate with the server in a synchronous manner following a common clock. Unfortunately, in edge networks where many FL systems are deployed, these assumptions are restrictive or even problematic due to the following reasons. First, many requested edge devices on the worker side may not be available in each round because of, e.g., communication errors or battery outages. Second, the use of synchronous communication and an identical number of local updates across all workers ignores the fact that worker devices in edge-based FL systems are heterogeneous in computation and communication capabilities. As a result, stragglers (i.e., slow workers) could significantly slow

down the training process. To mitigate the straggler effect, various robust FL algorithms have been developed. For example, the server in FedAvg (McMahan et al., 2016) can simply ignore and drop the information from the stragglers to speedup learning. However, this may lead to other problems such as wasted computation/energy (Wang et al., 2019), slower convergence (Li et al., 2018), or biased/unfair uses of worker data (Kairouz et al., 2019). Moreover, the synchronous nature of the server-centric approaches implies many networking problems (e.g., interference between workers, periodic traffic spikes, high complexity in maintaining a network-wide common clock).

The above limitations of the current server-centric FL approaches motivate us to propose a new paradigm in FL, which we call **Anarchic Federated Learning** (AFL). In stark contrast to server-centric FL, workers in AFL are completely *free* of the "dictation" from the server. Specifically, each worker has complete freedom to choose when and how long to participate in FL without following any control signals from the server. As a result, the information fed back from workers is inherently asynchronous. Also, each worker can independently determine the number of local update steps to perform in each round based on its current local situation (e.g., battery level, communication channels, privacy concerns). In other words, the amount of local computation at each worker is time-varying, device-dependent, and fully controlled by the worker itself. Clearly, AFL has a much lower server-worker coordination complexity and avoids the aforementioned pitfalls in server-centric FL approaches. However, AFL also introduces significant challenges in algorithmic design on the server-side because the server needs to work much harder to handle the chaotic worker behaviors in AFL (e.g., asynchrony, spatial and temporal heterogeneity in computing). Toward this end, several fundamental questions naturally arise: *1) Is it possible to design algorithms that converge under AFL? 2) If the answer to the previous question is yes, how fast could the algorithms converge? 3) Can the new AFL-based algorithms still achieve the desired "linear speedup effect?"*

In this paper, we answer the above fundamental questions of AFL affirmatively. Our main contributions and key results are summarized as follows:

- We propose a new FL paradigm called Anarchic Federated Learning (AFL), where the workers are allowed to engage in training at will and choose the number of local update steps based on their own time-varying situations (computing resources, energy levels, etc.). This *loose worker-server coupling* significantly simplifies the implementations and renders AFL particularly suitable for FL deployments in edge computing environments. For any AFL algorithms under general worker information arrival processes and non-i.i.d. data across workers, we first establish a fundamental convergence error lower bound that depends on the data heterogeneity in the AFL system. Then, we propose two Anarchic Federated Averaging (AFA) algorithms with two-sided learning rates for two classes of FL problems(cross-device and cross-silo) (Kairouz et al., 2019; Wang et al., 2021).

- For AFL in the *cross-device* (CD) setting, our AFA-CD algorithm converges to an error ball whose size matches the fundamental lower bound, with an $\mathcal{O}(1/\sqrt{mT})$ convergence rate where $m$ is the number of collected workers in each round of update and $T$ is the total number of rounds. We note that this convergence rate retains the highly desirable "linear speedup effect" under AFL.[1] Moreover, under the special case with uniform workers' participation (equivalent to uniform workers sampling in conventional FL (Li et al., 2019c; Karimireddy et al., 2020b;a; Acar et al., 2021)), AFA-CD can further converge to a stationary point (i.e., a singleton) at a convergence rate that matches the state-of-the-art of conventional distributed and federated learning.

- For AFL in the *cross-silo* (CS) setting, our proposed AFA-CS algorithm achieves an enhanced convergence rate of $\mathcal{O}(1/\sqrt{MT})$ by leveraging historical feedback and variance reduction techniques, where $M$ is the total number of workers. This suggests that, not only can "linear speedup" be achieved under AFL-CS, the speedup factor also depends on the total number of workers $M$ instead of the number of collected workers $m$ in each round ($M > m$). To our knowledge, this result is new in the FL literature.

- We validate the proposed algorithms with extensive experiments on CV and NLP tasks and further explore the effect of the asynchrony and local step number in AFL. We also numerically show that our AFL is a general algorithmic framework in the sense that various advanced FL techniques (e.g., FedProx (Li et al., 2018) and SCAFFOLD (Karimireddy et al., 2020b)) can be integrated as the optimizers in our AFA framework to further enhance the AFL performance.

---

[1]To attain $\epsilon$-accuracy, it takes $\mathcal{O}(1/\epsilon^2)$ steps for an algorithm with an $\mathcal{O}(1/\sqrt{T})$ convergence rate, while needing $\mathcal{O}(1/m\epsilon^2)$ steps for another algorithm with an $\mathcal{O}(1/\sqrt{mT})$ convergence rate (the hidden constant in Big-O is the same). In this sense, $\mathcal{O}(1/\sqrt{mT})$ implies a *linear speedup* with respect to the number of workers.

The rest of the paper is organized as follows. In Section 2, we review related work. In Section 3, we introduce AFL and our AFA algorithms, which are followed by their convergence analysis in Section 4. We present the numerical results in Section 5 and conclude the work in Section 6.

## 2   RELATED WORK

**Server-Centric Federated Learning Algorithms:** To date, one of the prevailing FL algorithms is Federated Averaging (FedAvg), which was first proposed in (McMahan et al., 2016) as a heuristic to improve communication efficiency and data privacy for FL. Since then, there have been substantial follow-ups of FedAvg that focus on non-i.i.d. (heterogeneous) data (see, e.g., FedProx (Li et al., 2018), FedPD (Zhang et al., 2020b), SCAFFOLD (Karimireddy et al., 2020b), FedNova (Wang et al., 2020), FedDyn (Acar et al., 2021), and MIME (Karimireddy et al., 2020a)), which are closely related to our work. The main idea for these algorithms is to control the "model drift" (due to heterogeneous datasets and the use of multiple local update steps on the worker side of FedAvg). While these algorithms achieved various degrees of success in dealing with data heterogeneity, they are all server-centric synchronous algorithms that are not easy to implement in edge-based FL due to straggler issues (see discussions in Section 1).

**Federated Learning with Flexible Worker Participation:** Recently, some attempts have been made to alleviate the strict requirements on worker's participation, such as allowing different local steps (Ruan et al., 2021; Wang et al., 2020) and asynchronous FL (Avdiukhin & Kasiviswanathan, 2021; Xie et al., 2019). However, most of these works either lack theoretical performance guarantees or require strong assumptions. For example, Ruan et al. (2021) assumed strongly convex loss function and bounded aggregation coefficient; Avdiukhin & Kasiviswanathan (2021) assumed bounded gradients and same computation time per iteration for all workers. Our AFL paradigm considered in this paper is more general and subsumes all the above settings as special cases. We note, however, that AFL differs from conventional FL with flexible worker participation in that the worker's participation in AFL and its local optimization process are completely determined by the workers, and *not* by the sampling requests from the server. This is more practical since it allows workers to participate in FL under drastically different situations in network, charging/idle cycles, etc. Due to the complex couplings between various sources of randomness and multiple layers of heterogeneity in spatial and temporal domains in AFL, the training algorithm design for AFL and its theoretical analysis is far from a straightforward combination of existing FL techniques for flexible worker participation.

**Asynchronous Distributed Optimization:** The asynchrony in AFL also shares some similarity with asynchronous distributed optimization. The basic idea of asynchronous distributed optimization is to forgo the common clock in the system to lower the system implementation complexity in distributed optimization. However, due to extra noise introduced by asynchrony, it is highly non-trivial to establish the convergence performance of asynchronous distributed optimization algorithms. To address this challenge, asynchronous distributed optimization has been studied extensively in the machine learning and optimization literature (see, e.g., Lian et al. (2018); Niu et al. (2011); Agarwal & Duchi (2012); Paine et al. (2013); Xie et al. (2019); Zhang et al. (2020a) and references therein). We note that the AFL paradigm considered in this paper is more general and subsumes asynchronous distributed optimization as a special case. To see this, note that in addition to the asynchronous updates at the server, the workers in AFL could further have different numbers of local update steps. Moreover, the workers may not even need to be work-conserving (i.e., workers could be idle between rounds of updates). As a result, the convergence analysis of AFL is much more challenging.

## 3   ANARCHIC FEDERATED LEARNING

**1) Overview of Anarchic Federated Learning:** The goal of FL is to solve an optimization problem in the form of $\min_{x \in \mathbb{R}^d} f(x) := \frac{1}{M} \sum_{i=1}^{M} f_i(x)$, where $f_i(x) \triangleq \mathbb{E}_{\xi_i \sim D_i}[f_i(x, \xi_i)]$ is the local (non-convex) loss function associated with a local data distribution $D_i$ and $M$ is the total number of workers. For the setting with heterogeneous (non-i.i.d.) datasets at the workers, we have $D_i \neq D_j, \text{if } i \neq j$. In terms of the assumption on the size of workers, FL can be classified as cross-device FL and cross-silo FL (Kairouz et al., 2019; Wang et al., 2021). Cross-device FL is designed for large-scale FL with a massive number of mobile or IoT devices ($M$ is large). As a result, the server can only afford to collect information from a subset of workers in each round of update and does not have sufficient

---

**Algorithm 1** The Basic Framework of Anarchic Federated Learning (AFL).

---

**At the Server (Concurrently with Workers):**
1. (Concurrent Thread) Collect local updates returned from the workers.
2. (Concurrent Thread) Aggregate local update returned from the collected workers and update global model following some server-side optimization process.

**At Each Worker (Concurrently with Server):**
1. Once decided to participate in the training, pull the global model with current timestamp.
2. Perform (multiple) local update steps following some worker-side optimization process.
3. Return the computation result and the associated pulling timestamp to the server, with extra processing if so desired.

---

memory space to store workers' information across rounds. In comparison, the number of workers in cross-silo FL is relatively small. Although the server in cross-silo FL may still have to collect information only from a subset of workers in each round, it has enough memory space to store each worker's most recent information.

The basic process of AFL is illustrated in Algorithm 1. Here, the server is responsible for: 1) collecting the local updates returned from workers, and 2) aggregating the obtained updates once certain conditions are satisfied (e.g., upon collecting $m \in (0, M]$ local updates from workers) to update the global model. In AFL, each worker could be non-work-conserving (i.e., idling is allowed between each two successive participations in training). Whenever a worker intends to participate in the training, it first pulls the current model parameters from the server. Then, after finishing multiple local update steps (more on this next) with some worker-side optimization process (e.g., following stochastic gradients or additional information such as variance-reduced and/or momentum adjustments), the worker reports the computation results to the server (potentially with extra processing if so desired, such as rescaling, compression for communication-efficiency, etc.). Note that the above operations can happen *concurrently* on the server and worker sides, i.e., the processings at the workers and the server are independent of each other in the temporal domain.

We remark that AFL is a general computing architecture that subsumes the conventional FL and asynchronous distributed optimization as special cases. From an optimization perspective, the server and the workers may adopt independent optimization processes, thus enabling a much richer set of learning *"control knobs"* (e.g., separated learning rates, separated batch sizes). More importantly, each worker is able to completely take control of its own optimization process, even using a time-varying number of local update steps and optimizers, which depend on its local dataset and/or its device status (e.g., battery level, privacy preference).

**2) A Fundamental Convergence Error Lower Bound of AFL:** Before developing training algorithms for AFL, it is insightful to first obtain a fundamental understanding of the *performance limit of any AFL training algorithms*. Toward this end, we first state several assumptions that are needed for our theoretical analysis throughout the rest of this paper.

**Assumption 1.** *(L-Lipschitz Continuous Gradient) There exists a constant $L > 0$, such that $\|\nabla f_i(\mathbf{x}) - \nabla f_i(\mathbf{y})\| \leq L\|\mathbf{x} - \mathbf{y}\|$, $\forall \mathbf{x}, \mathbf{y} \in \mathbb{R}^d$, and $i \in [M]$.*

**Assumption 2.** *(Unbiased Local Stochastic Gradient) Let $\xi^i$ be a random local data sample at worker $i$. The local stochastic gradient is unbiased, i.e., $\mathbb{E}[\nabla f_i(\mathbf{x}, \xi^i)] = \nabla f_i(\mathbf{x})$, $\forall i \in [m]$, where the expectation is taken over the local data distribution $D_i$.*

**Assumption 3.** *(Bounded Local and Global Variances) There exist two constants $\sigma_L \geq 0$ and $\sigma_G \geq 0$, such that the variance of each local stochastic gradient estimator is bounded by $\mathbb{E}[\|\nabla f_i(\mathbf{x}, \xi^i) - \nabla f_i(\mathbf{x})\|^2] \leq \sigma_L^2$, $\forall i \in [M]$, and the global variability of the local gradient of the cost function is bounded by $\|\nabla f_i(\mathbf{x}) - \nabla f(\mathbf{x})\|^2 \leq \sigma_G^2$, $\forall i \in [M], \forall k$.*

The first two assumptions are standard in the convergence analysis of non-convex optimization (see, e.g., (Ghadimi & Lan, 2013; Bottou et al., 2018)). For Assumption 3, the bounded local variance is also a standard assumption. We utilize a universal bound $\sigma_G$ to quantify the data heterogeneity among different workers. This assumption is also frequently used in the literature of FL with non-i.i.d. datasets (Reddi et al., 2020; Wang et al., 2019; Yang et al., 2021) as well as in decentralized optimization (Kairouz et al., 2019).

To establish a fundamental convergence error lower bound, we consider the most general case where no assumption on the arrival processes of the worker information is made, except that each worker's participation in FL is independent of each other. In such general worker information arrival processes, we prove the following fundamental lower bound of convergence error by constructing a worst-case scenario (see Appendix B.1 for proof details):

**Theorem 1** (Convergence Error Lower Bound for General Worker Information Arrival Processes). *For any level of heterogeneity characterized by $\sigma_G$, there exists loss functions and worker participation process satisfying Assumptions 1- 3 for which the output $\hat{\mathbf{x}}$ of any randomized FL algorithm satisfies:*

$$\mathbb{E}\|\nabla f(\hat{\mathbf{x}})\|^2 = \Omega(\sigma_G^2), \tag{1}$$

**Remark 1.** The lower bound in Theorem 1 indicates that no algorithms for AFL could converge to a stationary point under general worker information arrival processes, due to the significant system heterogeneity and randomness caused by such general worker information arrivals. The possible system heterogeneity coupled with non-i.i.d. data result in objective value drifts, which further lead to an inevitable error in convergence. We note that our lower bound also holds for non-i.i.d. FL including synchronous FedAvg and its variants, thus also providing insights for conventional FL. To ensure convergence to a stationary point, extra assumptions for the worker information arrivals have to be made, e.g., uniformly distributed arrivals (see Theorem 3) and bounded delays (see Theorem 4).

## 4 THE ANARCHIC FEDERATED AVERAGING (AFA) ALGORITHMS

Upon obtaining a fundamental understanding of the training algorithm performance limit from the convergence error in Theorem 1, we are now in a position to develop AFL algorithms that are guided by our fundamental lower bound. Toward this end, in this section, we will develop two anarchic federated averaging (AFA) algorithms for the cross-device (CD) and cross-silo (CS) settings in Section 4.1 and 4.2, respectively.

### 4.1 THE AFA-CD ALGORITHM FOR CROSS-DEVICE AFL

**1) The AFA-CD Algorithm:** First, we propose the AFA-CD algorithm for the cross-device AFL setting. As mentioned earlier, cross-device AFL is suitable for cases with a massive number of edge devices. In each round of global model update, only a small subset of workers are used in the training. The server is assumed to have no historical information of the workers. As shown in Algorithm 2, AFA-CD closely follows the AFL architecture shown in Algorithm 1. Here, we use standard stochastic gradient descent (SGD) as the server- and worker-side optimizer. In each update $t$, $t = 1, \dots, T$, the server waits until collecting $m$ local updates $\{\mathbf{G}_i(\mathbf{x}_{t-\tau_{t,i}})\}$ from workers to form a set $\mathcal{M}_t$ with $|\mathcal{M}_t| = m$, where $\tau_{t,i}$ represents the random delay of the local update of worker $i$, $i \in \mathcal{M}_t$ (Server Code, Line 1). Once $\mathcal{M}_t$ is formed, the server aggregates all local updates $\mathbf{G}_i(\mathbf{x}_{t-\tau_{t,i}}), i \in \mathcal{M}_t$ and updates global model (Server Code, Line 2). We count each global model update as one communication round. Meanwhile, for each worker, it pulls the current global model parameters with time stamp $\mu$ once it decides to participate in training (Worker Code, Line 1). Each worker can then choose a desired number of local update steps $K_{t,i}$ (could be time-varying and device-dependent) to perform SGD updates for $K_{t,i}$ times, and then return the rescaled sum of all stochastic gradients with timestamp $\mu$ to the server (Worker Code, Lines 2–3).

**2) Convergence Analysis of the AFA-CD Algorithm:** We first conduct the convergence of AFA-CD under general worker information arrival processes. We let $K$ be the maximum local update steps, which is defined as $K := \max_{t \in [T], i \in [M]} \{K_{t,i}\}$. We let $\tau$ be the maximum delay of workers' returned information, which is defined as $\tau := \max_{\mathbf{t} \in [T], i \in \mathcal{M}_t} \{\tau_{t,i}\}$. Also, we use $f_0 = f(\mathbf{x}_0)$ and $f_*$ to denote the initial and the optimal objective values, respectively. Then, we have the following convergence result for the AFA-CD algorithm (see proof details in Appendix B.2):

**Theorem 2** (AFA-CD with General Worker Information Arrival Processes). *Under Assumptions 1- 3, choose server-side and worker-side learning rates $\eta$ and $\eta_L$ such that the following relationships hold: $180\eta_L^2 K^2 L^2 \tau < 1$ and $2L\eta\eta_L + 3\tau^2 L^2 \eta^2 \eta_L^2 \leq 1$. Then, the output sequence $\{\mathbf{x}_t\}$ generated by AFedAvg-TSLR-CD with general worker information arrival processes satisfies:*

$$\frac{1}{T} \sum_{t=0}^{T-1} \mathbb{E}\|\nabla f(\mathbf{x}_t)\|^2 \leq \frac{4(f_0 - f_*)}{\eta\eta_L T} + 4\big(\alpha_L \sigma_L^2 + \alpha_G \sigma_G^2\big), \tag{2}$$

---

**Algorithm 2** The AFA-CD Algorithm for Cross-Device AFL.

---

**At the Server (Concurrently with Workers):**

    1. In the $t-$th update round, collect $m$ local updates $\{\mathbf{G}_i(\mathbf{x}_{t-\tau_{t,i}}), i \in \mathcal{M}_t\}$ returned from the workers to form the set $\mathcal{M}_t$, where $\tau_{t,i}$ represents the random delay of the worker $i$'s local update, $i \in \mathcal{M}_t$.

    2. Aggregate and update: $\mathbf{G}_t = \frac{1}{m}\sum_{i\in\mathcal{M}_t}\mathbf{G}_i(\mathbf{x}_{t-\tau_{t,i}}), \quad \mathbf{x}_{t+1} = \mathbf{x}_t - \eta\mathbf{G}_t.$

**At Each Worker (Concurrently with Server):**

    1. Once decided to participate in the training, retrieve the parameter $\mathbf{x}_\mu$ from the server and its timestamp, set the local model: $\mathbf{x}^i_{\mu,0} = \mathbf{x}_\mu$.

    2. Choose a number of local steps $K_{t,i}$, which can be time-varying and device-dependent. Let $\mathbf{x}^i_{\mu,k+1} = \mathbf{x}^i_{\mu,k} - \eta_L\mathbf{g}^i_k$, where $\mathbf{g}^i_k = \nabla f_i(\mathbf{x}^i_{\mu,k}, \xi^i_k), k = 0, \ldots, K_{t,i} - 1$.

    3. Sum and rescale the stochastic gradients: $\mathbf{G}_i(\mathbf{x}_\mu) = \frac{1}{K_{t,i}}\sum_{j=0}^{K_{t,i}-1}\mathbf{g}^i_j$. Return $\mathbf{G}_i(\mathbf{x}_\mu)$.

---

*where* $\alpha_L \triangleq \left[\frac{2\eta\eta_L}{m} + \frac{3\tau^2 L^2\eta^2\eta_L^2}{2m} + \frac{15\eta_L^2 KL^2}{2}\right]$ *and* $\alpha_G \triangleq \left[\frac{3}{2} + 45K^2 L^2\eta_L^2\right]$.

With Theorem 2, we immediately have the following convergence rate result for AFA-CD, which implies the highly desirable "linear speedup effect" can be achieved by AFA-CD.

**Corollary 1** (Linear Speedup to Error Ball). *By setting* $\eta_L = \frac{1}{\sqrt{T}}$, *and* $\eta = \sqrt{m}$, *the convergence rate of AFA-CD with general worker information arrival processes is:*

$$\frac{1}{T}\sum_{t=0}^{T-1}\mathbb{E}\|\nabla f(\mathbf{x}_t)\|^2 = \mathcal{O}\left(\frac{1}{m^{1/2}T^{1/2}}\right) + \mathcal{O}\left(\frac{\tau^2}{T}\right) + \mathcal{O}\left(\frac{K^2}{T}\right) + \mathcal{O}(\sigma_G^2).$$

**Remark 2.** First, we note that the non-vanishing error term $\mathcal{O}(\sigma_G^2)$ in Corollary 1 matches the lower bound in Theorem 1. This implies that the convergence error of AFL-CD is *order-optimal*. Recall that this non-vanishing convergence error is a direct consequence of objective function drift under the general worker information arrivals (no assumption on the arrivals of the worker participation in each round of update), which is independent of the choices of learning rates, the number of local update steps, and the number of global update rounds (more discussion in the supplementary material). Also, for a sufficiently large $T$, the dominant term $\mathcal{O}(\frac{1}{m^{1/2}T^{1/2}})$ implies that AFA-CD achieves the linear speedup effect before reaching a constant error neighborhood with size $\mathcal{O}(\sigma_G^2)$.

**Remark 3.** The conditions for the learning rates $\eta$ and $\eta_L$ are a natural extension of SGD. When $\tau = 0$ (synchronous setting) and $K = 1$ (single local update), the condition becomes $\eta\eta_L \leq \frac{1}{2L}$, which recovers the same learning rate condition for SGD in (Ghadimi & Lan, 2013). Also, the suggested worker-side learning rate $\eta_L$ is independent of local update steps. This is different from previous work that $\eta_L$ depends heavily on local update steps (Wang et al., 2020; Yang et al., 2021), thus making it more practical for implementation.

Given the weak convergence result implied by the lower bound in Theorem 1 under general workers' information arrivals, it is important to understand what extra assumptions on the worker information arrivals are needed in order to guarantee convergence. Toward this end, we consider a special case where the arrivals of worker returned information in each round for global update is uniformly distributed among the workers. In this setting, $\mathcal{M}_t$ can be viewed as a subset with size $m$ independently and uniformly sampled from $[M]$ without replacement. For FL systems with a massive number of workers, this assumption of uniformly distributed arrivals is a good approximation for worker participation in cross-device FL (McMahan et al., 2016; Li et al., 2019a). Also, one can map this setting into the conventional cross-device FL systems with uniform worker sampling in partial worker participation, which is a widely-used assumption (McMahan et al., 2016; Li et al., 2019c; 2018; Yang et al., 2021; Wang et al., 2020). For this special case, the convergence performance of AFA-CD can be improved as follows (see proof details in Appendix B.3):

**Theorem 3.** *Under Assumptions 1- 3, choose server-side and worker-side learning rates* $\eta$ *and* $\eta_L$ *such that the following relationships hold:* $\eta_L^2[6(2K^2 - 3K + 1)L^2] \leq 1$, $120L^2 K^2\eta_L^2\tau + 4(L\eta\eta_L + L^2\eta^2\eta_L^2\tau^2)\frac{M-m}{m(M-1)}(90K^2 L^2\eta_L^2\tau + 3\tau) < 1$. *Then, the output sequence* $\{\mathbf{x}_t\}$ *generated by AFA-CD*

---

**Algorithm 3** The AFA-CS Algorithm for Cross-Silo AFL.

---
**At the Server (Concurrently with Workers):**
    1. In the $t-$th update round, collect $m$ local updates $\{\mathbf{G}_i(\mathbf{x}_{t-\tau_{t_i}})\}$ from the workers to form the set $\mathcal{M}_t$.
    2. Update worker $i$'s information in the memory using the returned local update $\mathbf{G}_i$.
    3. Aggregate and update: $\mathbf{G}_t = \frac{1}{M}\sum_{i\in[M]}\mathbf{G}_i,\quad \mathbf{x}_{t+1} = \mathbf{x}_t - \eta\mathbf{G}_t$.

**At Each Worker (Concurrently with Server):** Same as AFA-CD Worker Code.

---

*with uniformly distributed worker information arrivals satisfies:*

$$\frac{1}{T}\sum_{t=0}^{T-1}\mathbb{E}\|\nabla f(\mathbf{x}_t)\|_2^2 \leq \frac{4(f_0 - f_*)}{\eta\eta_L T} + 4\big(\alpha_L\sigma_L^2 + \alpha_G\sigma_G^2\big), \tag{3}$$

*where $\alpha_L$ and $\alpha_G$ are constants that are defined as:*

$$\alpha_L \triangleq \left[\left(\frac{L\eta\eta_L}{m} + \frac{\tau^2 L^2\eta^2\eta_L^2}{m} + 5KL^2\eta_L^2\right) + (L\eta\eta_L + L^2\eta^2\eta_L^2\tau^2)\frac{M-m}{m(M-1)}(15KL^2\eta_L^2)\right],$$

$$\alpha_G \triangleq \left[30K^2L^2\eta_L^2 + (L\eta\eta_L + L^2\eta^2\eta_L^2\tau^2)\frac{M-m}{m(M-1)}(90K^2L^2\eta_L^2 + 3)\right].$$

Furthermore, by choosing the server- and worker-side learning rates appropriately, we immediately have the following linear speedup convergence result for AFA-CD:

**Corollary 2** (Linear Speedup to Stationary Point). *By setting $\eta_L = \frac{1}{\sqrt{T}}$ and $\eta = \sqrt{m}$, the convergence rate of AFA-CD with uniformly distributed worker information arrivals is:*

$$\frac{1}{T}\sum_{t=0}^{T-1}\mathbb{E}\|\nabla f(\mathbf{x}_t)\|_2^2 = \mathcal{O}(\frac{1}{m^{1/2}T^{1/2}}) + \mathcal{O}\left(\frac{\tau^2}{T}\right) + \mathcal{O}\left(\frac{K^2}{T}\right) + \mathcal{O}\left(\frac{K^2}{m^{1/2}T^{3/2}}\right) + \mathcal{O}\left(\frac{K^2\tau^2}{T^2}\right).$$

**Remark 4.** For a sufficiently large $T$, the linear speedup effect to a stationary point (rather than a constant error neighborhood) can be achieved with any finitely bounded maximum local update steps $K$ and maximum delay $\tau$, i.e., $\mathcal{O}(\frac{1}{\sqrt{mT}})$. Note this rate does not depend on the delay $\tau$ and local computation steps $K$ after sufficiently many rounds $T$, the negative effect of using outdated information in such asynchronous setting vanishes asymptotically. Further, for $\sigma_G = 0$ (i.i.d. data) and $K = 1$ (single local update step), our AFA-CD algorithm is exactly the same as the AsySG-con algorithm (Lian et al., 2018) in asynchronous parallel distributed optimization. It can be readily verified that AFA-CD achieves the same rate as that of the AsySG-con algorithm. When $\tau = 0$ (synchronous setting) and $K_{t,i} = K$ (same number of local update steps across workers), AFA-CD becomes the generalized synchronous FedAvg algorithm with *two-sided* learning rates (Yang et al., 2021; Karimireddy et al., 2020b; Reddi et al., 2020). Remarkably, the result of AFA-CD shows that we can achieve a faster convergence rate than $\mathcal{O}(\sqrt{K}/\sqrt{mT})$ in (Yang et al., 2021) and the same rate as FedAvg analysis in (Karimireddy et al., 2020b).

### 4.2 THE AFA-CS ALGORITHM FOR CROSS-SILO FL

**1) The AFA-CS Algorithm:** As mentioned earlier, cross-silo FL is suitable for collaborative learning among a relatively small number of (organizational) workers. Thanks to the relatively small number of workers, each worker's feedback can be stored at the server. As a result, the server could reuse the historical information of each specific worker in each round of global update.

As shown in Algorithm 3, the AFA-CS algorithm also closely follows the AFL architecture as shown in Algorithm 1. In each round of global model update, a subset of workers could participate in the training (Server Code, Line 1). Compared to AFA-CD, the key difference in AFA-CS is in Line 2 of the Server Code, where the server stores the collected local updates $\{\mathbf{G}_i\}$ for each worker $i \in \mathcal{M}_t$ into the memory space at the server (Server Code, Line 2). As a result, whenever a worker $i$ returns a local update to the server upon finishing its local update steps, the server will update the memory space corresponding to worker $i$ to replace the old information with this newly received update from

worker $i$. Similar to AFA-CD, every $m$ new updates in the AFA-CS algorithm trigger the server to aggregate all the $\mathbf{G}_i, i \in [M]$ (newly received information if $i \in \mathcal{M}_t$ or stored information if $i \notin \mathcal{M}_t$) and update the global model. The Worker Code in AFA-CS is exactly the same as AFA-CD and its description is omitted for brevity.

**2) Convergence Analysis of the AFA-CS Algorithm:** For cross-silo AFL, our AFA-CS algorithm achieves the following convergence performance (see proof details in Appendix C):

**Theorem 4.** *Under Assumptions 1- 3, choose sever- and worker-side learning rates $\eta$ and $\eta_L$ in such a way that there exists a non-negative constant series $\{\beta_\mu\}_{u=0}^{\tau-1}$ satisfying the following relationship:*

$$12L\eta\eta_L + \frac{540(M-m')^2}{M^2}(1+L\eta\eta_L)K^2L^2\eta_L^2(1+\tau) + 180K^2L^2\eta_L^2 + 320L^3K^2\eta\eta_L^3 < 1, \quad (4)$$

$$\eta\eta_L\left(\frac{9(M-m')^2}{2M^2}(1+L\eta\eta_L)\right)3\tau L^2 + (\beta_{u+1} - \beta_u) \leq 0, \quad (5)$$

$$\eta\eta_L\left(\frac{9(M-m')^2}{2M^2}(1+L\eta\eta_L)\right)3\tau L^2 - \beta_{\tau-1} \leq 0, \quad (6)$$

$$\frac{3}{2M}\sigma_L^2 \leq (\frac{1}{2} - \beta_0\eta\eta_L)\mathbb{E}\|\mathbf{G}_t\|_2^2, \quad (7)$$

*the output sequence $\{\mathbf{x}_t\}$ generated by the AFA-CS algorithm for general worker information arrival processes with bounded delay ($\tau := \max_{\mathbf{t}\in[T], i\in[M]}\{\tau_{t,i}\}$) satisfies:*

$$\frac{1}{T}\sum_{t=0}^{T-1}\mathbb{E}\|\nabla f(\mathbf{x}_t)\|_2^2 \leq \frac{4(V(x_0) - V(x_*))}{\eta\eta_L T} + 4\big(\alpha_L\sigma_L^2 + \alpha_G\sigma_G^2\big), \quad (8)$$

*where $\alpha_L$ and $\alpha_G$ are constants defined as follows:*

$$\alpha_L \triangleq [\frac{3L\eta\eta_L}{2M} + 5KL^2\eta_L^2(\frac{9(M-m')^2}{M^2}(1+L\eta\eta_L) + (\frac{3}{2} + 3L\eta\eta_L))],$$

$$\alpha_G \triangleq (\frac{9(M-m')^2}{M^2}(1+L\eta\eta_L) + (\frac{3}{2} + 3L\eta\eta_L))(30K^2L^2\eta_L^2),$$

*and $V(\cdot)$ is defined as $V(\mathbf{x}_t) \triangleq f(\mathbf{x}_t) + \sum_{u=0}^{\tau-1}\beta_u\|\mathbf{x}_{t-u} - \mathbf{x}_{t-u-1}\|^2$, $m'$ is the number of updates in the memory space with no time delay ($\tau_{t,i} = 0$).*

**Remark 5.** To see how stringent the conditions for learning rates in Theorem 4 are, note that Eq. (4) implies that $\mathcal{O}(\eta\eta_L + \eta\eta_L^3K^2 + \tau K^2\eta_L^2 + \tau K^2\eta\eta_L^3) < 1$. With a sufficiently small learning rate $\eta_L$ for given bounded time delay $\tau$ and maximum local steps $K$, Eq. (4) can be satisfied. Eqs. (5)–(6) imply that $\{\beta_\mu\}_{u=0}^{\tau-1}$ is a non-negative decreasing series with difference $\mathcal{O}(\eta\eta_L\tau + \eta^2\eta_L^2\tau)$. As a result, $\beta_0 = \Omega(\eta\eta_L\tau^2 + \eta^2\eta_L^2\tau^2)$. Plugging it into Eq. (7) yields that the update term $\mathbf{G}_t$ should be larger than the variance term $\sigma_L^2/M$, which can be satisfied if the number of workers $M$ is sufficiently large or $\sigma_L^2$ is sufficiently small (i.e., workers use a sufficiently large batch size).

By choosing appropriate learning rates, we immediately have *stronger* linear speedup convergence:

**Corollary 3** (Linear Speedup). *By setting $\eta_L = \frac{1}{\sqrt{T}}$, and $\eta = \sqrt{M}$, the convergence rate of the AFA-CS algorithm for general worker information arrival processes with bounded delay is:*

$$\frac{1}{T}\sum_{t=0}^{T-1}\mathbb{E}\|\nabla f(\mathbf{x}_t)\|_2^2 = \mathcal{O}\left(\frac{1}{M^{1/2}T^{1/2}}\right) + \mathcal{O}\left(\frac{K^2}{T}\right) + \mathcal{O}\left(\frac{K^2M^{1/2}}{T^{3/2}}\right).$$

**Remark 6.** Compared to Corollary 1, we can see that, by reusing historical data, the AFA-CS algorithm can eliminate the non-vanishing $\mathcal{O}(\sigma_G^2)$ error term even for general worker information arrival processes with bounded delay. The bounded delay implicitly requires each workers at least participate in the training process, eliminating the worse-case scenario shown in Theorem 1. On the other hand, although the server only collects $m$ workers' feedback in each round of global model update, the server leverages all $M$ workers' feedback by reusing historical information. Intuitively, this translates the potential objection function drift originated from general worker information arrival process into the negative effect of delayed returns $\mathbf{G}(\mathbf{x}_{t-\tau_{t,i}})$ from workers. It can be shown that such a negative effect of delayed returns vanishes asymptotically as the number of communication rounds $T$ gets large and in turn diminishes the convergence error.

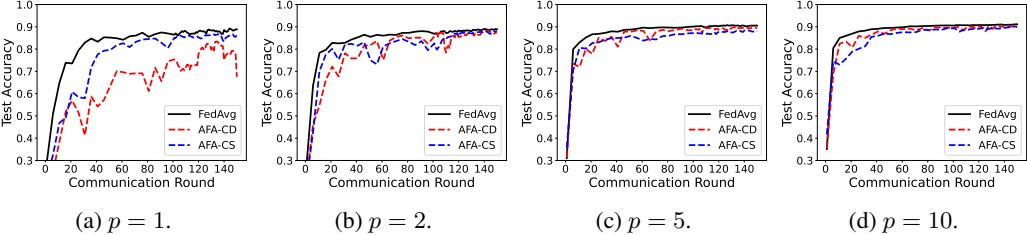

Figure 1: Test accuracy for logistic regression on non-i.i.d. MNIST with different $p$-values.

**Remark 7.** AFA-CS achieves a *stronger* linear speedup $\mathcal{O}(1/\sqrt{MT})$, which is new in the FL literature to our knowledge. Specifically, even though partial ($m$) workers participation is used in each round, AFA-CS is able to achieve a surprising speedup with respect to total number of workers $M$ ($M > m$). Also, AFA-CS generalizes the lazy aggregation strategy in distributed learning (e.g., LAG (Chen et al., 2018)) by setting $K = 1$ (single local update), $\tau = 0$ (synchronous setting) and $\sigma_L = 0$ (using full gradient descents instead of stochastic gradients) and further improve the rate of LSAG (Chen et al., 2020) from $\mathcal{O}(1/\sqrt{T})$ to $\mathcal{O}(1/\sqrt{MT})$.

## 5 NUMERICAL RESULTS

In this section, we conduct experiments to verify our theoretical results. We use i) logistic regression (LR) on manually partitioned non-i.i.d. MNIST dataset (LeCun et al., 1998), ii) convolutional neural network (CNN) for manually partitioned CIFAR-10 (Krizhevsky, 2009), and iii) recurrent neural network (RNN) on natural non-i.i.d. dataset *Shakespeare* (McMahan et al., 2016). In order to impose data heterogeneity in MNIST and CIFAR-10 data, we distribute the data evenly into each worker in label-based partition following the same process in the literature (e.g., McMahan et al. (2016); Yang et al. (2021); Li et al. (2019c)). Therefore, we can use a parameter $p$ to represent the classes of labels in each worker's dataset, which signifies data heterogeneity: the smaller the $p$-value, the more heterogeneous the data across workers (cf. Yang et al. (2021); Li et al. (2019c) for details). Due to space limitation, we relegate the details of models, datasets and hyper-parameters, and further results of CNN and RNN to the appendix.

In Figure 1, we illustrate the test accuracy for LR on MNIST with different $p$-values. We use the classical FedAvg algorithm (McMahan et al., 2016) for conventional FL with uniform worker sampling as a baseline, since it corresponds to the most ideal scenario where workers are fully cooperative with the server. We examine the learning performance degradation of AFA algorithms (due to anarchic worker behaviors) compared to this ideal baseline. For our AFA-CD and AFA-CS with general worker information arrival processes, the test accuracy is comparable to or nearly the same as that of FedAvg. This confirms our theoretical results and validates the effectiveness of our AFA algorithms. We further evaluate the impacts of various factors in AFL, including asynchrony, heterogeneous computing, worker's arrival process, and non-i.i.d. datasets, on convergence rate of our proposed AFA algorithms. Note that AFL subsumes FedAvg and many variants when the above hyper-parameters are set as constant. Also, AFL coupled with other FL algorithms such as FedProx (Li et al., 2018) and SCAFFOLD (Karimireddy et al., 2020b) is tested. Our results show that the AFA algorithms are robust against all asynchrony and heterogeneity factors in AFL. Due to space limitation, we refer readers to the appendix for all these experimental results.

## 6 CONCLUSIONS

In this paper, we propose a new paradigm in FL called "Anarchic Federated Learning" (AFL). In stark contrast to conventional FL models where the server and the worker are tightly coupled, AFL has a much lower sever-worker coordination complexity, allowing a flexible worker participation. We propose two Anarchic Federated Averaging algorithms with two-sided learning rates for both cross-device and cross-silo settings, which are named AFA-CD and AFA-CS, respectively. We show that both algorithms retain the highly desirable linear speedup effect in the new AFL paradigm. Moreover, we show that our AFL framework works well numerically by employing advance FL algorithms FedProx and SCAFFOLD as the optimizer in worker's side.

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

# Appendix

In this supplementary material, we provide the detailed proofs for all theoretical results in this paper. Before presenting the proofs, we introduce some notations that will be used subsequently.. We assume there exists $M$ workers in total in the FL systems. In each communication round, we assume a subset $\mathcal{M}_t$ of workers to be used, with $|\mathcal{M}_t| = m$. We use $\mathbf{G}_i(\mathbf{x}_t)$ to represent the local update returned from worker $i, i \in [M]$ given global model parameter $\mathbf{x}_t^0 = \mathbf{x}_t$. Also, we define $\mathbf{G}_i(\mathbf{x}_t) \triangleq \frac{1}{K_{t,i}} \sum_{j=0}^{K_{t,i}-1} \nabla f_i(\mathbf{x}_t^j, \xi_{t,i})$, where $\mathbf{x}_t^j$ represents the trajectory of the local model in the worker. We use $\Delta_i$ to denote the average of the full gradients long the trajectory of local updates, i.e., $\Delta_i(\mathbf{x}_t) = \frac{1}{K_{t,i}} \sum_{j=0}^{K_{t,i}-1} \nabla f_i(\mathbf{x}_t^j)$. With the above notations, we are now in a position to present the proofs of the theoretical results in this paper.

## A    Proofs of Lemma 1 and Lemma 2

We start with proving two results stated in the following two lemmas, which will be useful in the rest of the proofs.

**Lemma 1.** $\mathbb{E}[\mathbf{G}_i(\mathbf{x}_t)] = \Delta_i(\mathbf{x}_t)$, $\mathbb{E}[\|\mathbf{G}_i(\mathbf{x}_t) - \Delta_i(\mathbf{x}_t)\|^2] \leq \sigma_L^2, \forall i \in [M]$.

*Proof.* Taking the expectation of $\mathbf{G}_i(\mathbf{x}_t)$, we have:

$$\mathbb{E}[\mathbf{G}_i(\mathbf{x}_t)] = \mathbb{E}\left[\frac{1}{K_{t,i}} \sum_{j=0}^{K_{t,i}-1} \nabla f_i(\mathbf{x}_t^j, \xi_{t,i})\right]$$

$$= \frac{1}{K_{t,i}} \sum_{j=0}^{K_{t,i}-1} \mathbb{E}\nabla f_i(\mathbf{x}_t^j, \xi_{t,i})$$

$$= \Delta_i(\mathbf{x}_t).$$

Also, by computing the mean square error between $\mathbf{G}_i(\mathbf{x}_t)$ and $\Delta_i(\mathbf{x}_t)$, we have:

$$\mathbb{E}[\|\mathbf{G}_i(\mathbf{x}_t) - \Delta_i(\mathbf{x}_t)\|^2] = \mathbb{E}\left[\|\frac{1}{K_{t,i}} \sum_{j=0}^{K_{t,i}-1} \nabla f_i(\mathbf{x}_t^j, \xi_{t,i}) - \sum_{j=0}^{K_{t,i}-1} \nabla f_i(\mathbf{x}_t^j)\|^2\right]$$

$$= \frac{1}{K_{t,i}^2}\mathbb{E}\left[\|\sum_{j=0}^{K_{t,i}-1} \nabla f_i(\mathbf{x}_t^j, \xi_{t,i}) - \sum_{j=0}^{K_{t,i}-1} \nabla f_i(\mathbf{x}_t^j)\|^2\right]$$

$$\leq \frac{1}{K_{t,i}} \sum_{j=0}^{K_{t,i}-1} \mathbb{E}\left[\|\nabla f_i(\mathbf{x}_t^j, \xi_{t,i}) - \sum_{j=0}^{K_{t,i}-1} \nabla f_i(\mathbf{x}_t^j)\|^2\right]$$

$$\leq \sigma_L^2.$$

This completes the proof of Lemma 1.    $\square$

**Lemma 2.** *For a fixed set $\mathcal{M}_t$ with cardinality $m$,* $\mathbb{E}\left[\|\sum_{i \in \mathcal{M}_t} \mathbf{G}_i(\mathbf{x}_{t-\tau_{t,i}})\|^2\right] \leq \|\sum_{i \in \mathcal{M}_t} \Delta_i(\mathbf{x}_{t-\tau_{t,i}})\|^2 + m\sigma_L^2.$

*Proof.* By the definition of variance ($\mathbb{E}[(X - \mathbb{E}[X])^2] = \mathbb{E}[X^2] - [\mathbb{E}[X]]^2$), we have:

$$\mathbb{E}\left[\|\sum_{i \in \mathcal{M}_t} \mathbf{G}_i(\mathbf{x}_{t-\tau_{t,i}})\|^2\right] = \mathbb{E}\left[\|\sum_{i \in \mathcal{M}_t} \mathbf{G}_i(\mathbf{x}_{t-\tau_{t,i}}) - \Delta_i(\mathbf{x}_{t-\tau_{t,i}})\|^2\right] + \|\sum_{i \in \mathcal{M}_t} \Delta_i(\mathbf{x}_{t-\tau_{t,i}})\|^2$$

$$\leq m\sigma_L^2 + \|\sum_{i \in \mathcal{M}_t} \Delta_i(\mathbf{x}_{t-\tau_{t,i}})\|^2.$$

Here $\{\mathbf{G}_i(\mathbf{x}_{t-\tau_{t,i}}) - \Delta_i(\mathbf{x}_{t-\tau_{t,i}})\}$ forms a martingale difference sequence.    $\square$

## B  PROOF OF THE PERFORMANCE OF THE AFA-CD ALGORITHM

In this section, we provide the proofs of the theoretical results of the AFA-CD algorithm. We consider two cases: i) general worker information arrival processes and ii) uniformly distributed worker information arrivals. As mentioned earlier, for general worker information arrival processes, we do not make any assumptions on the worker information arrival processes except the independence of workers' participation. For uniformly distributed worker information arrivals, $\mathcal{M}_t$ can be viewed as a subset with size m independently and uniformly sampled from $[M]$ without replacement. The similar convergence analysis for independently and uniformly sampling with replacement can be derived in the same way following the techniques in (Yang et al., 2021; Li et al., 2019c).

### B.1  LOWER BOUND FOR GENERAL WORKER INFORMATION ARRIVAL PROCESSES

**Theorem 1** (Convergence Error Lower Bound for General Worker Information Arrival Processes).
*For any level of heterogeneity characterized by $\sigma_G$, there exists loss functions and worker participation process satisfying Assumptions 1- 3 for which the output $\hat{\mathbf{x}}$ of any randomized FL algorithm satisfies:*

$$\mathbb{E}\|\nabla f(\hat{\mathbf{x}})\|^2 = \Omega(\sigma_G^2), \tag{1}$$

*Proof.* We prove the lower bound by considering a worst-case scenario for simple one-dimensional functions. Let the FL system has two workers with the following loss functions: $f_1(\mathbf{x}) = (\mathbf{x} + G)^2, f_2(\mathbf{x}) = (\mathbf{x} - G)^2, f(\mathbf{x}) = \frac{1}{2}(f_1(\mathbf{x}) + f_2(\mathbf{x})) = \mathbf{x}^2 + G^2$. It is easy to verify that $\|\nabla f_1(\mathbf{x}) - \nabla f(\mathbf{x})\|^2 \le 4G^2 = \sigma_G^2$ and $\|\nabla f_2(\mathbf{x}) - \nabla f(\mathbf{x})\|^2 \le 4G^2 = \sigma_G^2$. We consider a special case for the general arrival process when only the first one worker participates in the training, equivalent to optimizing $f_1(\mathbf{x})$ rather than $f(\mathbf{x})$. In such case, any decent algorithm would return $\hat{\mathbf{x}} = -G + \epsilon$, where $\epsilon$ is a small error term. As a result, $\mathbb{E}\|\nabla f(\hat{\mathbf{x}})\|^2 = \Omega(\sigma_G^2)$. □

### B.2  GENERAL WORKER INFORMATION ARRIVAL PROCESSES

**Theorem 2** (AFA-CD with General Worker Information Arrival Processes). *Under Assumptions 1- 3, choose server-side and worker-side learning rates $\eta$ and $\eta_L$ such that the following relationships hold: $180\eta_L^2 K^2 L^2 \tau < 1$ and $2L\eta\eta_L + 3\tau^2 L^2 \eta^2 \eta_L^2 \le 1$. Then, the output sequence $\{\mathbf{x}_t\}$ generated by AFedAvg-TSLR-CD with general worker information arrival processes satisfies:*

$$\frac{1}{T}\sum_{t=0}^{T-1} \mathbb{E}\|\nabla f(\mathbf{x}_t)\|^2 \le \frac{4(f_0 - f_*)}{\eta\eta_L T} + 4(\alpha_L \sigma_L^2 + \alpha_G \sigma_G^2), \tag{2}$$

*where $\alpha_L \triangleq \left[\frac{2\eta\eta_L}{m} + \frac{3\tau^2 L^2 \eta^2 \eta_L^2}{2m} + \frac{15\eta_L^2 K L^2}{2}\right]$ and $\alpha_G \triangleq \left[\frac{3}{2} + 45K^2 L^2 \eta_L^2\right]$.*

*Proof.* Due to the $L$-smoothness assumption, taking expectation of $f(\mathbf{x}_{t+1})$ over the randomness in communication round $t$, we have:

$$\mathbb{E}[f(\mathbf{x}_{t+1})] \le f(\mathbf{x}_t) + \underbrace{\langle \nabla f(\mathbf{x}_t), \mathbb{E}[\mathbf{x}_{t+1} - \mathbf{x}_t]\rangle}_{A_1} + \frac{L}{2}\underbrace{\mathbb{E}[\|\mathbf{x}_{t+1} - \mathbf{x}_t\|^2]}_{A_2}.$$

First, we bound the term $A_2$ as follows:

$$A_2 = \mathbb{E}\|\mathbf{x}_{t+1} - \mathbf{x}_t\|^2$$
$$= \eta^2 \eta_L^2 \mathbb{E}\|\frac{1}{m}\sum_{i=1}^{m} G_i(\mathbf{x}_{t-\tau_{t,i}})\|^2$$
$$\overset{(a1)}{\le} \frac{\eta^2 \eta_L^2}{m^2}\|\sum_{i=1}^{m} \Delta_i(\mathbf{x}_{t-\tau_{t,i}})\|^2 + \frac{\eta^2 \eta_L^2}{m}\sigma_L^2,$$

where $(a1)$ is due to Lemma 2. Next, we bound the term $A_1$ as follows:

$$A_1 = \langle \nabla f(\mathbf{x}_t), \mathbb{E}[\mathbf{x}_{t+1} - \mathbf{x}_t]\rangle$$

$$= -\eta\eta_L \left\langle \nabla f(\mathbf{x}_t), \mathbb{E}\frac{1}{m}\sum_{i=1}^{m}\mathbf{G}_i(\mathbf{x}_{t-\tau_{t,i}})\right\rangle$$

$$\overset{(a2)}{=} -\frac{1}{2}\eta\eta_L\|\nabla f(\mathbf{x}_t)\|^2 - \frac{1}{2}\eta\eta_L\|\frac{1}{m}\sum_{i=1}^{m}\Delta_i(\mathbf{x}_{t-\tau_{t,i}})\|^2 + \frac{1}{2}\eta\eta_L\underbrace{\|\nabla f(\mathbf{x}_t) - \frac{1}{m}\sum_{i=1}^{m}\Delta_i(\mathbf{x}_{t-\tau_{t,i}})\|^2}_{A_3},$$

where $(a2)$ is due to Lemma 1 and the fact that $\langle \mathbf{x}, \mathbf{y}\rangle = \frac{1}{2}(\|\mathbf{x}\|^2 + \|\mathbf{y}\|^2 - \|\mathbf{x}-\mathbf{y}\|^2)$. To further bound the term $A_3$, we have:

$$A_3 = \|\nabla f(\mathbf{x}_t) - \frac{1}{m}\sum_{i=1}^{m}\Delta_i(\mathbf{x}_{t-\tau_{t,i}})\|^2$$

$$\leq \frac{1}{m}\sum_{i=1}^{m}\|\nabla f(\mathbf{x}_t) - \Delta_i(\mathbf{x}_{t-\tau_{t,i}})\|^2$$

$$= \frac{1}{m}\sum_{i=1}^{m}\|\nabla f(\mathbf{x}_t) - \nabla f(\mathbf{x}_{t-\tau_{t,i}}) + \nabla f(\mathbf{x}_{t-\tau_{t,i}}) - \nabla f_i(\mathbf{x}_{t-\tau_{t,i}}) + \nabla f_i(\mathbf{x}_{t-\tau_{t,i}}) - \Delta_i(\mathbf{x}_{t-\tau_{t,i}})\|^2$$

$$\overset{(a3)}{\leq} \frac{1}{m}\sum_{i=1}^{m}\Big[3\|\nabla f(\mathbf{x}_t) - \nabla f(\mathbf{x}_{t-\tau_{t,i}})\|^2 + 3\|\nabla f(\mathbf{x}_{t-\tau_{t,i}}) - \nabla f_i(\mathbf{x}_{t-\tau_{t,i}})\|^2$$

$$+ 3\|\nabla f_i(\mathbf{x}_{t-\tau_{t,i}}) - \Delta_i(\mathbf{x}_{t-\tau_{t,i}})\|^2\Big]$$

$$\overset{(a4)}{\leq} \underbrace{\frac{3L^2}{m}\sum_{i=1}^{m}\|\mathbf{x}_t - \mathbf{x}_{t-\tau_{t,i}}\|^2}_{A_4} + 3\sigma_G^2 + \frac{3}{m}\sum_{i=1}^{m}\underbrace{\|\nabla f_i(\mathbf{x}_{t-\tau_{t,i}}) - \Delta_i(\mathbf{x}_{t-\tau_{t,i}})\|^2}_{A_5},$$

where $(a3)$ followings from the inequality $\|\mathbf{x}_1 + \mathbf{x}_2 + \cdots + \mathbf{x}_n\|^2 \leq n\sum_{i=1}^{n}\|\mathbf{x}_i\|^2$, and $(a4)$ is due to the L-smoothness assumption (Assumption 1) and bounded global variance assumption (Assumption 3).

To further bound the term $A_4$, we have:

$$A_4 = \frac{1}{m}\sum_{i\in[m]}\|\mathbf{x}_t - \mathbf{x}_{t-\tau_{t,i}}\|^2$$

$$\overset{(a5)}{\leq} \|\mathbf{x}_t - \mathbf{x}_{t-\tau_{t,u}}\|^2$$

$$= \|\sum_{k=t-\tau_{t,u}}^{t-1}\mathbf{x}_{k+1} - \mathbf{x}_k\|^2$$

$$= \mathbb{E}\|\sum_{k=t-\tau_{t,u}}^{t-1}\frac{1}{m}\eta\eta_L\sum_{i\in\mathcal{M}_k}G_i(\mathbf{x}_{k-\tau_{k,i}})\|^2$$

$$= \mathbb{E}\left[\frac{\eta^2\eta_L^2}{m^2}\|\sum_{k=t-\tau_{t,u}}^{t-1}\sum_{i\in\mathcal{M}_k}G_i(\mathbf{x}_{k-\tau_{k,i}})\|^2\right]$$

$$\overset{(a6)}{\leq} \mathbb{E}\left[\frac{\eta^2\eta_L^2}{m^2}\tau\sum_{k=t-\tau_{t,u}}^{t-1}\|\sum_{i\in\mathcal{M}_k}G_i(\mathbf{x}_{k-\tau_{k,i}})\|^2\right]$$

$$\overset{(a7)}{\leq} \mathbb{E}\left[\frac{\eta^2\eta_L^2\tau}{m^2}[(\sum_{k=t-\tau_{t,u}}^{t-1}\|\sum_{i\in\mathcal{M}_k}\Delta_i(\mathbf{x}_{k-\tau_{k,i}})\|^2) + \tau m\sigma_L^2]\right].$$

In the derivations above, we let $u := \operatorname{argmax}_{i \in [M]} \|\mathbf{x}_t - \mathbf{x}_{t-\tau_{t,i}}\|^2$, which yields $(a5)$. Note also that the maximum delay assumption $\tau \geq \tau_{k,i}, \forall i \in [M]$ implies $(a6)$. Lastly, $(a7)$ follows from Lemma 2.

To further bound the term $A_5$, we have:

$$
\begin{aligned}
A_5 &= \|\nabla f_i(\mathbf{x}_{t-\tau_{t,i}}) - \Delta_i(\mathbf{x}_{t-\tau_{t,i}})\|^2 \\
&= \|\nabla f_i(\mathbf{x}_{t-\tau_{t,i}}) - \frac{1}{K_{t,i}} \sum_{j=0}^{K_{t,i}-1} \nabla f_i(\mathbf{x}_{t-\tau_{t,i}}^j)\|^2 \\
&= \frac{1}{K_{t,i}} \sum_{j=0}^{K_{t,i}-1} \|\nabla f_i(\mathbf{x}_{t-\tau_{t,i}}) - \nabla f_i(\mathbf{x}_{t-\tau_{t,i}}^j)\|^2 \\
&\overset{(a8)}{\leq} \frac{L^2}{K_{t,i}} \sum_{j=0}^{K_{t,i}-1} \underbrace{\|\mathbf{x}_{t-\tau_{t,i}} - \mathbf{x}_{t-\tau_{t,i}}^j\|^2}_{A_6} \\
&\overset{(a9)}{\leq} 5K_{t,i}L^2\eta_L^2(\sigma_L^2 + 6K_{t,i}\sigma_G^2) + 30K_{t,i}^2 L^2\eta_L^2\|\nabla f(\mathbf{x}_{t-\tau_{t,i}})\|^2 \\
&\overset{(a10)}{\leq} 5KL^2\eta_L^2(\sigma_L^2 + 6K\sigma_G^2) + 30K^2 L^2\eta_L^2\|\nabla f(\mathbf{x}_{t-\tau_{t,i}})\|^2,
\end{aligned}
$$

where $(a8)$ is due to the $L$-smoothness assumption (Assumption 1), and $(a9)$ follows from the bound of $A_6$ shown below. Here, we denote maximum number of local steps of all workers as $K$, i.e., $K_{t,i} \leq K, \forall t, i$. This definition of $K$ implies $(a10)$.

Now, it remains to bound term $A_6$ in the derivations above. Note that the bounding proof of $A_6$ in what follows is the same as Lemma 4 in (Reddi et al., 2020). we restate the proof here in order for this paper to be self-contained. For any worker $i$ in the $k$-th local step, we have the following results for the norm of parameter changes for one local computation:

$$
\begin{aligned}
A_6 &= \mathbb{E}[\|\mathbf{x}_{t,k}^i - \mathbf{x}_t\|^2] = \mathbb{E}[\|\mathbf{x}_{t,k-1}^i - \mathbf{x}_t - \eta_L g_{t,k-1}^i\|^2] \\
&\leq \mathbb{E}[\|\mathbf{x}_{t,k-1}^i - \mathbf{x}_t - \eta_L(g_{t,k-1}^i - \nabla f_i(\mathbf{x}_{t,k-1}^i) + \nabla f_i(\mathbf{x}_{t,k-1}^i) - \nabla f_i(\mathbf{x}_t) \\
&\quad + \nabla f_i(\mathbf{x}_t) - \nabla f(\mathbf{x}_t) + \nabla f(\mathbf{x}_t))\|^2] \\
&\leq (1 + \frac{1}{2K-1})\mathbb{E}[\|\mathbf{x}_{t,k-1}^i - \mathbf{x}_t\|^2] + \mathbb{E}[\|\eta_L(g_{t,k-1}^i - \nabla f_i(\mathbf{x}_{t,k-1}^i))\|^2] \\
&\quad + 6K\mathbb{E}[\|\eta_L(\nabla f_i(\mathbf{x}_{t,k-1}^i) - \nabla f_i(\mathbf{x}_t))\|^2] + 6K\mathbb{E}[\|\eta_L(\nabla f_i(\mathbf{x}_t) - \nabla f(\mathbf{x}_t))\|^2] \\
&\quad + 6K\|\eta_L\nabla f(\mathbf{x}_t))\|^2 \\
&\leq (1 + \frac{1}{2K-1})\mathbb{E}[\|\mathbf{x}_{t,k-1}^i - \mathbf{x}_t\|^2] + \eta_L^2\sigma_L^2 + 6K\eta_L^2 L^2\mathbb{E}[\|\mathbf{x}_{t,k-1}^i - \mathbf{x}_t\|^2] \\
&\quad + 6K\eta_L^2\sigma_G^2 + 6K\|\eta_L\nabla f(\mathbf{x}_t)\|^2 \\
&= (1 + \frac{1}{2K-1} + 6K\eta_L^2 L^2)\mathbb{E}[\|\mathbf{x}_{t,k-1}^i - \mathbf{x}_t\|^2] + \eta_L^2\sigma_L^2 + 6K\eta_L^2\sigma_G^2 + 6K\|\eta_L\nabla f(\mathbf{x}_t)\|^2 \\
&\overset{(a11)}{\leq} (1 + \frac{1}{K-1})\mathbb{E}[\|\mathbf{x}_{t,k-1}^i - \mathbf{x}_t\|^2] + \eta_L^2\sigma_L^2 + 6K\eta_L^2\sigma_G^2 + 6K\|\eta_L\nabla f(\mathbf{x}_t)\|^2,
\end{aligned}
$$

where $(a11)$ follows from the fact that $\frac{1}{2K-1} + 6K\eta_L^2 L^2 \leq \frac{1}{K-1}$ if $\eta_L^2 \leq \frac{1}{6(2K^2-3K+1)L^2}$.

Unrolling the recursion, we obtain:

$$
\begin{aligned}
\mathbb{E}[\|\mathbf{x}_{t,k}^i - \mathbf{x}_t\|^2] &\leq \sum_{p=0}^{k-1}(1 + \frac{1}{K-1})^p[\eta_L^2\sigma_L^2 + 6K\sigma_G^2 + 6K\eta_L^2\|\eta_L\nabla f(\mathbf{x}_t))\|^2] \\
&\leq (K-1)[(1 + \frac{1}{K-1})^K - 1][\eta_L^2\sigma_L^2 + 6K\eta_L^2\sigma_G^2 + 6K\|\eta_L\nabla f(\mathbf{x}_t))\|^2] \\
&\leq 5K\eta_L^2(\sigma_L^2 + 6K\sigma_G^2) + 30K^2\eta_L^2\|\nabla f(\mathbf{x}_t)\|^2.
\end{aligned} \tag{9}
$$

With the above results of the terms $A_1$ through $A_5$, we have:

$$\mathbb{E}[f(\mathbf{x}_{t+1})] - f(\mathbf{x}_t) \le \underbrace{\langle \nabla f(\mathbf{x}_t), \mathbb{E}[\mathbf{x}_{t+1} - \mathbf{x}_t] \rangle}_{A_1} + \frac{L}{2}\underbrace{\mathbb{E}[\|\mathbf{x}_{t+1} - \mathbf{x}_t\|^2]}_{A_2}$$

$$= -\frac{1}{2}\eta\eta_L\|\nabla f(\mathbf{x}_t)\|^2 - \frac{1}{2}\eta\eta_L\|\frac{1}{m}\sum_{i=1}^m \Delta_i(\mathbf{x}_{t-\tau_{t,i}})\|^2 + \frac{1}{2}\eta\eta_L\underbrace{\|\nabla f(\mathbf{x}_t) - \frac{1}{m}\sum_{i=1}^m \Delta_i(\mathbf{x}_{t-\tau_{t,i}})\|^2}_{A_3}$$

$$+ \frac{L\eta^2\eta_L^2}{m^2}\|\sum_{i=1}^m \Delta_i(\mathbf{x}_{t-\tau_{t,i}})\|^2 + \frac{L\eta^2\eta_L^2}{m}\sigma_L^2$$

$$\le -\frac{1}{2}\eta\eta_L\|\nabla f(\mathbf{x}_t)\|^2 - \frac{1}{2}\eta\eta_L\|\frac{1}{m}\sum_{i=1}^m \Delta_i(\mathbf{x}_{t-\tau_{t,i}})\|^2 + \frac{L\eta^2\eta_L^2}{m^2}\|\sum_{i=1}^m \Delta_i(\mathbf{x}_{t-\tau_{t,i}})\|^2 + \frac{L\eta^2\eta_L^2}{m}\sigma_L^2$$

$$+ \frac{3}{2}\eta\eta_L\sigma_G^2 + \frac{3L^2}{2}\eta\eta_L\underbrace{\left[\frac{1}{m}\sum_{i=1}^m \|\mathbf{x}_t - \mathbf{x}_{t-\tau_{t,i}}\|^2\right]}_{A_4} + \frac{3\eta\eta_L}{2m}\sum_{i=1}^m \underbrace{\|\nabla f_i(\mathbf{x}_{t-\tau_{t,i}}) - \Delta_i(\mathbf{x}_{t-\tau_{t,i}})\|^2}_{A_5}$$

$$\le -\frac{1}{2}\eta\eta_L\|\nabla f(\mathbf{x}_t)\|^2 - \frac{1}{2}\eta\eta_L\|\frac{1}{m}\sum_{i=1}^m \Delta_i(\mathbf{x}_{t-\tau_{t,i}})\|^2 + \frac{L\eta^2\eta_L^2}{m^2}\|\sum_{i=1}^m \Delta_i(\mathbf{x}_{t-\tau_{t,i}})\|^2 + \frac{L\eta^2\eta_L^2}{m}\sigma_L^2$$

$$+ \frac{3}{2}\eta\eta_L\sigma_G^2 + \frac{3L^2}{2}\eta\eta_L\left[\frac{\eta^2\eta_L^2\tau}{m^2}\left(\sum_{k=t-\tau_{t,u}}^{t-1}\|\sum_{i=1}^m \Delta_i(\mathbf{x}_{k-\tau_{k,i}})\|^2 + \tau m\sigma_L^2\right)\right]$$

$$+ \frac{3\eta\eta_L}{2}[5KL^2\eta_L^2(\sigma_L^2 + 6K\sigma_G^2) + 30K^2L^2\eta_L^2\frac{1}{m}\sum_{i=1}^m \|\nabla f(\mathbf{x}_{t-\tau_{t,i}})\|^2]$$

$$\le -\frac{1}{2}\eta\eta_L\|\nabla f(\mathbf{x}_t)\|^2 + 45\eta\eta_L^3 K^2 L^2\frac{1}{m}\sum_{i=1}^m \|\nabla f(\mathbf{x}_{t-\tau_{t,i}})\|^2$$

$$+ \left[-\frac{\eta\eta_L}{2m^2} + \frac{L\eta^2\eta_L^2}{m^2}\right]\|\sum_{i=1}^m \Delta_i(\mathbf{x}_{t-\tau_{t,i}})\|^2 + \frac{3\tau\eta^3\eta_L^3}{2m^2}\sum_{k=t-\tau_{t,u}}^{t-1}\|\sum_{i=1}^m \Delta_i(\mathbf{x}_{k-\tau_{k,i}})\|^2$$

$$+ \left[\frac{L\eta^2\eta_L^2}{m} + \frac{3\tau^2 L^2\eta^3\eta_L^3}{2m} + \frac{15\eta\eta_L^3 KL^2}{2}\right]\sigma_L^2 + \left[\frac{3}{2}\eta\eta_L + 45K^2 L^2\eta\eta_L^3\right]\sigma_G^2.$$

Summing the above inequality from $t = 0$ to $t = T - 1$ yields:

$$\mathbb{E}f(\mathbf{x}_T) - f(\mathbf{x}_0)$$

$$\le \sum_{t=0}^{T-1}\left[-\frac{1}{2}\eta\eta_L\|\nabla f(\mathbf{x}_t)\|^2 + 45\eta\eta_L^3 K^2 L^2\frac{1}{m}\sum_{i=1}^m \|\nabla f(\mathbf{x}_{t-\tau_{t,i}})\|^2\right]$$

$$+ \sum_{t=0}^{T-1}\left[[-\frac{\eta\eta_L}{2m^2} + \frac{L\eta^2\eta_L^2}{m^2}]\|\sum_{i=1}^m \Delta_i(\mathbf{x}_{t-\tau_{t,i}})\|^2 + \frac{3\tau L^2\eta^3\eta_L^3}{2m^2}\sum_{k=t-\tau_{t,u}}^{t-1}\|\sum_{i=1}^m \Delta_i(\mathbf{x}_{k-\tau_{k,i}})\|^2\right]$$

$$+ T\left[\frac{L\eta^2\eta_L^2}{m} + \frac{3\tau^2 L^2\eta^3\eta_L^3}{2m} + \frac{15\eta\eta_L^3 KL^2}{2}\right]\sigma_L^2 + T\left[\frac{3}{2}\eta\eta_L + 45K^2 L^2\eta\eta_L^3\right]\sigma_G^2$$

$$\overset{(a12)}{\le} \sum_{t=0}^{T-1}\left[-\frac{1}{2}\eta\eta_L + 45\eta\eta_L^3 K^2 L^2\tau\right]\|\nabla f(\mathbf{x}_t)\|^2$$

$$+ \sum_{t=0}^{T-1}\left[-\frac{\eta\eta_L}{2m^2} + \frac{L\eta^2\eta_L^2}{m^2} + \frac{3\tau^2 L^2\eta^3\eta_L^3}{2m^2}\right]\|\sum_{i=1}^m \Delta_i(\mathbf{x}_{t-\tau_{t,i}})\|^2$$

$$+ T\left[\frac{L\eta^2\eta_L^2}{m} + \frac{3\tau^2 L^2\eta^3\eta_L^3}{2m} + \frac{15\eta\eta_L^3 KL^2}{2}\right]\sigma_L^2 + T\left[\frac{3}{2}\eta\eta_L + 45K^2 L^2\eta\eta_L^3\right]\sigma_G^2$$

$$\overset{(a13)}{\leq} \sum_{t=0}^{T-1} -\frac{1}{4}\eta\eta_L\|\nabla f(\mathbf{x}_t)\|^2$$

$$+ T\eta\eta_L\left[\frac{L\eta\eta_L}{m} + \frac{3\tau^2 L^2\eta^2\eta_L^2}{2m} + \frac{15\eta_L^2 KL^2}{2}\right]\sigma_L^2 + T\eta\eta_L\left[\frac{3}{2} + 45K^2 L^2\eta_L^2\right]\sigma_G^2$$

$$\overset{(a14)}{=} \sum_{t=0}^{T-1} -\frac{1}{4}\eta\eta_L\|\nabla f(\mathbf{x}_t)\|^2 + T\eta\eta_L\left[\alpha_L\sigma_L^2 + \alpha_G\sigma_G^2\right],$$

where $(a12)$ is due to maximum time delay $\tau$ in the system, $(a13)$ holds if $\frac{1}{4} \leq [\frac{1}{2} - 45\eta_L^2 K^2 L^2\tau]$, i.e., $180\eta_L^2 K^2 L^2\tau < 1$, and $\left[-\frac{\eta\eta_L}{2m^2} + \frac{L\eta^2\eta_L^2}{m^2} + \frac{3L^2\tau^2\eta^3\eta_L^3}{2m^2}\right] \leq 0$, i.e., $2L\eta\eta_L + 3\tau^2 L^2\eta^2\eta_L^2 \leq 1$.

Lastly, $(a14)$ follows from the following definitions: $\alpha_L = \left[\frac{2\eta\eta_L}{m} + \frac{3\tau^2 L^2\eta^2\eta_L^2}{2m} + \frac{15\eta_L^2 KL^2}{2}\right], \alpha_G = \left[\frac{3}{2} + 45K^2 L^2\eta_L^2\right]$. Rearranging terms, we have:

$$\frac{1}{T}\sum_{t=0}^{T-1}\|\nabla f(\mathbf{x}_t)\|^2 \leq \frac{4(f_0 - f_*)}{\eta\eta_L T} + 4\left[\alpha_L\sigma_L^2 + \alpha_G\sigma_G^2\right],$$

and the proof is complete. □

**Corollary 1** (Linear Speedup to Error Ball). *By setting $\eta_L = \frac{1}{\sqrt{T}}$, and $\eta = \sqrt{m}$, the convergence rate of AFA-CD with general worker information arrival processes is:*

$$\frac{1}{T}\sum_{t=0}^{T-1}\mathbb{E}\|\nabla f(\mathbf{x}_t)\|^2 = \mathcal{O}\left(\frac{1}{m^{1/2}T^{1/2}}\right) + \mathcal{O}\left(\frac{\tau^2}{T}\right) + \mathcal{O}\left(\frac{K^2}{T}\right) + \mathcal{O}(\sigma_G^2).$$

*Proof.* Let $\eta_L = \frac{1}{\sqrt{T}}$, and $\eta = \sqrt{m}$. It then follows that:

$$\alpha_L = \mathcal{O}(\frac{1}{m^{1/2}T^{1/2}}) + \mathcal{O}(\frac{\tau^2}{T}) + \mathcal{O}(\frac{K}{T}).$$

$$\alpha_G = \mathcal{O}(\sigma_G^2) + \mathcal{O}(\frac{K^2}{T}).$$

This completes the proof. □

### B.3 UNIFORMLY DISTRIBUTED WORKER INFORMATION ARRIVALS

Now, we consider the special case where all workers have a statistically identical speed so that the worker information arrivals are uniformly distributed. As mentioned earlier, this special case acts as a widely-used assumption in FL and could deepen our understanding on the AFA-CD algorithm's performance in large-scale AFL systems.

**Theorem 3.** *Under Assumptions 1- 3, choose server-side and worker-side learning rates $\eta$ and $\eta_L$ such that the following relationships hold: $\eta_L^2[6(2K^2 - 3K + 1)L^2] \leq 1$, $120L^2 K^2\eta_L^2\tau + 4(L\eta\eta_L + L^2\eta^2\eta_L^2\tau^2)\frac{M-m}{m(M-1)}(90K^2 L^2\eta_L^2\tau + 3\tau) < 1$. Then, the output sequence $\{\mathbf{x}_t\}$ generated by AFA-CD with uniformly distributed worker information arrivals satisfies:*

$$\frac{1}{T}\sum_{t=0}^{T-1}\mathbb{E}\|\nabla f(\mathbf{x}_t)\|_2^2 \leq \frac{4(f_0 - f_*)}{\eta\eta_L T} + 4\left(\alpha_L\sigma_L^2 + \alpha_G\sigma_G^2\right), \tag{3}$$

*where $\alpha_L$ and $\alpha_G$ are constants that are defined as:*

$$\alpha_L \triangleq \left[\left(\frac{L\eta\eta_L}{m} + \frac{\tau^2 L^2\eta^2\eta_L^2}{m} + 5KL^2\eta_L^2\right) + (L\eta\eta_L + L^2\eta^2\eta_L^2\tau^2)\frac{M-m}{m(M-1)}(15KL^2\eta_L^2)\right],$$

$$\alpha_G \triangleq \left[30K^2 L^2\eta_L^2 + (L\eta\eta_L + L^2\eta^2\eta_L^2\tau^2)\frac{M-m}{m(M-1)}(90K^2 L^2\eta_L^2 + 3)\right].$$

*Proof.* The one-step update can be rewritten as: $\mathbf{x}_{t+1} - \mathbf{x}_t = -\eta\eta_L\mathbf{G}_t$. For cross-device FL, $\mathbf{G}_t = \frac{1}{m}\sum_{i\in\mathcal{M}_t}\mathbf{G}_i(\mathbf{x}_{t-\tau_{t,i}})$, where $\tau_{t,i}$ is the delay for client $i$ in terms of the current global communication round $t$. When $\tau_{t,i} = 0, \forall i \in \mathcal{M}_t$, it degenerates to synchronous FL with partial worker participation.

Due to the $L$-smoothness in Assumption 1 , taking expectation of $f(\mathbf{x}_{t+1})$ over the randomness in communication round $t$, we have:

$$\mathbb{E}[f(\mathbf{x}_{t+1})] \leq f(\mathbf{x}_t) + \underbrace{\langle\nabla f(\mathbf{x}_t), \mathbb{E}[\mathbf{x}_{t+1} - \mathbf{x}_t]\rangle}_{A_1} + \frac{L}{2}\underbrace{\mathbb{E}[\|\mathbf{x}_{t+1} - \mathbf{x}_t\|^2}_{A_2}$$

We first bound $A_2$ as follows:

$$\begin{aligned}
A_2 &= \mathbb{E}\|\mathbf{x}_{t+1} - \mathbf{x}_t\|^2 \\
&= \eta^2\eta_L^2\mathbb{E}\|\frac{1}{m}\sum_{i\in\mathcal{M}_t}G_i(\mathbf{x}_{t-\tau_{t,i}})\|^2 \\
&\overset{(b1)}{\leq} \frac{\eta^2\eta_L^2}{m^2}\mathbb{E}\left[\|\sum_{i=1}^m\Delta_i(\mathbf{x}_{t-\tau_{t,i}})\|^2 + m\sigma_L^2\right] \\
&\overset{(b2)}{\leq} \frac{\eta^2\eta_L^2}{m^2}\mathbb{E}\|\sum_{i=1}^M\mathbb{I}\{i\in\mathcal{M}_t\}\Delta_i(\mathbf{x}_{t-\tau_{t,i}})\|^2 + \frac{\eta^2\eta_L^2}{m}\sigma_L^2,
\end{aligned}$$

where $(b1)$ is due to Lemma 2 and $(b2)$ is due to the uniformly independent information arrival assumption.

To bound the term $A_1$, we have:

$$\begin{aligned}
A_1 &= \langle\nabla f(\mathbf{x}_t), \mathbb{E}[\mathbf{x}_{t+1} - \mathbf{x}_t]\rangle \\
&= -\eta\eta_L\langle\nabla f(\mathbf{x}_t), \mathbb{E}\frac{1}{m}\sum_{i\in\mathcal{M}_t}\mathbf{G}_i(\mathbf{x}_{t-\tau_{t,i}})\rangle \\
&\overset{(b3)}{=} -\eta\eta_L\langle\nabla f(\mathbf{x}_t), \frac{1}{M}\sum_{i\in[M]}\Delta_i(\mathbf{x}_{t-\tau_{t,i}})\rangle \\
&\overset{(b4)}{=} -\frac{1}{2}\eta\eta_L\|\nabla f(\mathbf{x}_t)\|^2 - \frac{1}{2}\eta\eta_L\|\frac{1}{M}\sum_{i\in[M]}\Delta_i(\mathbf{x}_{t-\tau_{t,i}})\|^2 \\
&\quad + \frac{1}{2}\eta\eta_L\underbrace{\left\|\nabla f(\mathbf{x}_t) - \frac{1}{M}\sum_{i\in[M]}\Delta_i(\mathbf{x}_{t-\tau_{t,i}})\right\|^2}_{A_3},
\end{aligned}$$

where $(b3)$ is due to the uniformly independent worker information arrival assumption and Lemma 1, $(b4)$ is due to the fact that $\langle\mathbf{x}, \mathbf{y}\rangle = \frac{1}{2}(\|\mathbf{x}\|^2 + \|\mathbf{y}\|^2 - \|\mathbf{x} - \mathbf{y}\|^2)$.

To further bound the term $A_3$, we have:

$$\begin{aligned}
A_3 &= \|\nabla f(\mathbf{x}_t) - \frac{1}{M}\sum_{i\in[M]}\Delta_i(\mathbf{x}_{t-\tau_{t,i}})\|^2 \\
&\overset{(b5)}{=} \|\frac{1}{M}\sum_{i\in[M]}[\nabla f_i(\mathbf{x}_t) - \Delta_i(\mathbf{x}_{t-\tau_{t,i}})]\|^2 \\
&\leq \frac{1}{M}\sum_{i\in[M]}\|\nabla f_i(\mathbf{x}_t) - \Delta_i(\mathbf{x}_{t-\tau_{t,i}})\|^2 \\
&= \frac{1}{M}\sum_{i\in[M]}\|\nabla f_i(\mathbf{x}_t) - \nabla f_i(\mathbf{x}_{t-\tau_{t,i}}) + \nabla f_i(\mathbf{x}_{t-\tau_{t,i}}) - \Delta_i(\mathbf{x}_{t-\tau_{t,i}})\|^2
\end{aligned}$$

$$\overset{(b6)}{\leq} \frac{1}{M} \sum_{i \in [M]} \left[ 2\|\nabla f_i(\mathbf{x}_t) - \nabla f_i(\mathbf{x}_{t-\tau_{t,i}})\|^2 + 2\|\nabla f_i(\mathbf{x}_{t-\tau_{t,i}}) - \Delta_i(\mathbf{x}_{t-\tau_{t,i}})\|^2 \right]$$

$$\overset{(b7)}{\leq} \underbrace{\frac{2L^2}{M} \sum_{i=1}^{M} \|\mathbf{x}_t - \mathbf{x}_{t-\tau_{t,i}}\|^2}_{A_4} + \frac{2}{M} \sum_{i=1}^{M} \underbrace{\|\nabla f_i(\mathbf{x}_{t-\tau_{t,i}}) - \Delta_i(\mathbf{x}_{t-\tau_{t,i}})\|^2}_{A_5},$$

where $(b5)$ is due to the fact that $\nabla f(\mathbf{x}) = \frac{1}{M} \sum_{i \in [M]} \nabla f_i(\mathbf{x})$, $(b6)$ follows from the inequality $\|\mathbf{x}_1 + \mathbf{x}_2 + \cdots + \mathbf{x}_n\|^2 \leq n \sum_{i=1}^{n} \|\mathbf{x}_i\|^2$, and $(b7)$ follows from the $L$-smoothness assumption (Assumption 1).

For $A_4$ and $A_5$, we have the same bounds as in the case of general worker information arrival processes:

$$A_4 \leq \mathbb{E}\left[ \frac{\eta^2 \eta_L^2 \tau}{m^2} \left[ \left( \sum_{k=t-\tau_{t,\mu}}^{t-1} \| \sum_{i \in \mathcal{M}_k} \Delta_i(\mathbf{x}_{k-\tau_{k,i}})\|^2 \right) + \tau m \sigma_L^2 \right] \right]$$

$$\leq \frac{\eta^2 \eta_L^2 \tau}{m^2} \left[ \left( \sum_{k=t-\tau_{t,\mu}}^{t-1} \mathbb{E}\| \sum_{i=1}^{M} \mathbb{I}\{i \in \mathcal{M}_k\} \Delta_i(\mathbf{x}_{k-\tau_{k,i}})\|^2 \right) + \tau m \sigma_L^2 \right].$$

$$A_5 \leq 5KL^2 \eta_L^2 (\sigma_L^2 + 6K\sigma_G^2) + 30K^2 L^2 \eta_L^2 \|\nabla f(\mathbf{x}_{t-\tau_{t,i}})\|^2,$$

With the above results of the term $A_1$ through $A_5$, we have:

$$\mathbb{E}_t[f(\mathbf{x}_{t+1})] - f(\mathbf{x}_t) \leq \underbrace{\langle \nabla f(\mathbf{x}_t), \mathbb{E}_t[\mathbf{x}_{t+1} - \mathbf{x}_t] \rangle}_{A_1} + \frac{L}{2} \underbrace{\mathbb{E}_t[\|\mathbf{x}_{t+1} - \mathbf{x}_t\|^2]}_{A_2}$$

$$= -\frac{1}{2}\eta\eta_L \|\nabla f(\mathbf{x}_t)\|^2 - \frac{1}{2}\eta\eta_L \frac{1}{M} \sum_{i \in [M]} \Delta_i(\mathbf{x}_{t-\tau_{t,i}})\|^2 + \frac{1}{2}\eta\eta_L \underbrace{\|\nabla f(\mathbf{x}_t) - \frac{1}{M} \sum_{i \in [M]} \Delta_i(\mathbf{x}_{t-\tau_{t,i}})\|^2}_{A_3}$$

$$+ \frac{L\eta^2 \eta_L^2}{m^2} \mathbb{E}\| \sum_{i=1}^{M} \mathbb{I}\{i \in \mathcal{M}_t\} \Delta_i(\mathbf{x}_{t-\tau_{t,i}})\|^2 + \frac{L\eta^2 \eta_L^2}{m} \sigma_L^2$$

$$\leq -\frac{1}{2}\eta\eta_L \|\nabla f(\mathbf{x}_t)\|^2 - \frac{1}{2}\eta\eta_L \frac{1}{M} \sum_{i \in [M]} \Delta_i(\mathbf{x}_{t-\tau_{t,i}})\|^2 + \frac{L\eta^2 \eta_L^2}{m^2} \mathbb{E}\| \sum_{i=1}^{M} \mathbb{I}\{i \in \mathcal{M}_t\} \Delta_i(\mathbf{x}_{t-\tau_{t,i}})\|^2$$

$$+ \frac{1}{2}\eta\eta_L \left[ \underbrace{\frac{2L^2}{M} \sum_{i=1}^{M} \|\mathbf{x}_t - \mathbf{x}_{t-\tau_{t,i}}\|^2}_{A_4} + \frac{2}{M} \sum_{i=1}^{M} \underbrace{\|\nabla f_i(\mathbf{x}_{t-\tau_{t,i}}) - \Delta_i(\mathbf{x}_{t-\tau_{t,i}})\|^2}_{A_5} \right] + \frac{L\eta^2 \eta_L^2}{m} \sigma_L^2$$

$$\leq -\frac{1}{2}\eta\eta_L \|\nabla f(\mathbf{x}_t)\|^2 - \frac{1}{2}\eta\eta_L \frac{1}{M} \sum_{i \in [M]} \Delta_i(\mathbf{x}_{t-\tau_{t,i}})\|^2 + \frac{L\eta^2 \eta_L^2}{m^2} \mathbb{E}\| \sum_{i=1}^{M} \mathbb{I}\{i \in \mathcal{M}_t\} \Delta_i(\mathbf{x}_{t-\tau_{t,i}})\|^2$$

$$+ \eta\eta_L L^2 \left[ \frac{\eta^2 \eta_L^2 \tau}{m^2} \left( \sum_{k=t-\tau_{t,\mu}}^{t-1} \mathbb{E}\| \sum_{i=1}^{M} \mathbb{I}\{i \in \mathcal{M}_k\} \Delta_i(\mathbf{x}_{k-\tau_{k,i}})\|^2 + \tau m \sigma_L^2 \right) \right]$$

$$+ \eta\eta_L \left[ 5KL^2 \eta_L^2 (\sigma_L^2 + 6K\sigma_G^2) + 30K^2 L^2 \eta_L^2 \frac{1}{M} \sum_{i \in [M]} \|\nabla f(\mathbf{x}_{t-\tau_{t,i}})\|^2 \right] + \frac{L\eta^2 \eta_L^2}{m} \sigma_L^2$$

$$\overset{(b8)}{\leq} \left[ -\frac{1}{2}\eta\eta_L \|\nabla f(\mathbf{x}_t)\|^2 + (30\eta K^2 L^2 \eta_L^3) \|\nabla f(\mathbf{x}_{t-\tau_{t,j}})\|^2 \right]$$

$$+ \left[ -\frac{\eta\eta_L}{2M^2} \| \sum_{i=1}^{M} \Delta_i(\mathbf{x}_{t-\tau_{t,i}})\|^2 + \frac{L\eta^2 \eta_L^2}{m^2} \mathbb{E}\| \sum_{i=1}^{M} \mathbb{I}\{i \in \mathcal{M}_t\} \Delta_i(\mathbf{x}_{t-\tau_{t,i}})\|^2 \right]$$

$$+ \frac{L^2\eta^3\eta_L^3\tau}{m^2} \sum_{k=t-\tau_{t,\mu}}^{t-1} \mathbb{E}\|\sum_{i=1}^{M} \mathbb{I}\{i \in \mathcal{M}_k\}\Delta_i(\mathbf{x}_{k-\tau_{k,i}})\|^2\Big]$$

$$+ \sigma_L^2\Big[\frac{L\eta^2\eta_L^2}{m} + \frac{\tau^2 L^2\eta^3\eta_L^3}{m} + 5K\eta L^2\eta_L^3\Big] + 30\eta K^2 L^2\eta_L^3\sigma_G^2,$$

where $(b8)$ follows from $j := \text{argmax}_{i\in[M]}\|\nabla f(\mathbf{x}_{t-\tau_{t,i}})\|^2$. Note $j$ is dependent on $t$ but we omit it for brevity.

Summing the above inequality from $t=0$ to $t=T-1$ yields:

$$\mathbb{E}f(\mathbf{x}_T) - f(\mathbf{x}_0)$$

$$\leq \sum_{t=0}^{T-1}\Big[-\frac{1}{2}\eta\eta_L\|\nabla f(\mathbf{x}_t)\|^2 + (30\eta K^2 L^2\eta_L^3)\|\nabla f(\mathbf{x}_{t-\tau_{t,j}})\|^2\Big]$$

$$+ \sum_{t=0}^{T-1}\Big[-\frac{\eta\eta_L}{2M^2}\|\sum_{i\in[M]}\Delta_i(\mathbf{x}_{t-\tau_{t,i}})\|^2 + \frac{L\eta^2\eta_L^2}{m^2}\mathbb{E}\|\sum_{i=1}^{M}\mathbb{I}\{i\in\mathcal{M}_t\}\Delta_i(\mathbf{x}_{t-\tau_{t,i}})\|^2$$

$$+ \frac{L^2\eta^3\eta_L^3\tau}{m^2}\sum_{k=t-\tau_{t,\mu}}^{t-1}\mathbb{E}\|\sum_{i\in[M]}\mathbb{I}\{i\in\mathcal{M}_k\}\Delta_i(\mathbf{x}_{k-\tau_{k,i}})\|^2\Big]$$

$$+ T\Big[\sigma_L^2(\frac{L\eta^2\eta_L^2}{m} + \frac{\tau^2 L^2\eta^3\eta_L^3}{m} + 5K\eta L^2\eta_L^3) + 30\eta K^2 L^2\eta_L^3\sigma_G^2\Big]$$

$$\overset{(b9)}{\leq} \sum_{t=0}^{T-1}\Big[-\frac{1}{2}\eta\eta_L\|\nabla f(\mathbf{x}_t)\|^2 + (30\eta K^2 L^2\eta_L^3\tau)\|\nabla f(\mathbf{x}_t)\|^2\Big]$$

$$+ \sum_{t=0}^{T-1}\Big[-\frac{\eta\eta_L}{2M^2}\|\sum_{i\in[M]}\Delta_i(\mathbf{x}_{t-\tau_{t,i}})\|^2 + \frac{L\eta^2\eta_L^2}{m^2}\mathbb{E}\|\sum_{i\in[M]}\mathbb{I}\{i\in\mathcal{M}_t\}\Delta_i(\mathbf{x}_{t-\tau_{t,i}})\|^2$$

$$+ \frac{L^2\eta^3\eta_L^3\tau^2}{m^2}\mathbb{E}\|\sum_{i\in[M]}\mathbb{I}\{i\in\mathcal{M}_t\}\Delta_i(\mathbf{x}_{t-\tau_{t,i}})\|^2\Big]$$

$$+ T\Big[\sigma_L^2(\frac{L\eta^2\eta_L^2}{m} + \frac{\tau^2 L^2\eta^3\eta_L^3}{m} + 5K\eta L^2\eta_L^3) + 30\eta K^2 L^2\eta_L^3\sigma_G^2\Big],$$

where $(b9)$ is due to the fact that the delay in the system is less than $\tau$.

By letting $\mathbf{z}_i = \Delta_i(\mathbf{x}_{t-\tau_{t,i}})$ (omitting the communication round index $t$ for notation simplicity), we have that:

$$\|\sum_{i=1}^{M}\mathbf{z}_i\|^2 = \sum_{i\in[M]}\|\mathbf{z}_i\|^2 + \sum_{i\neq j}\langle\mathbf{z}_i, \mathbf{z}_j\rangle,$$

$$\overset{(b10)}{=} \sum_{i\in[M]}M\|\mathbf{z}_i\|^2 - \frac{1}{2}\sum_{i\neq j}\|\mathbf{z}_i - \mathbf{z}_j\|^2,$$

$$\mathbb{E}\|\sum_{i=1}^{M}\mathbb{I}\{i\in\mathcal{M}_t\}\mathbf{z}_i\|^2 = \sum_{i\in[M]}\mathbb{P}\{i\in\mathcal{M}_t\}\|\mathbf{z}_i\|^2 + \sum_{i\neq j}\mathbb{P}\{i,j\in\mathcal{M}_t\}\langle\mathbf{z}_i, \mathbf{z}_j\rangle$$

$$\overset{(b11)}{=} \frac{m}{M}\sum_{i\in[M]}\|\mathbf{z}_i\|^2 + \frac{m(m-1)}{M(M-1)}\sum_{i\neq j}\langle\mathbf{z}_i, \mathbf{z}_j\rangle$$

$$\overset{(b12)}{=} \frac{m^2}{M}\sum_{i\in[M]}\|\mathbf{z}_i\|^2 - \frac{m(m-1)}{2M(M-1)}\sum_{i\neq j}\|\mathbf{z}_i - \mathbf{z}_j\|^2,$$

where $(b10)$ and $(b12)$ are due to the fact that $\langle\mathbf{x}, \mathbf{y}\rangle = \frac{1}{2}[\|\mathbf{x}\|^2 + \|\mathbf{y}\|^2 - \|\mathbf{x}-\mathbf{y}\|^2] \leq \frac{1}{2}[\|\mathbf{x}\|^2 + \|\mathbf{y}\|^2]$, $(b11)$ follows from the fact that $\mathbb{P}\{i\in\mathcal{M}_t\} = \frac{m}{M}$ and $\mathbb{P}\{i,j\in\mathcal{M}_t\} = \frac{m(m-1)}{M(M-1)}$. It then follows

that:

$$- \frac{\eta \eta_L}{2M^2} \| \sum_{i=1}^M \mathbf{z}_i \|^2 + \frac{L\eta^2 \eta_L^2}{m^2} \mathbb{E}\| \sum_{i=1}^M \mathbb{I}\{i \in \mathcal{M}_t\}\mathbf{z}_i \|^2 + \frac{L^2\eta^3\eta_L^3\tau^2}{m^2} \mathbb{E}\| \sum_{i=1}^M \mathbb{I}\{i \in \mathcal{M}_t\}\mathbf{z}_i \|^2$$

$$= \left[ -\frac{\eta\eta_L}{2M} + (\frac{L\eta^2\eta_L^2}{M} + \frac{L^2\eta^3\eta_L^3\tau^2}{M}) \right] \sum_{i=1}^M \|\mathbf{z}_i\|^2$$

$$+ \left[ \frac{\eta\eta_L}{4M^2} - (\frac{L\eta^2\eta_L^2}{m^2} + \frac{L^2\eta^3\eta_L^3\tau^2}{m^2})\frac{m(m-1)}{2M(M-1)} \right] \sum_{i \neq j} \|\mathbf{z}_i - \mathbf{z}_j\|^2$$

$$\leq \left[ -\frac{\eta\eta_L}{2M} + (\frac{L\eta^2\eta_L^2}{M} + \frac{L^2\eta^3\eta_L^3\tau^2}{M}) + (\frac{\eta\eta_L}{2M} - (\frac{L\eta^2\eta_L^2}{m^2} + \frac{L^2\eta^3\eta_L^3\tau^2}{m^2})\frac{m(m-1)}{(M-1)}) \right] \sum_{i=1}^M \|\mathbf{z}_i\|^2$$

$$= \left[ (\frac{L\eta^2\eta_L^2}{M} + \frac{L^2\eta^3\eta_L^3\tau^2}{M}) - (\frac{L\eta^2\eta_L^2}{m^2} + \frac{L^2\eta^3\eta_L^3\tau^2}{m^2})\frac{m(m-1)}{(M-1)} \right] \sum_{i=1}^M \|\mathbf{z}_i\|^2$$

$$= \left[ (L\eta^2\eta_L^2 + L^2\eta^3\eta_L^3\tau^2)\frac{M-m}{mM(M-1)} \right] \sum_{i=1}^M \|\mathbf{z}_i\|^2.$$

Note also that:
$$\|\mathbf{z}_i\|^2 = \|\Delta_i(\mathbf{x}_{t-\tau_{t,i}})\|^2$$
$$= \|\Delta_i(\mathbf{x}_{t-\tau_{t,i}}) - \nabla f_i(\mathbf{x}_{t-\tau_{t,i}}) + \nabla f_i(\mathbf{x}_{t-\tau_{t,i}}) - \nabla f(\mathbf{x}_{t-\tau_{t,i}}) + \nabla f(\mathbf{x}_{t-\tau_{t,i}})\|^2$$
$$\leq 3\|\Delta_i(\mathbf{x}_{t-\tau_{t,i}}) - \nabla f_i(\mathbf{x}_{t-\tau_{t,i}})\|^2 + 3\|\nabla f_i(\mathbf{x}_{t-\tau_{t,i}}) - \nabla f(\mathbf{x}_{t-\tau_{t,i}})\|^2 + 3\|\nabla f(\mathbf{x}_{t-\tau_{t,i}})\|^2$$
$$\leq 3\underbrace{\|\nabla f_i(\mathbf{x}_{t-\tau_{t,i}}) - \Delta_i(\mathbf{x}_{t-\tau_{t,i}})\|^2}_{A_5} + 3\sigma_G^2 + 3\|\nabla f(\mathbf{x}_{t-\tau_{t,i}})\|^2.$$

Using the above results, we finally have:

$$\mathbb{E}f(\mathbf{x}_T) - f(\mathbf{x}_0)$$

$$\leq \sum_{t=0}^{T-1} \left[ -\frac{1}{2}\eta\eta_L\|\nabla f(\mathbf{x}_t)\|^2 + (30\eta K^2 L^2\eta_L^3\tau)\|\nabla f(\mathbf{x}_t)\|^2 \right]$$

$$+ \sum_{t=0}^{T-1} \left[ -\frac{\eta\eta_L}{2M^2}\| \sum_{i=1}^M \Delta_i(\mathbf{x}_{t-\tau_{t,i}})\|^2 + \frac{L\eta^2\eta_L^2}{m^2}\mathbb{E}\| \sum_{i=1}^M \mathbb{I}\{i \in \mathcal{M}_t\}\Delta_i(\mathbf{x}_{t-\tau_{t,i}})\|^2 \right.$$

$$\left. + \frac{L^2\eta^3\eta_L^3\tau^2}{m^2}\mathbb{E}\| \sum_{i=1}^M \mathbb{I}\{i \in \mathcal{M}_t\}\Delta_i(\mathbf{x}_{k-\tau_{k,i}})\|^2 \right]$$

$$+ T\left[ \sigma_L^2(\frac{L\eta^2\eta_L^2}{m} + \frac{\tau^2 L^2\eta^3\eta_L^3}{m} + 5K\eta L^2\eta_L^3) + 30\eta K^2 L^2\eta_L^3\sigma_G^2 \right]$$

$$\leq \sum_{t=0}^{T-1} \left[ -\frac{1}{2}\eta\eta_L\|\nabla f(\mathbf{x}_t)\|^2 + (30\eta K^2 L^2\eta_L^3\tau)\|\nabla f(\mathbf{x}_t)\|^2 \right]$$

$$+ \sum_{t=0}^{T-1} \left[ \left[ (L\eta^2\eta_L^2 + L^2\eta^3\eta_L^3\tau^2)\frac{M-m}{m(M-1)} \right] \left[ (15KL^2\eta_L^2(\sigma_L^2 + 6K\sigma_G^2) \right. \right.$$

$$\left. \left. + 90K^2 L^2\eta_L^2\frac{1}{M}\sum_{i \in [M]}\|\nabla f(\mathbf{x}_{t-\tau_{t,i}})\|^2 + 3\sigma_G^2 + \frac{3}{M}\sum_{i=1}^M\|\nabla f(\mathbf{x}_{t-\tau_{t,i}})\|^2) \right] \right]$$

$$+ T\left[ \sigma_L^2(\frac{L\eta^2\eta_L^2}{m} + \frac{\tau^2 L^2\eta^3\eta_L^3}{m} + 5K\eta L^2\eta_L^3) + 30\eta K^2 L^2\eta_L^3\sigma_G^2 \right]$$

$$\leq \sum_{t=0}^{T-1} \eta\eta_L\|\nabla f(\mathbf{x}_t)\|^2 \times$$

$$\left[ -\frac{1}{2} + 30L^2K^2\eta_L^2\tau + \left[ (L\eta\eta_L + L^2\eta^2\eta_L^2\tau^2)\frac{M-m}{m(M-1)} \right](90K^2L^2\eta_L^2\tau + 3\tau) \right]$$

$$+ T\eta\eta_L\sigma_L^2 \times$$

$$\left[ \left[ (\frac{L\eta\eta_L}{m} + \frac{\tau^2L^2\eta^2\eta_L^2}{m} + 5KL^2\eta_L^2) + (L\eta\eta_L + L^2\eta^2\eta_L^2\tau^2)\frac{M-m}{m(M-1)}(15KL^2\eta_L^2) \right] \right.$$

$$\left. + \left[ 30K^2L^2\eta_L^2 + (L\eta\eta_L + L^2\eta^2\eta_L^2\tau^2)\frac{M-m}{m(M-1)}(90K^2L^2\eta_L^2 + 3) \right]\sigma_G^2 \right]$$

$$\overset{(b13)}{\leq} \sum_{t=0}^{T-1} -\frac{1}{4}\eta\eta_L\|\nabla f(\mathbf{x}_t)\|^2 + T\eta\eta_L\left[ \alpha_L\sigma_L^2 + \alpha_G\sigma_G^2 \right],$$

where $(b13)$ follows from the fact that

$$\frac{1}{4} \leq \left[ \frac{1}{2} - 30L^2K^2\eta_L^2\tau - \left[ (L\eta\eta_L + L^2\eta^2\eta_L^2\tau^2)\frac{M-m}{m(M-1)} \right](90K^2L^2\eta_L^2\tau + 3\tau) \right]$$

if $120L^2K^2\eta_L^2\tau + 4(L\eta\eta_L + L^2\eta^2\eta_L^2\tau^2)\frac{M-m}{m(M-1)}](90K^2L^2\eta_L^2\tau + 3\tau) < 1,$

$$\alpha_L = \left[ (\frac{L\eta\eta_L}{m} + \frac{\tau^2L^2\eta^2\eta_L^2}{m} + 5KL^2\eta_L^2) + (L\eta\eta_L + L^2\eta^2\eta_L^2\tau^2)\frac{M-m}{m(M-1)}(15KL^2\eta_L^2) \right],$$

and

$$\alpha_G = \left[ 30K^2L^2\eta_L^2 + (L\eta\eta_L + L^2\eta^2\eta_L^2\tau^2)\frac{M-m}{m(M-1)}(90K^2L^2\eta_L^2 + 3) \right].$$

Lastly, by rearranging and telescoping, we have

$$\frac{1}{T}\sum_{t=0}^{T-1}\mathbb{E}\|\nabla f(\mathbf{x}_t)\|^2 \leq \frac{4(f_0 - f_*)}{\eta\eta_L T} + 4\left[ \alpha_L\sigma_L^2 + \alpha_G\sigma_G^2 \right].$$

This completes the proof. $\qquad\square$

**Corollary 2** (Linear Speedup to Stationary Point). *By setting $\eta_L = \frac{1}{\sqrt{T}}$ and $\eta = \sqrt{m}$, the convergence rate of AFA-CD with uniformly distributed worker information arrivals is:*

$$\frac{1}{T}\sum_{t=0}^{T-1}\mathbb{E}\|\nabla f(\mathbf{x}_t)\|_2^2 = \mathcal{O}(\frac{1}{m^{1/2}T^{1/2}}) + \mathcal{O}\left(\frac{\tau^2}{T}\right) + \mathcal{O}\left(\frac{K^2}{T}\right) + \mathcal{O}\left(\frac{K^2}{m^{1/2}T^{3/2}}\right) + \mathcal{O}\left(\frac{K^2\tau^2}{T^2}\right).$$

*Proof.* Let $\eta_L = \frac{1}{\sqrt{T}}$, and $\eta = \sqrt{m}$. It then follows that:

$$\alpha_L = \mathcal{O}(\frac{1}{m^{1/2}T^{1/2}}) + \mathcal{O}(\frac{\tau^2}{T}) + \mathcal{O}(\frac{K}{T}) + \mathcal{O}(\frac{K}{m^{1/2}T^{3/2}}) + \mathcal{O}(\frac{K\tau^2}{T^2}),$$

$$\alpha_G = \mathcal{O}(\frac{1}{m^{1/2}T^{1/2}}) + \mathcal{O}(\frac{\tau^2}{T}) + \mathcal{O}(\frac{K^2}{T}) + \mathcal{O}(\frac{K^2}{m^{1/2}T^{3/2}}) + \mathcal{O}(\frac{K^2\tau^2}{T^2}),$$

and the proof is complete. $\qquad\square$

## C  PROOF OF THE PERFORMANCE RESULTS OF THE AFA-CS ALGORITHM

**Theorem 4.** *Under Assumptions 1- 3, choose sever- and worker-side learning rates $\eta$ and $\eta_L$ in such a way that there exists a non-negative constant series $\{\beta_\mu\}_{u=0}^{\tau-1}$ satisfying the following relationship:*

$$12L\eta\eta_L + \frac{540(M-m')^2}{M^2}(1 + L\eta\eta_L)K^2L^2\eta_L^2(1 + \tau) + 180K^2L^2\eta_L^2 + 320L^3K^2\eta\eta_L^3 < 1, \quad (4)$$

$$\eta\eta_L\left( \frac{9(M-m')^2}{2M^2}(1 + L\eta\eta_L) \right)3\tau L^2 + (\beta_{u+1} - \beta_u) \leq 0, \quad (5)$$

$$\eta\eta_L \left( \frac{9(M-m')^2}{2M^2}(1+L\eta\eta_L) \right) 3\tau L^2 - \beta_{\tau-1} \leq 0, \tag{6}$$

$$\frac{3}{2M}\sigma_L^2 \leq (\frac{1}{2} - \beta_0\eta\eta_L)\mathbb{E}\|\mathbf{G}_t\|_2^2, \tag{7}$$

*the output sequence $\{\mathbf{x}_t\}$ generated by the AFA-CS algorithm for general worker information arrival processes with bounded delay ($\tau := \max_{\mathbf{t}\in[T],i\in[M]}\{\tau_{t,i}\}$) satisfies:*

$$\frac{1}{T}\sum_{t=0}^{T-1}\mathbb{E}\|\nabla f(\mathbf{x}_t)\|_2^2 \leq \frac{4(V(x_0)-V(x_*))}{\eta\eta_L T} + 4(\alpha_L\sigma_L^2 + \alpha_G\sigma_G^2), \tag{8}$$

*where $\alpha_L$ and $\alpha_G$ are constants defined as follows:*

$$\alpha_L \triangleq [\frac{3L\eta\eta_L}{2M} + 5KL^2\eta_L^2(\frac{9(M-m')^2}{M^2}(1+L\eta\eta_L) + (\frac{3}{2}+3L\eta\eta_L))],$$

$$\alpha_G \triangleq (\frac{9(M-m')^2}{M^2}(1+L\eta\eta_L) + (\frac{3}{2}+3L\eta\eta_L))(30K^2L^2\eta_L^2),$$

*and $V(\cdot)$ is defined as $V(\mathbf{x}_t) \triangleq f(\mathbf{x}_t) + \sum_{u=0}^{\tau-1}\beta_u\|\mathbf{x}_{t-u}-\mathbf{x}_{t-u-1}\|^2$, $m'$ is the number of updates in the memory space with no time delay ($\tau_{t,i}=0$).*

*Proof.* We divide the stochastic gradient returns $\{\mathbf{G}_i\}$ into two groups, one is for those without delay ($\mathbf{G}_i(x_t), i \in \mathcal{M}_t, |\mathcal{M}_t| = m'$) and the other is for those with delay ($\mathbf{G}_i(\mathbf{x}_{t-\tau_{t,i}}), i \in \mathcal{M}_t^c, |\mathcal{M}_t^c| = M - m'$).

Then, the update step can be written as follows:

$$\mathbf{x}_{t+1} - \mathbf{x}_t = -\frac{\eta\eta_L}{M}\left[ \sum_{i\in\mathcal{M}_t} G_i(\mathbf{x}_t) + \sum_{i\in\mathcal{M}_t^c} G_i(\mathbf{x}_{t-\tau_{t,i}}) \right] \tag{10}$$

$$= -\frac{\eta}{M}\left[ \sum_{i\in[M]} G_i(\mathbf{x}_t) + \sum_{i\in\mathcal{M}_t^c}\left( G_i(\mathbf{x}_{t-\tau_{t,i}}) - G_i(x_t) \right) \right]. \tag{11}$$

Due to the $L$-smoothness assumption, taking expectation of $f(\mathbf{x}_{t+1})$ over the randomness in communication round $t$, we have:

$$\mathbb{E}[f(\mathbf{x}_{t+1})] \leq f(\mathbf{x}_t) + \underbrace{\langle\nabla f(\mathbf{x}_t), \mathbb{E}[\mathbf{x}_{t+1}-\mathbf{x}_t]\rangle}_{A_1} + \frac{L}{2}\underbrace{\mathbb{E}[\|\mathbf{x}_{t+1}-\mathbf{x}_t\|^2}_{A_2}$$

We first bound $A_2$ as follows:

$$A_2 = \mathbb{E}[\|\mathbf{x}_{t+1}-\mathbf{x}_t\|^2]$$

$$= \frac{\eta^2\eta_L^2}{M^2}\mathbb{E}\left[\left\| \sum_{i\in\mathcal{M}_t} G_i(\mathbf{x}_t) + \sum_{i\in\mathcal{M}_t^c} G_i(\mathbf{x}_{t-\tau_{t,i}}) \right\|^2\right]$$

$$= \frac{\eta^2\eta_L^2}{M^2}\mathbb{E}\left[\left\| \sum_{i\in\mathcal{M}_t}[G_i(\mathbf{x}_t) - \Delta_i(\mathbf{x}_t)]\right.\right.$$

$$\left.\left. + \sum_{i\in\mathcal{M}_t^c}[G_i(\mathbf{x}_{t-\tau_{t,i}}) - \Delta_i(\mathbf{x}_{t-\tau_{t,i}}) + \Delta_i(\mathbf{x}_{t-\tau_{t,i}}) - \Delta_i(\mathbf{x}_t)] + \sum_{i\in[M]}\Delta_i(\mathbf{x}_t) \right\|^2\right]$$

$$= \frac{3\eta^2\eta_L^2}{M^2}\left[\mathbb{E}\left\| \sum_{i\in\mathcal{M}_t}[G_i(\mathbf{x}_t) - \Delta_i(\mathbf{x}_t)] + \sum_{i\in\mathcal{M}_t^c}[G_i(\mathbf{x}_{t-\tau_{t,i}}) - \Delta_i(\mathbf{x}_{t-\tau_{t,i}})] \right\|^2\right.$$

$$\left. + \mathbb{E}\left\| \sum_{i\in\mathcal{M}_t^c}[\Delta_i(\mathbf{x}_{t-\tau_{t,i}}) - \Delta_i(\mathbf{x}_t)] \right\|^2 + \mathbb{E}\left\| \sum_{i\in[M]}\Delta_i(\mathbf{x}_t) \right\|^2\right]$$

$$\leq \frac{3\eta^2\eta_L^2}{M}\sigma_L^2 + \frac{3(M-m')}{M^2}\eta^2\eta_L^2\sum_{i\in\mathcal{M}_t^c}\|\Delta_i(\mathbf{x}_t) - \Delta_i(\mathbf{x}_{t-\tau_{t,i}})\|^2 + \frac{3\eta^2\eta_L^2}{M^2}\|\sum_{i\in[M]}\Delta_i(\mathbf{x}_t)\|^2$$

$$\leq \frac{3\eta^2\eta_L^2}{M}\sigma_L^2 + \frac{3(M-m')}{M^2}\eta^2\eta_L^2\sum_{i\in\mathcal{M}_t^c}\|\Delta_i(\mathbf{x}_t) - \Delta_i(\mathbf{x}_{t-\tau_{t,i}})\|^2$$

$$+ \frac{6\eta^2\eta_L^2}{M}\sum_{i\in[M]}\|\Delta_i(\mathbf{x}_t) - \nabla f_i(\mathbf{x}_t)\|^2 + 6\eta^2\eta_L^2\|\nabla f(\mathbf{x}_t)\|^2.$$

To bound the term $A_1$, we have:

$$A_1 = \mathbb{E}\langle \nabla f(\mathbf{x}_t), \mathbf{x}_{t+1} - \mathbf{x}_t\rangle$$

$$= \mathbb{E}\left\langle \nabla f(\mathbf{x}_t), -\frac{\eta\eta_L}{M}\left[\sum_{i\in\mathcal{M}_t}\mathbf{G}_i(\mathbf{x}_t) + \sum_{i\in\mathcal{M}_t^c}G_i(\mathbf{x}_{t-\tau_{t,i}})\right]\right\rangle$$

$$= -\eta\eta_L\mathbb{E}\left[\frac{1}{2}\|\nabla f(\mathbf{x}_t)\|^2 + \frac{1}{2M^2}\left\|\sum_{i\in\mathcal{M}_t}\mathbf{G}_i(\mathbf{x}_t) + \sum_{i\in\mathcal{M}_t^c}\mathbf{G}_i(\mathbf{x}_{t-\tau_{t,i}})\right\|^2\right.$$

$$\left. - \frac{1}{2}\left\|\nabla f(\mathbf{x}_t) - \frac{1}{M}\Big[\sum_{i\in\mathcal{M}_t}\mathbf{G}_i(\mathbf{x}_t) + \sum_{i\in\mathcal{M}_t^c}\mathbf{G}_i(\mathbf{x}_{t-\tau_{t,i}})\Big]\right\|^2\right]$$

$$\overset{(c1)}{=} -\frac{\eta\eta_L}{2}\|\nabla f(\mathbf{x}_t)\|^2 - \frac{\eta\eta_L}{2}\mathbb{E}\|\frac{1}{\eta\eta_L}(\mathbf{x}_{t+1} - \mathbf{x}_t)\|^2$$

$$+ \frac{\eta\eta_L}{2M^2}\mathbb{E}\|\sum_{i\in\mathcal{M}_t}[\nabla f_i(\mathbf{x}_t) - \mathbf{G}_i(\mathbf{x}_t)] + \sum_{i\in\mathcal{M}_t^c}[\nabla f_i(\mathbf{x}_t) - \mathbf{G}_i(\mathbf{x}_{t-\tau_{t,i}})]\|^2$$

$$= -\frac{\eta\eta_L}{2}\|\nabla f(\mathbf{x}_t)\|^2 - \frac{1}{2\eta\eta_L}\mathbb{E}\|\mathbf{x}_{t+1} - \mathbf{x}_t\|^2$$

$$+ \frac{\eta\eta_L}{2M^2}\mathbb{E}\left\|\sum_{i\in\mathcal{M}_t}[\nabla f_i(\mathbf{x}_t) - \Delta_i(\mathbf{x}_t) + \Delta_i(\mathbf{x}_t) - \mathbf{G}_i(\mathbf{x}_t)]\right.$$

$$\left. + \sum_{i\in\mathcal{M}_t^c}[\nabla f_i(\mathbf{x}_t) - \Delta_i(\mathbf{x}_t) + \Delta_i(\mathbf{x}_t) - \Delta_i(\mathbf{x}_{t-\tau_{t,i}}) + \Delta_i(\mathbf{x}_{t-\tau_{t,i}}) - \mathbf{G}_i(\mathbf{x}_{t-\tau_{t,i}})]\right\|^2$$

$$= -\frac{\eta\eta_L}{2}\|\nabla f(\mathbf{x}_t)\|^2 - \frac{1}{2\eta\eta_L}\mathbb{E}\|\mathbf{x}_{t+1} - \mathbf{x}_t\|^2$$

$$+ \frac{\eta\eta_L}{2M^2}\mathbb{E}\left\|\sum_{i\in[M]}[\nabla f_i(\mathbf{x}_t) - \Delta_i(\mathbf{x}_t)] + \sum_{i\in\mathcal{M}_t}[\Delta_i(\mathbf{x}_t) - \mathbf{G}_i(\mathbf{x}_t)]\right.$$

$$\left. + \sum_{i\in\mathcal{M}_t^c}[\Delta_i(\mathbf{x}_t) - \Delta_i(\mathbf{x}_{t-\tau_{t,i}}) + \Delta_i(\mathbf{x}_{t-\tau_{t,i}}) - \mathbf{G}_i(\mathbf{x}_{t-\tau_{t,i}})]\right\|^2$$

$$\leq -\frac{\eta\eta_L}{2}\|\nabla f(\mathbf{x}_t)\|^2 - \frac{1}{2\eta\eta_L}\mathbb{E}\|\mathbf{x}_{t+1} - \mathbf{x}_t\|^2 + \frac{3\eta\eta_L}{2M^2}\mathbb{E}\left\|\sum_{i\in\mathcal{M}_t^c}[\Delta_i(\mathbf{x}_t) - \Delta_i(\mathbf{x}_{t-\tau_{t,i}})]\right\|^2$$

$$+ \frac{3\eta\eta_L}{2M^2}\left\|\sum_{i\in[M]}\nabla f_i(\mathbf{x}_t) - \Delta_i(\mathbf{x}_t)\right\|^2$$

$$+ \frac{3\eta\eta_L}{2M^2}\mathbb{E}\left\|\sum_{i\in\mathcal{M}_t}[\Delta_i(\mathbf{x}_t) - \mathbf{G}_i(\mathbf{x}_t)] + \sum_{i\in\mathcal{M}_t^c}\Delta_i(\mathbf{x}_{t-\tau_{t,i}}) - \mathbf{G}_i(\mathbf{x}_{t-\tau_{t,i}})\right\|^2$$

$$\leq -\frac{\eta\eta_L}{2}\|\nabla f(\mathbf{x}_t)\|^2 - \frac{1}{2\eta\eta_L}\mathbb{E}\|\mathbf{x}_{t+1} - \mathbf{x}_t\|^2 + \frac{3(M-m')\eta\eta_L}{2M^2}\sum_{i\in\mathcal{M}_t^c}\mathbb{E}\left\|\Delta_i(\mathbf{x}_t) - \Delta_i(\mathbf{x}_{t-\tau_{t,i}})\right\|^2$$

$$+ \frac{3\eta\eta_L}{2M}\sum_{i\in[M]}\left\|\nabla f_i(\mathbf{x}_t) - \Delta_i(\mathbf{x}_t)\right\|^2 + \frac{3\eta\eta_L}{2M}\sigma_L^2$$

where $(c1)$ follows from the update step of the algorithm specified in Eq. 10.

Combining $A_1$ abd $A_2$, we have:

$$\mathbb{E}[f(\mathbf{x}_{t+1})] - f(\mathbf{x}_t) \le \underbrace{\left\langle \nabla f(\mathbf{x}_t), \mathbb{E}[\mathbf{x}_{t+1} - \mathbf{x}_t] \right\rangle}_{A_1} + \frac{L}{2} \underbrace{\mathbb{E}[\|\mathbf{x}_{t+1} - \mathbf{x}_t\|^2]}_{A_2}$$

$$\le f(\mathbf{x}_t) - \frac{\eta \eta_L}{2} \|\nabla f(\mathbf{x}_t)\|^2 - \frac{1}{2\eta\eta_L} \mathbb{E}\|\mathbf{x}_{t+1} - \mathbf{x}_t\|^2$$

$$+ \left( \frac{3(M-m')\eta\eta_L}{2M^2} + \frac{3L(M-m')\eta^2\eta_L^2}{2M^2} \right) \sum_{i \in \mathcal{M}_t^c} \mathbb{E} \underbrace{\left\| \Delta_i(\mathbf{x}_t) - \Delta_i(\mathbf{x}_{t-\tau_{t,i}}) \right\|^2}_{C1}$$

$$+ \left( \frac{3\eta\eta_L}{2M} + \frac{3L\eta^2\eta_L^2}{M} \right) \sum_{i \in [M]} \underbrace{\left\| \nabla f_i(\mathbf{x}_t) - \Delta_i(\mathbf{x}_t) \right\|^2}_{C2}$$

$$+ \frac{3\eta\eta_L}{2M}\sigma_L^2 + \frac{3L\eta^2\eta_L^2}{2M}\sigma_L^2 + 3L\eta^2\eta_L^2\|\nabla f(\mathbf{x}_t)\|^2 \tag{12}$$

For each worker $i$, we have:

$$C2 = \|\nabla f_i(\mathbf{x}_t) - \Delta_i(\mathbf{x}_t)\|^2$$

$$= \|\nabla f_i(\mathbf{x}_t) - \frac{1}{K_{t,i}} \sum_{j=0}^{K_{t,i}-1} \nabla f_i(\mathbf{x}_t^j)\|^2$$

$$= \frac{1}{K_{t,i}} \sum_{j=0}^{K_{t,i}-1} \|\nabla f_i(\mathbf{x}_t) - \nabla f_i(\mathbf{x}_t^j)\|^2$$

$$\le \frac{L^2}{K_{t,i}} \sum_{j=0}^{K_{t,i}-1} \|\mathbf{x}_t - \mathbf{x}_t^j\|^2$$

$$\overset{(c2)}{\le} 5KL^2\eta_L^2(\sigma_L^2 + 6K\sigma_G^2) + 30K^2L^2\eta_L^2\|\nabla f(\mathbf{x}_t)\|^2,$$

where $(c2)$ follows from the same bound of $A6$ specified in Eq. (9).

Also, note that:

$$C1 = \|\Delta_i(\mathbf{x}_t) - \Delta_i(\mathbf{x}_{t-\tau_{t,i}})\|$$

$$\le 3\|\Delta_i(\mathbf{x}_t) - \nabla f_i(\mathbf{x}_t)\|^2 + 3\|\nabla f_i(\mathbf{x}_t) - \nabla f_i(\mathbf{x}_{t-\tau_{t,i}})\|^2 + 3\|\nabla f_i(\mathbf{x}_{t-\tau_{t,i}}) - \Delta_i(\mathbf{x}_{t-\tau_{t,i}})\|^2$$

$$\le 3\|\Delta_i(\mathbf{x}_t) - \nabla f_i(\mathbf{x}_t)\|^2 + 3\|\nabla f_i(\mathbf{x}_{t-\tau_{t,i}}) - \Delta_i(\mathbf{x}_{t-\tau_{t,i}})\|^2 + 3L^2\|\mathbf{x}_t - \mathbf{x}_{t-\tau_{t,i}}\|^2$$

$$\le 3\|\Delta_i(\mathbf{x}_t) - \nabla f_i(\mathbf{x}_t)\|^2 + 3\|\nabla f_i(\mathbf{x}_{t-\tau_{t,i}}) - \Delta_i(\mathbf{x}_{t-\tau_{t,i}})\|^2 + 3L^2\|\sum_{u=0}^{\tau_{t,i}-1} \mathbf{x}_{t-u} - \mathbf{x}_{t-u-1}\|^2$$

$$\le 3\|\Delta_i(\mathbf{x}_t) - \nabla f_i(\mathbf{x}_t)\|^2 + 3\|\nabla f_i(\mathbf{x}_{t-\tau_{t,i}}) - \Delta_i(\mathbf{x}_{t-\tau_{t,i}})\|^2 + 3\tau L^2 \sum_{u=0}^{\tau-1} \|\mathbf{x}_{t-u} - \mathbf{x}_{t-u-1}\|^2,$$

where $\tau$ is the maximum delay, i.e., $\tau = \max\{\tau_{t,i}\}, \forall t \in [T], i \in [M]$.

Plugging $C1$ and $C2$ into the inequality in Eq. (12), we have:

$$\mathbb{E}[f(\mathbf{x}_{t+1})] - f(\mathbf{x}_t) \le \underbrace{\left\langle \nabla f(\mathbf{x}_t), \mathbb{E}[\mathbf{x}_{t+1} - \mathbf{x}_t] \right\rangle}_{A_1} + \frac{L}{2} \underbrace{\mathbb{E}[\|\mathbf{x}_{t+1} - \mathbf{x}_t\|^2]}_{A_2}$$

$$\le f(\mathbf{x}_t) - \frac{\eta\eta_L}{2}\|\nabla f(\mathbf{x}_t)\|^2 - \frac{1}{2\eta\eta_L}\mathbb{E}\|\mathbf{x}_{t+1} - \mathbf{x}_t\|^2$$

$$+ \left( \frac{9(M-m')\eta\eta_L}{2M^2} + \frac{9L(M-m')\eta^2\eta_L^2}{2M^2} \right) \sum_{i \in \mathcal{M}_t^c} \mathbb{E}\left\| \nabla f_i(\mathbf{x}_t) - \Delta_i(\mathbf{x}_t) \right\|^2$$

$$+ \left( \frac{9(M-m')\eta\eta_L}{2M^2} + \frac{9L(M-m')\eta^2\eta_L^2}{2M^2} \right) \sum_{i \in \mathcal{M}_t^c} \mathbb{E} \left\| \nabla f_i(\mathbf{x}_{t-\tau_{t,i}}) - \Delta_i(\mathbf{x}_{t-\tau_{t,i}}) \right\|^2$$

$$+ \left( \frac{9(M-m')^2\eta\eta_L}{2M^2} + \frac{9L(M-m')^2\eta^2\eta_L^2}{2M^2} \right) 3\tau L^2 \sum_{u=0}^{\tau-1} \mathbb{E} \left\| \mathbf{x}_{t-u} - \mathbf{x}_{t-u-1} \right\|^2$$

$$+ \left( \frac{3\eta\eta_L}{2M} + \frac{3L\eta^2\eta_L^2}{M} \right) \sum_{i \in [M]} \left\| \nabla f_i(\mathbf{x}_t) - \Delta_i(\mathbf{x}_t) \right\|^2$$

$$+ \frac{3\eta\eta_L}{2M}\sigma_L^2 + \frac{3L\eta^2\eta_L^2}{2M}\sigma_L^2 + 3L\eta^2\eta_L^2\|\nabla f(\mathbf{x}_t)\|^2$$

$$\leq f(\mathbf{x}_t) - \frac{\eta\eta_L}{2}\|\nabla f(\mathbf{x}_t)\|^2 - \frac{1}{2\eta\eta_L}\mathbb{E}\|\mathbf{x}_{t+1} - \mathbf{x}_t\|^2$$

$$+ \eta\eta_L \left( \frac{9(M-m')^2}{2M^2}(1 + L\eta\eta_L) + (\frac{3}{2} + 3L\eta\eta_L) \right) (5KL^2\eta_L^2(\sigma_L^2 + 6K\sigma_G^2) + 30K^2L^2\eta_L^2\|\nabla f(\mathbf{x}_t)\|^2)$$

$$+ \eta\eta_L \left( \frac{9(M-m')^2}{2M^2}(1 + L\eta\eta_L) \right) (5KL^2\eta_L^2(\sigma_L^2 + 6K\sigma_G^2) + 30K^2L^2\eta_L^2\frac{1}{M-m}\sum_{i \in \mathcal{M}_t^c}\|\nabla f(\mathbf{x}_{t-\tau_{t,i}})\|^2)$$

$$+ \eta\eta_L \left( \frac{9(M-m')^2}{2M^2}(1 + L\eta\eta_L) \right) 3\tau L^2 \sum_{u=0}^{\tau-1} \mathbb{E} \left\| \mathbf{x}_{t-u} - \mathbf{x}_{t-u-1} \right\|^2$$

$$+ \frac{3\eta\eta_L}{2M}\sigma_L^2 + \frac{3L\eta^2\eta_L^2}{2M}\sigma_L^2 + 3L\eta^2\eta_L^2\|\nabla f(\mathbf{x}_t)\|^2$$

$$= f(\mathbf{x}_t) - \eta\eta_L \left[ \frac{1}{2} - \left( \frac{9(M-m')^2}{2M^2}(1 + L\eta\eta_L) + (\frac{3}{2} + 3L\eta\eta_L) \right) 30K^2L^2\eta_L^2 - 3L\eta\eta_L \right] \|\nabla f(\mathbf{x}_t)\|^2$$

$$- \frac{1}{2\eta\eta_L}\mathbb{E}\|\mathbf{x}_{t+1} - \mathbf{x}_t\|^2 + \eta\eta_L \left( \frac{9(M-m')^2}{M^2}(1 + L\eta\eta_L) + (\frac{3}{2} + 3L\eta\eta_L) \right) [5KL^2\eta_L^2(\sigma_L^2 + 6K\sigma_G^2)]$$

$$+ \eta\eta_L \left( \frac{9(M-m')^2}{2M^2}(1 + L\eta\eta_L) \right) (30K^2L^2\eta_L^2\frac{1}{M-m}\sum_{i \in \mathcal{M}_t^c}\|\nabla f(\mathbf{x}_{t-\tau_{t,i}})\|^2)$$

$$+ \eta\eta_L \left( \frac{9(M-m')^2}{2M^2}(1 + L\eta\eta_L) \right) 3\tau L^2 \sum_{u=0}^{\tau-1} \mathbb{E} \left\| \mathbf{x}_{t-u} - \mathbf{x}_{t-u-1} \right\|^2 + \frac{3\eta\eta_L}{2M}\sigma_L^2 + \frac{3L\eta^2\eta_L^2}{2M}\sigma_L^2.$$

Now, define $V(\mathbf{x}_t) = f(\mathbf{x}_t) + \sum_{u=0}^{\tau-1} \beta_u \|\mathbf{x}_{t-u} - \mathbf{x}_{t-u-1}\|^2$. Based on the above bound of $\mathbb{E}[f(\mathbf{x}_{t+1})] - f(\mathbf{x}_t)$, it then follows that:

$$\mathbb{E}V(\mathbf{x}_{t+1}) - V(\mathbf{x}_t)$$

$$= \mathbb{E}f(\mathbf{x}_{t+1}) - f(\mathbf{x}_t)$$

$$+ \sum_{u=0}^{\tau-2}(\beta_{u+1} - \beta_u)\|\mathbf{x}_{t-u} - \mathbf{x}_{t-u-1}\|^2 + \beta_0\|x_{t+1} - \mathbf{x}_t\|^2 - \beta_{\tau-1}\|\mathbf{x}_{t-\tau-1} - \mathbf{x}_{t-\tau-2}\|^2$$

$$\leq -\eta\eta_L \left[ \frac{1}{2} - \left( \frac{9(M-m')^2}{2M^2}(1 + L\eta\eta_L) + (\frac{3}{2} + 3L\eta\eta_L) \right) 30K^2L^2\eta_L^2 - 3L\eta\eta_L \right] \|\nabla f(\mathbf{x}_t)\|^2$$

$$+ \eta\eta_L \left( \frac{9(M-m')^2}{M^2}(1 + L\eta\eta_L) + (\frac{3}{2} + 3L\eta\eta_L) \right) [5KL^2\eta_L^2(\sigma_L^2 + 6K\sigma_G^2)]$$

$$+ \eta\eta_L \left( \frac{9(M-m')^2}{2M^2}(1 + L\eta\eta_L) \right) (30K^2L^2\eta_L^2\frac{1}{M-m}\sum_{i \in \mathcal{M}_t^c}\|\nabla f(\mathbf{x}_{t-\tau_{t,i}})\|^2)$$

$$+ [\eta\eta_L \left( \frac{9(M-m')^2}{2M^2}(1 + L\eta\eta_L) \right) 3\tau L^2 + (\beta_{u+1} - \beta_u)] \sum_{u=0}^{\tau-2} \mathbb{E} \left\| \mathbf{x}_{t-u} - \mathbf{x}_{t-u-1} \right\|^2$$

$$+ \left[\eta\eta_L\left(\frac{9(M-m')^2}{2M^2}(1+L\eta\eta_L)\right)3\tau L^2 - \beta_{\tau-1}\right]\|\mathbf{x}_{t-\tau-1} - \mathbf{x}_{t-\tau-2}\|^2$$

$$+ (\beta_0 - \frac{1}{2\eta\eta_L})\|x_{t+1} - \mathbf{x}_t\|^2 + \frac{3\eta\eta_L}{2M}\sigma_L^2 + \frac{3L\eta^2\eta_L^2}{2M}\sigma_L^2.$$

Telescoping the above inequality from $t = 0$ to $T - 1$ of the above inequality yields:

$$\mathbb{E}V(\mathbf{x}_T) - V(\mathbf{x}_0)$$

$$\leq -\eta\eta_L\sum_{t=0}^{T-1}\left[\frac{1}{2} - \left(\frac{9(M-m')^2}{2M^2}(1+L\eta\eta_L) + (\frac{3}{2} + 3L\eta\eta_L)\right)30K^2L^2\eta_L^2 - 3L\eta\eta_L\right]\|\nabla f(\mathbf{x}_t)\|^2$$

$$+ \eta\eta_L\left(\frac{9(M-m')^2}{2M^2}(1+L\eta\eta_L)\right)(30K^2L^2\eta_L^2\sum_{t=0}^{T-1}\frac{1}{M-m}\sum_{i\in\mathcal{M}_t^c}\|\nabla f(\mathbf{x}_{t-\tau_{t,i}})\|^2)$$

$$+ \eta\eta_L T\left(\frac{9(M-m')^2}{M^2}(1+L\eta\eta_L) + (\frac{3}{2} + 3L\eta\eta_L)\right)[5KL^2\eta_L^2(\sigma_L^2 + 6K\sigma_G^2)]$$

$$+ \sum_{t=0}^{T-1}\underbrace{\left[\eta\eta_L\left(\frac{9(M-m')^2}{2M^2}(1+L\eta\eta_L)\right)3\tau L^2 + (\beta_{u+1} - \beta_u)\right]\sum_{u=0}^{\tau-2}\mathbb{E}\left\|\mathbf{x}_{t-u} - \mathbf{x}_{t-u-1}\right\|^2}_{C3}$$

$$+ \sum_{t=0}^{T-1}\underbrace{\left[\eta\eta_L\left(\frac{9(M-m')^2}{2M^2}(1+L\eta\eta_L)\right)3\tau L^2 - \beta_{\tau-1}\right]\|\mathbf{x}_{t-\tau-1} - \mathbf{x}_{t-\tau-2}\|^2}_{C4}$$

$$+ \underbrace{(\beta_0 - \frac{1}{2\eta\eta_L})\sum_{t=0}^{T-1}\|x_{t+1} - \mathbf{x}_t\|^2 + \frac{3\eta\eta_L T}{2M}\sigma_L^2 + \frac{3LT\eta^2\eta_L^2}{2M}\sigma_L^2}_{C5}$$

$$\overset{(c3)}{\leq} -\frac{1}{4}\eta\eta_L\sum_{t=0}^{T-1}\|\nabla f(\mathbf{x}_t)\|^2$$

$$+ \eta\eta_L\left[\frac{3LT\eta\eta_L}{2M} + 5TKL^2\eta_L^2\left(\frac{9(M-m')^2}{M^2}(1+L\eta\eta_L) + (\frac{3}{2} + 3L\eta\eta_L)\right)\right]\sigma_L^2$$

$$+ \eta\eta_L T\left(\frac{9(M-m')^2}{M^2}(1+L\eta\eta_L) + (\frac{3}{2} + 3L\eta\eta_L)\right)(30K^2L^2\eta_L^2)\sigma_G^2,$$

where $(c3)$ holds if the following conditions are satisfied:

$$\frac{1}{4} \leq \left[\frac{1}{2} - \left(\frac{9(M-m')^2}{2M^2}(1+L\eta\eta_L) + (\frac{3}{2} + 3L\eta\eta_L)\right)30K^2L^2\eta_L^2 - 3L\eta\eta_L\right]$$

$$- \left(\frac{9(M-m')^2}{2M^2}(1+L\eta\eta_L)\right)(30\tau K^2L^2\eta_L^2),$$

$$C3 = \left[\eta\eta_L\left(\frac{9(M-m')^2}{2M^2}(1+L\eta\eta_L)\right)3\tau L^2 + (\beta_{u+1} - \beta_u)\right] \leq 0,$$

$$C4 = \left[\eta\eta_L\left(\frac{9(M-m')^2}{2M^2}(1+L\eta\eta_L)\right)3\tau L^2 - \beta_{\tau-1}\right] \leq 0,$$

$$C5 \leq 0 \leftarrow \frac{3}{2M}\sigma_L^2 \leq (\frac{1}{2} - \beta_0\eta\eta_L)\mathbb{E}\|\mathbf{G}_t\|^2.$$

By rearranging the inequality, we have:

$$\frac{1}{4}\eta\eta_L\sum_{t=0}^{T-1}\|\nabla f(\mathbf{x}_t)\|^2$$

$$
\begin{aligned}
\leq & V(x_0) - \mathbb{E}V(x_T) \\
& + \eta\eta_L T\left[\frac{3L\eta\eta_L}{2M} + 5KL^2\eta_L^2\left(\frac{9(M-m')^2}{M^2}(1+L\eta\eta_L) + (\frac{3}{2}+3L\eta\eta_L)\right)\right]\sigma_L^2 \\
& + \eta\eta_L T\left(\frac{9(M-m')^2}{M^2}(1+L\eta\eta_L) + (\frac{3}{2}+3L\eta\eta_L)\right)(30K^2L^2\eta_L^2)\sigma_G^2,
\end{aligned}
$$

That is,

$$
\frac{1}{T}\sum_{t=0}^{T-1}\mathbb{E}\|\nabla f(\mathbf{x}_t)\|^2 \leq \frac{4(V(x_0)-V(x_*))}{\eta\eta_L T} + 4[\alpha_L\sigma_L^2 + \alpha_G\sigma_G^2],
$$

where $\alpha_L = \left[\frac{3L\eta\eta_L}{2M} + 5KL^2\eta_L^2\left(\frac{9(M-m')^2}{M^2}(1+L\eta\eta_L) + (\frac{3}{2}+3L\eta\eta_L)\right)\right]$, $\alpha_G = \left(\frac{9(M-m')^2}{M^2}(1+L\eta\eta_L) + (\frac{3}{2}+3L\eta\eta_L)\right)(30K^2L^2\eta_L^2)$. This completes the proof. $\qquad\square$

**Corollary 3** (Linear Speedup). *By setting $\eta_L = \frac{1}{\sqrt{T}}$, and $\eta = \sqrt{M}$, the convergence rate of the AFA-CS algorithm for general worker information arrival processes with bounded delay is:*

$$
\frac{1}{T}\sum_{t=0}^{T-1}\mathbb{E}\|\nabla f(\mathbf{x}_t)\|_2^2 = \mathcal{O}\left(\frac{1}{M^{1/2}T^{1/2}}\right) + \mathcal{O}\left(\frac{K^2}{T}\right) + \mathcal{O}\left(\frac{K^2M^{1/2}}{T^{3/2}}\right).
$$

*Proof.* Let $\eta_L = \frac{1}{\sqrt{T}}$, and $\eta = \sqrt{M}$. It then follows that:

$$
\alpha_L = \mathcal{O}(\frac{1}{M^{1/2}T^{1/2}}) + \mathcal{O}(\frac{K}{T}) + \mathcal{O}(\frac{KM^{1/2}}{T^{3/2}}).
$$

$$
\alpha_G = \mathcal{O}(\frac{K^2}{T}) + \mathcal{O}(\frac{K^2M^{1/2}}{T^{3/2}}).
$$

This completes the proof. $\qquad\square$

## D    DISCUSSION

**Convergence Error:** The case with uniformly distributed worker information arrivals under AFL can be viewed as a uniformly independent sampling process from total workers $[M]$ under conventional FL. Also, the case with general worker information arrival processes under AFL can be equivalently mapped to an arbitrarily independent sampling under conventional FL. In each communication round, the surrogate objection function for partial worker participation in FL is $\tilde{f}(x) := \frac{1}{|\mathcal{M}_t|}\sum_{i\in\mathcal{M}_t}f_i(x)$. For uniformly independent sampling, the surrogate object function approximately equals to $f(x) := \frac{1}{M}\sum_{i=1}^M f_i(x)$ in expectation, i.e., $\mathbb{E}[\tilde{f}(x)] = f(x)$. However, the surrogate object function $\tilde{f}(x)$ may deviate from $f(x)$ with arbitrarily independent sampling. More specifically, for uniformly independent sampling, the bound of $\|\nabla f(\mathbf{x}_t) - \tilde{f}(\mathbf{x}_t)\|^2$ is independent of $\sigma_G$ ($A_3$ term in B.3). On the other hand, for arbitrarily independent sampling, $\|\nabla f(\mathbf{x}_t) - \tilde{f}(\mathbf{x}_t)\|^2 \leq \mathcal{O}(\sigma_G^2)$ ($A_3$ term in B.2). This deviation may happen in every communication round, so it is non-vanishing even with infinity communication rounds. As a result, such deviation is originated from the arbitrary sampling coupling with non-i.i.d. datasets. In other words, it is irrelevant to the optimization hyper-parameters such as the learning rate, local steps and others, which is different from the objective inconsistency due to different local steps shown in Wang et al. (2020). When we set $\tau = 0$ and $K_{t,i} = K, \forall t, i$, AFA-CD generalizes FedAvg. In such sense, the convergence error also exists in currently synchronous FL algorithms with such arbitrarily independent sampling and non-i.i.d. dataset. Moreover, this sampling process coupling with non-i.i.d. dataset not only results in convergence issue but also potentially induces a new source of bias/unfairness (Mohri et al., 2019; Li et al., 2019b). So how to model the practical worker participation process in practice and in turn tackle these potential bias are worth further exploration.

**Variance Reduction:** If we view the derivation between local loss function and global loss function as global variance, i.e., $\|\nabla f_i(\mathbf{x}_t) - \nabla f(\mathbf{x}_t)\|^2 \leq \sigma_G^2, \forall i \in [m], \forall t$ as shown in Assumption 3, the AFA-CS algorithm is indeed a variance reduction (VR) method, akin to SAG (Le Roux et al., 2012; Schmidt et al., 2017). SAG maintains an estimate stochastic gradient $v_i, i \in [n]$ for each data point ($n$ is the size of the dataset). In each iteration, SAG only samples one data point (say, $j$) and update the stochastic gradient on latest model ($v_j = \nabla f_j(x_t)$) stored in the memory space, but then use the average of all stored stochastic gradients as the estimate of a full gradient to update the model ($x_{t+1} = x_t - \eta_t g_t, g_t = \frac{1}{n} \sum_{i=1}^{n} v_i$). In such way, SAG is able to have a faster convergence rate by reducing the local variance due to the stochastic gradient. AFA-CS algorithm performs in the similar way. The server in the AFA-CS algorithm maintains a parameter for each worker as an estimate of the returned stochastic gradient. In each communication round, the server only receives $m$ updates in the memory space but updates the global model by the average of all the $M$ parameters. As a result, not only can it diminish the convergence error derived from the non-i.i.d. dataset and general worker information arrival processes (arbitrarily independent sampling), but also accelerate the convergence rate with a linear speedup factor $M$. Previous works have applied VR methods in FL, notably SCAFFOLD (Karimireddy et al., 2020b) and FedSVRG (Konecnỳ et al., 2016). The key difference is that we apply the VR on the server side to control the global variance while previous works focus on the worker side in order to tackle the model drift due to local update steps. Applying VR methods on server and worker side are orthogonal, and thus can be used simultaneously. We believe other variance reduction methods could be similarly extended on the server side in a similar fashion as what we do in AFA-CD. This will be left for future research.

# E   EXPERIMENTS

In this section, we provide the detailed experiment settings as well as extra experimental results that cannot fit in the page limit of the main paper.

## E.1   MODEL AND DATASETS

We run three models on three different datasets, including i) multinomial logistic regression (LR) on manually partitioned non-i.i.d. MNIST, ii) convolutional neural network (CNN) for manually partitioned non-i.i.d. CIFAR-10, and iii) recurrent neural network (RNN) on natural non-i.i.d. Shakespeare datasets. These dataset are curated from previous FL papers (McMahan et al., 2016; Li et al., 2018) and are now widely used as benchmarks in FL studies (Li et al., 2019c; Yang et al., 2021).

For MNIST and CIFAR-10, each dataset has ten classes of images. To impose statistical heterogeneity, we split the data based on the classes ($p$) of images each worker contains. We distribute the data to $M = 10$(or 100) workers such that each worker contains only certain classes with the same number of training/test samples. Specifically, each worker randomly chooses $p$ classes of labels and evenly samples training/testing data points only with these $p$ classes labels from the overall dataset without replacement. For example, for $p = 2$, each worker only has training/testing samples with two classes, which causes heterogeneity among different workers. For $p = 10$, each worker has samples with ten classes, which is nearly i.i.d. case. In this way, we can use the classes ($p$) in worker's local dataset to represent the non-i.i.d. degree qualitatively.

The Shakespeare dataset is built from *The Complete Works of William Shakespeare* (McMahan et al., 2016). We use a two-layer LSTM classifier containing 100 hidden units with an embedding layer. The learning task is the next-character prediction, and there are 80 classes of characters in total. The model takes as input a sequence of 80 characters, embeds each of the characters into a learned 8-dimensional space and outputs one character per training sample after two LSTM layers and a densely-connected layer. The dataset and model are taken from LEAF (Li et al., 2018).

For MNIST and CIFAR-10, we use global learning rate $\eta = 1.0$ and local learning rate $\eta_L = 0.1$. For MNIST, the batch size is 64 and the total communication round is 150. For CIFAR-10, the batch size is 500 and the total communication round is 10000. For the Shakespeare dataset, the global learning rate is $\eta = 50$, the local learning rate is $\eta_L = 0.8$, batch size is $b = 10$, and the total communication round is 300. In the following tables and figure captions, we use "$m/M$" to denote that, in each communication round, we randomly choose $m$ workers from $[M]$ to participate in the training.

We study the asynchrony and heterogeneity factors in AFL, including asynchrony, heterogeneous computing, worker's arrival process, and data heterogeneity. To simulate the asynchrony, each participated worker choose one global model from the last recent five models instead of only using the latest global model for synchronous case. To mimic the heterogeneous computing, we simulate two cases: constant and dynamic local steps. For constant local steps, each participated worker performs a fixed $c$ local update steps. In contrast, each worker takes a random local update steps uniformly sampled from $[1, 2 \times c]$ for dynamic local steps. To emulate the effect of various worker's arrival processes, we use uniform sampling without replacement to simulate the uniformly distributed worker information arrivals, and we use biased sampling with probability $[0.19, 0.19, 0.1, 0.1, 0.1, 0.1, 0.1, 0.1, 0.01, 0.01]$ without replacement for total 10 workers to investigate potential biases with general worker information arrival processes. To study the data heterogeneity, we use the value $p$ as a proxy to represent the non-i.i.d. degree for MNIST and CIFAR-10.

Table 1: CNN Architecture for CIFAR-10.

| Layer Type | Size |
|---|---|
| Convolution + ReLu | $5 \times 5 \times 32$ |
| Max Pooling | $2 \times 2$ |
| Convolution + ReLu | $5 \times 5 \times 64$ |
| Max Pooling | $2 \times 2$ |
| Fully Connected + ReLU | $1024 \times 512$ |
| Fully Connected + ReLU | $512 \times 128$ |
| Fully Connected | $128 \times 10$ |

### E.2 FURTHER EXPERIMENTAL RESULTS

Table 2: Test Accuracy for comparison of asynchrony and local steps.

| Models/ Dataset | Non-i.i.d. index (p) | Worker number | Local steps | Synchrony | | Asynchrony | |
|---|---|---|---|---|---|---|---|
| | | | | Constant steps | Dynamic steps | Constant Steps | Dynamic Steps |
| LR/ MNIST | $p = 1$ | 5/10 | 5 | 0.8916 | 0.8915 | 0.8888 | 0.8868 |
| | $p = 2$ | 5/10 | 5 | 0.8906 | 0.8981 | 0.8901 | 0.8931 |
| | $p = 5$ | 5/10 | 5 | 0.9072 | 0.9075 | 0.9059 | 0.9048 |
| | $p = 10$ | 5/10 | 5 | 0.9114 | 0.9111 | 0.9129 | 0.9143 |
| | $p = 1$ | 5/10 | 10 | 0.8743 | 0.8786 | 0.8701 | 0.8734 |
| | $p = 2$ | 5/10 | 10 | 0.8687 | 0.8813 | 0.8661 | 0.8819 |
| | $p = 5$ | 5/10 | 10 | 0.9016 | 0.9050 | 0.9034 | 0.9065 |
| | $p = 10$ | 5/10 | 10 | 0.9124 | 0.9135 | 0.9112 | 0.9111 |
| | $p = 1$ | 20/100 | 5 | 0.8898 | 0.8973 | 0.8909 | 0.8938 |
| | $p = 2$ | 20/100 | 5 | 0.8968 | 0.9007 | 0.8955 | 0.9000 |
| | $p = 5$ | 20/100 | 5 | 0.9088 | 0.9088 | 0.9097 | 0.9078 |
| | $p = 10$ | 20/100 | 5 | 0.9111 | 0.9106 | 0.9126 | 0.9125 |
| CNN/ CIFAR-10 | $p = 1$ | 5/10 | 5 | 0.7474 | 0.7606 | 0.7319 | 0.7350 |
| | $p = 2$ | 5/10 | 5 | 0.7677 | 0.7944 | 0.7662 | 0.777 |
| | $p = 5$ | 5/10 | 5 | 0.7981 | 0.802 | 0.8065 | 0.799 |
| | $p = 10$ | 5/10 | 5 | 0.8081 | 0.8072 | 0.8065 | 0.8119 |
| RNN/ Shakespeare | - | 72/143 | 50 | 0.4683 | 0.4831 | 0.4606 | 0.4687 |

**Effect of asynchrony, local update steps, and non-i.i.d. level.** In table 2, we examine three factors by comparing the top-1 test accuracy: synchrony versus asynchrony, constant steps versus dynamic steps and different levels of non-i.i.d. dataset. The worker sampling process is uniformly random sampling to simulate the uniformly distributed worker information arrivals. The baseline is synchrony with constant steps. When using asynchrony or/and dynamic local steps, the top-1 test accuracy shows

no obvious differences. This observation can be observed in all these three tasks. Asynchrony and dynamic local update steps enable each worker to participate flexibly and loosen the coupling between workers and the server. As a result, asynchrony and dynamic local steps introduce extra heterogeneity factors, but the performance of the model is as good as that of the synchronous approaches with constant local steps. Instead, the data heterogeneity is an important factor for the model performance. As the non-i.i.d. level increases (smaller $p$ value), the top-1 test accuracy decreases.

Next, we study convergence speed of the test accuracy for the model training under different settings. Figure 2 illustrates the test accuracy for LR on MNIST with different non-i.i.d. levels. We can see that asynchrony and dynamic local steps result in zigzagging convergence curves, but the final accuracy results have negligible differences. The zigzagging phenomenon is more dramatic as the non-i.i.d. level gets higher. Interestingly, from Figure 3 and Figure 4, we can see that for less non-i.i.d. settings such as $p = 10$ and $p = 5$, the curves of all algorithms are almost identical. Specifically, in Figure 4, the test accuracy curves of the LSTM model oscillates under asynchrony and dynamic local steps. Another observation is that it takes more rounds to converge as the non-i.i.d. level of the datasets increases. This trend can be clearly observed in Figure 3.

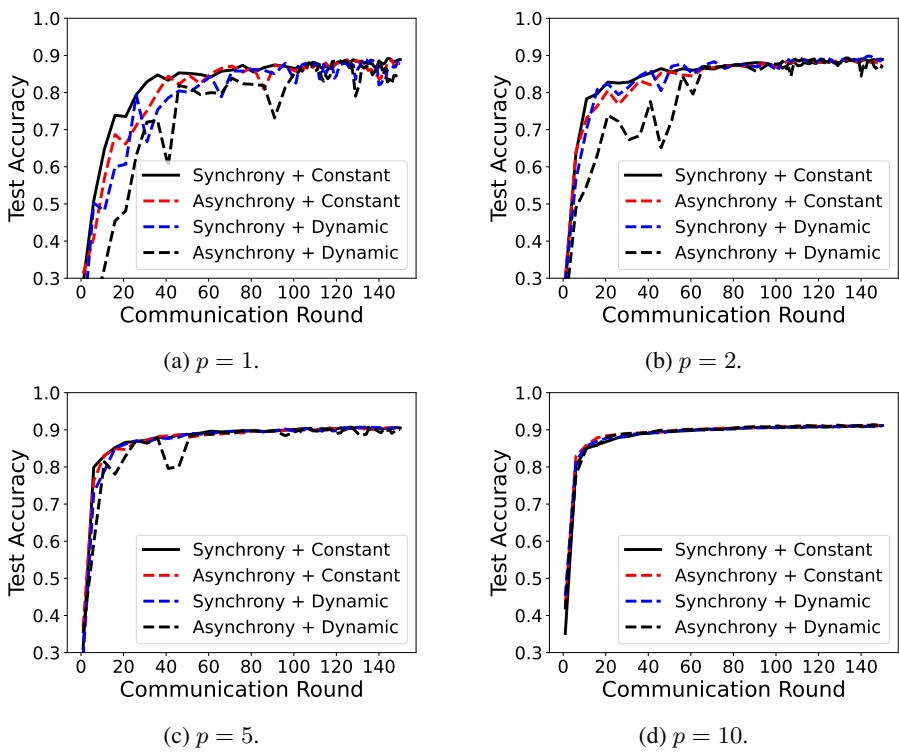

Figure 2: Test accuracy for LR on MNIST with worker number $5/10$, local steps $5$.

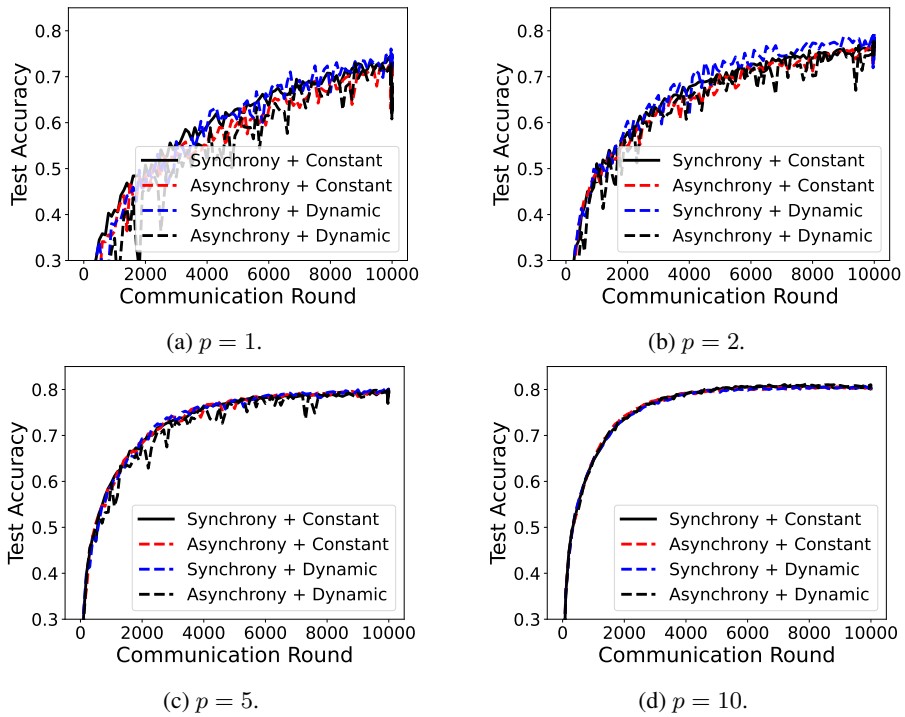

(a) $p = 1$.

(b) $p = 2$.

(c) $p = 5$.

(d) $p = 10$.

Figure 3: Test accuracy for CNN on CIFAR-10 with worker number $5/10$, local steps $5$.

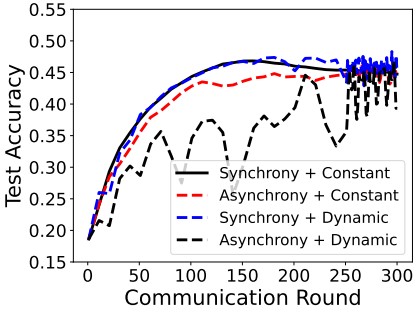

Figure 4: Test accuracy for LSTM on Shakespeare with worker number $72/143$, local steps $50$.

Table 3: Test Accuracy of FedProx and SCAFFOLD.

| Models/ Dataset | Non-i.i.d. index (p) | Worker number | Local steps | FedProx | SCAFFOLD | AFL + FedProx | AFL + SCAFFOLD |
|---|---|---|---|---|---|---|---|
| LR/ MNIST | $p = 1$ | 5/10 | 5 | 0.8893 | 0.8928 | 0.8775 | 0.8946 |
| | $p = 2$ | 5/10 | 5 | 0.8868 | 0.8970 | 0.8832 | 0.8954 |
| | $p = 5$ | 5/10 | 5 | 0.9036 | 0.9032 | 0.9004 | 0.9019 |
| | $p = 10$ | 5/10 | 5 | 0.9075 | 0.9057 | 0.9054 | 0.9022 |
| | $p = 1$ | 5/10 | 10 | 0.8752 | 0.8789 | 0.8669 | 0.8838 |
| | $p = 2$ | 5/10 | 10 | 0.8685 | 0.8967 | 0.8789 | 0.8978 |
| | $p = 5$ | 5/10 | 10 | 0.9019 | 0.9047 | 0.8998 | 0.9029 |
| | $p = 10$ | 5/10 | 10 | 0.9072 | 0.9071 | 0.9052 | 0.9038 |
| CNN/ CIFAR-10 | $p = 1$ | 5/10 | 5 | 0.7488 | 0.1641 | 0.7415 | 0.3935 |
| | $p = 2$ | 5/10 | 5 | 0.7728 | 0.6315 | 0.7890 | 0.6971 |
| | $p = 5$ | 5/10 | 5 | 0.7931 | 0.7828 | 0.8031 | 0.7884 |
| | $p = 10$ | 5/10 | 5 | 0.8150 | 0.8083 | 0.8143 | 0.8051 |
| RNN/ Shakespeare | - | 72/143 | 50 | 0.4690 | 0.4794 | 0.4550 | 0.4515 |

**Utilizing FedProx and SCAFFOLD as the optimizer on the worker-side.** Here, we choose FedProx and SCAFFOLD as two classes of algorithms in existing FL algorithms. FedProx represents these algorithms that modifies the local objective function. Other algorithms belonging to this category includes FedPD (Zhang et al., 2020b) and FedDyn (Acar et al., 2021). In such algorithms, no extra information exchange between worker and server is needed. On the other hand, SCAFFOLD represents VR-based (variance reduction) algorithms. It needs an extra control variate to perform the "variance reduction" step, so extra parameters are required in each communication round. Other algorithms in this class includes FedSVRG (Konecnỳ et al., 2016).

In Table 3, we show the effectiveness of utilizing existing FL algorithms, FedProx and SCAFFOLD, in the AFL framework. For FedProx and SCAFFOLD, we examine synchrony and constant local steps settings. When incorporating these two advanced FL algorithms in the AFL framework, we study the effects of asynchrony and dynamic local steps. We set $\mu = 0.1$ as default in FedProx algorithm. We can see from Table 3 that FedProx performs as good as FedAvg does (compare with the results in Table 2). Also, there is no performance degradation in AFL framework by utilizing FedProx as the worker's optimizer. However, while SCAFFOLD performs well for LR on MNIST, it dose not work well for CNN on CIFAR-10, especially in cases with higher non-i.i.d. levels. One possible reason is that the control variates can become stale in partial worker participation and in turn degrade the performance. Previous work also showed similar results (Acar et al., 2021; Reddi et al., 2020). If we view the SCAFFOLD ( in synchrony and constant steps setting) as the baseline, no obvious performance degradation happens under AFL with SCAFFOLD being used as the worker's optimizer.

**Effects of different worker information arrival processes.** In order to generate different workers' arrival processes, we use uniform sampling without replacement to simulate the uniformly distributed worker information arrivals and use biased sampling to simulate the potential bias in general worker information arrival processes. In Figures 5 and 6, we illustrate the effect of the sampling process for LR on MNIST and CNN on CIFAR-10 with asynchrony and dynamic local steps. For highly non-i.i.d. datasets ($p = 1$), the biased sampling process degrades the model performance. This is consistent with the larger convergence error as shown in our theoretical analysis. On the other hand, for other non-i.i.d. cases with $p = 2, 5, 10$, such biased sampling dose not lead to significant performance degradation. When applying variance reduction on such biased sampling process by reusing old gradients as shown in AFA-CS, we can see that AFA-CS performs well on MNIST, but not on CIFAR-10. We conjecture that AFA-CS, as a variance reduction method, does not always perform well in practice. This observation is consistent with the previous work (Defazio & Bottou, 2018; Reddi et al., 2020), which also demonstrated the ineffectiveness of variance reduction methods in deep learning and some cases of FL.

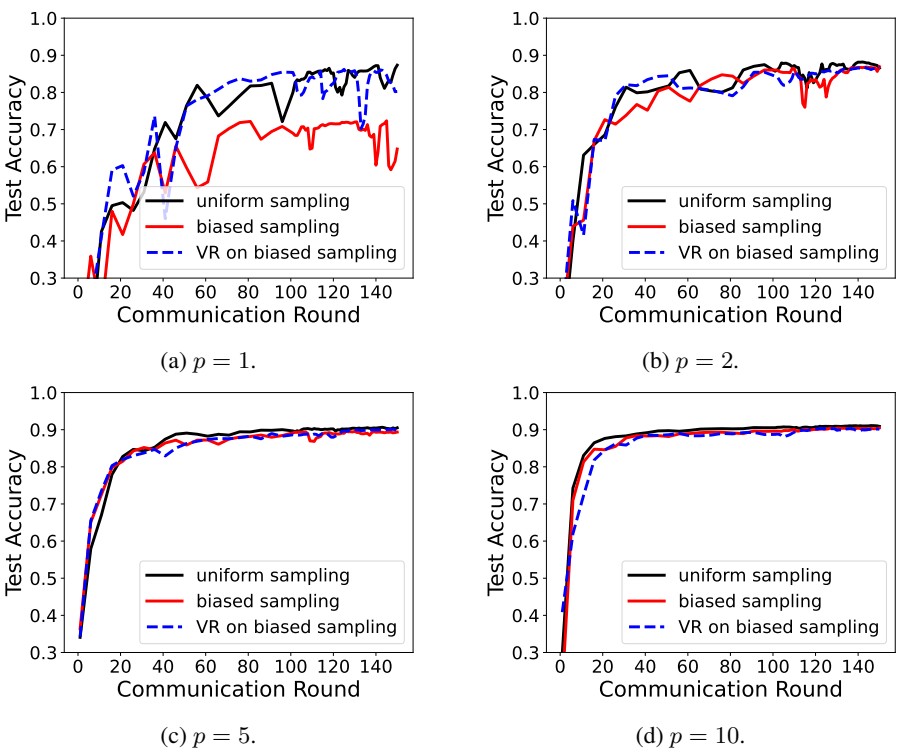

(a) $p = 1$.

(b) $p = 2$.

(c) $p = 5$.

(d) $p = 10$.

Figure 5: Test accuracy for LR on MNIST with asynchrony and dynamic local steps.

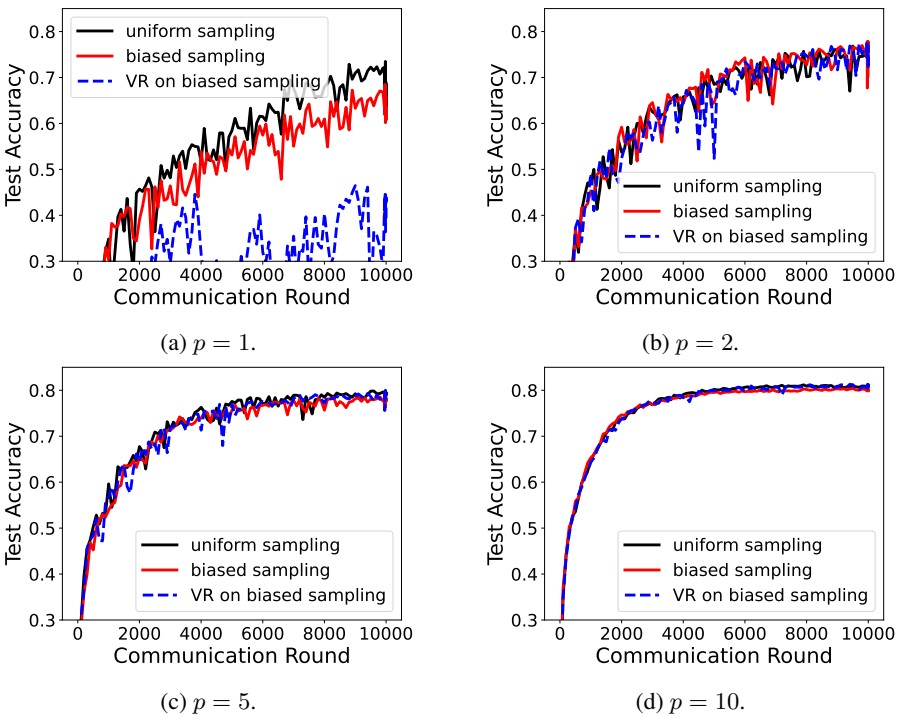

(a) $p = 1$.

(b) $p = 2$.

(c) $p = 5$.

(d) $p = 10$.

Figure 6: Test accuracy for CNN on CIFAR-10 with asynchrony and dynamic local steps.

