# OpenReview forum: "Anarchic Federated Learning"
_ICLR.cc/2022/Conference — ICLR 2022 Submitted_

### Official Review · Reviewer_rgtg · 2021-10-24

**Correctness:** 4
**Technical Novelty And Significance:** 3
**Empirical Novelty And Significance:** 3
**Recommendation:** 6
**Confidence:** 3

**Main Review:**

The overall idea of AFL is interesting and the theoretical results of AFA-CD and AFA-CS look reasonable. However, I found the following issues regarding AFL concerning. I hope the authors can address them in the rebuttal.

- The convergence rate of AFA-CD can be bad under the general worker information arrival process (truly anarchic). Take the equation in Corollary 1 for example, by the definition of $\tau$ and $K$, they can be as large as $O(T)$ if a few workers set their delay and local steps to be so. The algorithm 2 and 3 provides no guarantee that such delayed gradient will be rejected. That is, if a client starts local steps from $t=0$ and decide to submit its update at $T$, which is the *complete freedom* stated in the paper, algorithm 2 and 3 will incorpoate this update and ruin the training.

- The assumption of uniform distributed arrival of worker information is not realistic. The participants of cross-device federated learning have various computational power and communication frequency, etc.
So this assumption is most likely not true. It means in reality, AFL cannot achieve the same performance as FedAvg as FedAvg can enforce uniform sampling of clients.

- The assumption of bounded maximum delay and local steps is contrary to the "anarchic". In addition, the worker learning rate $\eta_L$ depends on the maximum delay. It can not be decided under the truly anarchic case.

- AFL is much more vulnerable to attacks than FedAvg. As the worker has the freedom to choose when to join training, the chance of attacker updates being incorporated in the aggregatrion is drastically increased by simply being eager to communicate to the server. This makes AFL vulnerable to backdoor/Byzantine attacks even if there is only a small percent of attackers. On the other hand, the FedAvg uniformly sample the clients and have a low chance of selecting many attackers.

### Typos
- Page 1, "stragglers (i.e. show workers)" => "slow workers"
- Page 2, "Clearly, AFL has a much lower sever-worker coordination" => "server-worker"
- Page 4, "Although the sever in cross-silo" => "server"

======

Post-rebuttal: I have read the authors' rebuttal and found them addressed my concerns so I decide to raise the score to 6.



**Summary Of The Paper:**

In order to address the issues of FedAvg (e.g. straggler, waste computation, slow convergence), this paper proposes a new federated training scheme "anarchic federated learning" (AFL) as an alternative. Instead of uniformly sampling participant clients, AFL let all workers to decide their number of local steps, when to communicate, and their step-sizes / batch sizes. The authors established a theoretical convergence rate of AFL and show that it recovers the rate of FedAvg under the assumption of uniform distributed arrival of worker information and bounded maximum delay and local steps. The authors also provides empirical evaluations on some experiments to demonstrate that


**Summary Of The Review:**

The anarchic federated learning is vulnerable to attacks and can be slower than FedAvg because of unbounded delay, local steps, and heterogeneity of clients.

---

> ### Author Response · Authors · 2021-11-23
> **Response to Reviewer rgtg (1/2)**
>
> Thank you very much for the review and the constructive comments. We believe the valuable suggestions from the reviewer have helped us significantly improve the quality of this paper. We believe that there is some misunderstanding regarding the results presented in this paper. In this revision, we have clarified the concerns raised by the reviewer and have carefully revised our paper based on reviewer's comments, questions, and suggestions. Please see the revised manuscript at the top of this page. The detailed point-by-point responses are as follows:
> > **Your Comment:** 1. The convergence rate of AFA-CD can be bad under the general worker information arrival process (truly anarchic). Take the equation in Corollary 1 for example, by the definition of $\tau$ and $K$, they can be as large as $\mathcal{O}(T)$ if a few workers set their delay and local steps to be so. The algorithm 2 and 3 provides no guarantee that such delayed gradient will be rejected. That is, if a client starts local steps from $t=0$ and decide to submit its update at $T$, which is the complete freedom stated in the paper, algorithm 2 and 3 will incorporate this update and ruin the training.
>
> **Our Response:** Thanks for your constructive comment. First, we would like to clarify that, in this paper, we only consider bounded maximum delays. We do agree that, for cross-device AFL, the delays may be unbounded in practice. However, the unbounded delay setting may be viewed as an extension of the setting where the server abandons/reject clients with large return delay (e.g., by setting a maximum time delay threshold). Under this setting, the maximum delay in the "reduced" system can be viewed as bounded. We have added the discussions in this revision. In this paper, our focus is to explore the effect of $\tau$, which could be large but remain bounded (i.e., $\tau$ does not grow with respect to $T$, hence $\tau \ne \mathcal{O}(T)$). But we thank the reviewer for pointing out this interesting direction, which will be left as our next step in our future studies.
>
> > **Your Comment:** 2. The assumption of uniform distributed arrival of worker information is not realistic. The participants of cross-device federated learning have various computational power and communication frequency, etc. So this assumption is most likely not true. It means in reality, AFL cannot achieve the same performance as FedAvg as FedAvg can enforce uniform sampling of clients.
>
> **Our Response:** We fully agree with you that uniform distributed arrival of worker information is not realistic in practice. In fact, this assumption is also not true for FedAvg even if FedAvg samples clients uniformly at random, since other uncontrollable system factors could dramatically affect workers' participation in real-world FL. It has been shown that more than 30% of devices never participate in the learning process and top 30% of devices can contribute 81% of the total computation (see [this paper](https://arxiv.org/pdf/2006.06983.pdf)),
> due to unstable device availability and large failure proportion in real-world FL systems such as [Google FL deployment](https://arxiv.org/pdf/1902.01046.pdf) and  [practical FL simulation platform](https://arxiv.org/pdf/2006.06983.pdf).
>
> The reasons we considered uniform distributed worker information arrivals are two-fold: i) we would like to study how well our proposed AFL algorithms could perform under this idealized assumption, which is often made in the literature; and ii) due to the fact that this idealized assumption is often made in the literature, studying our algorithms under this assumptions allows a direct performance comparisons with many conventional FL algorithms.

---

> > ### Author Response · Authors · 2021-11-23
> > **Response to Reviewer rgtg (2/2)**
> >
> > > **Your Comment:** 3. The assumption of bounded maximum delay and local steps is contrary to the "anarchic". In addition, the worker learning rate depends on the maximum delay. It can not be decided under the truly anarchic case.
> >
> > **Our Response:** Our meaning of the term "anarchic" is that the client could participate in the training at will and at anytime without system locking as in FL. However, we never meant it is possible to design a convergent learning algorithm for AFL systems with arbitrary worker behaviors (e.g., unbounded delay and arbitrary participation). In fact, revealing the sufficient conditions on "anarchic" behaviors (e.g., bounded maximum delay and local steps) is one of the key insights and contributions of this paper. Also, we have established the worst-case lower bound for such arbitrary behaviors. Based on your comment, we have added further explanations in this revision to clarify what we mean by "anarchic."
> >
> > Similarly, for the learning rate in anarchic settings, it should still be sufficiently small to satisfy certain conditions (e.g., dependence on smoothness parameter $L$, delay $\tau$ and local steps $K$) so that there is hope to design convergence training algorithms for AFL. We note that most of these conditions are similar to requirements in asynchronous distributed optimization (e.g., [the classic paper](http://proceedings.mlr.press/v80/lian18a/lian18a.pdf) or a [recent one](https://arxiv.org/abs/2106.06639)) and not too restrictive. In addition, we note that our learning rate settings are independent of delay as shown in our Corollary 1, 2, and 3. Again, revealing the sufficient "anarchic" conditions on learning rates is one of the key insights and contributions of this paper.
> >
> > > **Your Comment:** 4. AFL is much more vulnerable to attacks than FedAvg. As the worker has the freedom to choose when to join training, the chance of attacker updates being incorporated in the aggregation is drastically increased by simply being eager to communicate to the server. This makes AFL vulnerable to backdoor/Byzantine attacks even if there is only a small percent of attackers. On the other hand, the FedAvg uniformly sample the clients and have a low chance of selecting many attackers.
> >
> > **Our Response:** We fully agree with the reviewer that the AFL paradigm is more vulnerable to attacks. However, the security studies of AFL is an important topic beyond the scope of this paper and deserves an independent paper. This will be our next step in future studies. Here in this paper, our focus is to provide convergence analysis of AFL (and FL), assuming all clients are benign. But we thank the reviewer for suggesting this important topic.
> >
> > > **Your Comment:** 5. Typos
> >
> > **Our Response:** Thanks for catching the typos. We have corrected them in this revision.

---

### Official Review · Reviewer_8n7V · 2021-10-29

**Correctness:** 3
**Technical Novelty And Significance:** 2
**Empirical Novelty And Significance:** 2
**Recommendation:** 5
**Confidence:** 4

**Main Review:**

Advantages:

* It provides convergence guarantees for the proposed asynchronous algorithms in federated settings.

* It also characterizes the lower bound without any assumptions on the client communication pattern, which provides useful reference points on the improvements of the analysis with additional assumptions.

Concerns:

* The proposed algorithms are very similar to those proposed in this paper (https://arxiv.org/abs/2106.06639, Federated Learning with Buffered Asynchronous Aggregation).  Although the paper I mentioned lets the server sample clients asynchronously, algorithmically looks like these two works are doing the same thing. I understand the submission is more general and improves the analysis in certain ways (e.g., without bounded gradient), but the main algorithms and some motivation are largely overlapping. It would be very helpful to clarify the major differences, if there are any.

* I am confused by the difference and the motivation for two variants designed for cross-device and cross-silo settings. (1) I think the server in both settings is assumed to be powerful in terms of memory and computation. Can the work give some references on why the server is assumed to have no historical information of the works? (2) The submission assumes uniform arrival for cross-device FL, and bounded delay for cross-silo FL. They both control the staleness of the model updates (to make sure the global model some device starts from is not too stale compared with the current global model for aggregation). Both of them are reasonable for cross-silo FL; and hence, to me, there isn’t a clear mapping between which setting is more compatible with which assumption.

* I think one motivation for asynchronous training is that it leads to faster convergence (empirically) in the presence of stragglers. This point should be validated empirically (by simulating systems heterogeneity, for example).

* In practice, the delays may be unbounded (e.g., when each device only participates once or twice during the training); so the two assumptions can be too strong.

Minor:

* Why does the lower bound also hold for FedAvg (described in Remark 1)? Under proper step-sizes, there is usually no constant error term.


* The last paragraph in Section 3 (1) spends much text highlighting that AFL is a general computing architecture. But this (using any local solver with different hyperparameters, allowing for device-specific local steps) is not a unique property of AFL, and lots of previous federated optimization methods also support this.

* At the end of the first paragraph of related work (Sec 2), I don't think calling server-centric synchronous algorithms are not easy to implement is accurate or justified.

* The phrase 'general worker information arrival process' appears for the first time in the abstract, which is a bit abrupt and unclear---what is a worker information arrival process? general in what sense?

* The empirical evaluation does not describe how the stragglers are simulated precisely. For example, the starting points of the clients uniformly sampled (or sampled in a biased way) correspond to which time stamp? Does the 'bias sampling' scheme (described in the appendix) correspond to the bounded delay case?

**Summary Of The Paper:**

This work proposes a different worker-server asynchronous communication pattern in federated learning, designs algorithms for cross-device and cross-silo settings, and analyzes the convergence of the proposed asynchronous schemes in the cases without assumptions on worker participation, and with some assumptions on the distribution of the active workers or bounded delay. It also empirically demonstrates that the asynchronous scheme achieves competitive performance compared with synchronous baselines.

**Summary Of The Review:**

I think studying asynchronous and device-centric communication in the context of FL is definitely interesting, and I feel the algorithms and the analysis in the paper are correct and solid. My major concerns are the first three points in the 'concerns' part of my comments, especially the differences between the submission and that related work.

====update====

I have read the authors' responses. However, I don't think the differences from the previous work (https://arxiv.org/abs/2106.06639) are significant. My concerns regarding the uniform arrival and bounded delay assumptions in federated learning, the experiments, etc remain. Therefore, I do not think this submission in its current form is ready for publication and I retain my original score.

---

> ### Author Response · Authors · 2021-11-23
> **Response to Reviewer 8n7V (1/3)**
>
> Thank you very much for the review and the constructive comments. We believe the valuable suggestions from the reviewer have helped us significantly improve the quality of this paper. We believe that there is some misunderstanding regarding the results presented in this paper. In this revision, we have clarified the concerns raised by the reviewer and have carefully revised our paper based on reviewer's comments, questions, and suggestions. Please see the revised manuscript at the top of this page. The detailed point-by-point responses are as follows:
>
> > **Your Comment:** 1. The proposed algorithms are very similar to those proposed in this paper (https://arxiv.org/abs/2106.06639, Federated Learning with Buffered Asynchronous Aggregation). Although the paper I mentioned lets the server sample clients asynchronously, algorithmically looks like these two works are doing the same thing. I understand the submission is more general and improves the analysis in certain ways (e.g., without bounded gradient), but the main algorithms and some motivation are largely overlapping. It would be very helpful to clarify the major differences, if there are any.
>
> **Our Response:** Thanks for the comment. We would like to specify two main differences between AFL and asynchronous FL.
>
> 1) *System Flexibility:* This allows workers to participate in the training at will and at anytime. This worker-centric learning paradigm improves the dependence on systems heterogeneity and differentiates AFL from current FL. More specifically, AFL has concurrent and independent threads for each client and server, which is significantly different from the "sample-return-aggregation" sequential pattern in the current (asynchronous) FL.
> 2) *Optimization Flexibility:* AFL also enjoys some additional degrees of freedom in each client's optimization process such as local steps.
>
> Regarding your referred paper (https://arxiv.org/abs/2106.06639), there are substantial differences in both the algorithms and motivation. From an algorithmic perspective, the referred paper is only a special case of our AFA-CD algorithm, corresponding to Theorem 3 for uniformly distributed arrivals with constant local steps for each client. In our paper, we consider a more general learning paradigm in terms of both system architectures and optimization algorithms, both of which differentiate our work from the referred paper.
>
> From the motivation perspective, the motivation of the referred paper is to robustness against straggler effects, which is only part of our motivations. Besides this, our motivations include to design concurrent federated learning mechanism to boost client participation and provide theoretical understanding for arbitrary client participation. These differentiate our work from the referred paper.
>
> > **Your Comment:** 2. I am confused by the difference and the motivation for two variants designed for cross-device and cross-silo settings. (1) I think the server in both settings is assumed to be powerful in terms of memory and computation. Can the work give some references on why the server is assumed to have no historical information of the works? (2) The submission assumes uniform arrival for cross-device FL, and bounded delay for cross-silo FL. They both control the staleness of the model updates (to make sure the global model some device starts from is not too stale compared with the current global model for aggregation). Both of them are reasonable for cross-silo FL; and hence, to me, there isn’t a clear mapping between which setting is more compatible with which assumption
>
> **Our Response:** Cross-device and cross-silo federated learning are two typical settings in FL that are widely accepted. Cross-device is crresponding to large-scale federated learning systems with a large number of client devices that have unstable connections (e.g., such as edge devices). On the other hand, cross-silo mainly focus on collaborations among a small group of clients such as big institutions, banks, or hospitals. An overview of these two settings can be found in the excellent survey on FL [in this paper](https://arxiv.org/pdf/1912.04977.pdf) (see Table 1). We have added two references for this on Page 2 and brief introduction in Sec 3.1 in this revision.
>
> Your understanding is correct regarding these two cases (uniform arrival and bounded delay). Uniform arrival is considered in both cross-device and cross-silo AFL/FL, which is shown to have the same theoretical convergence result. For cross-silo AFL/FL with historical information being available, we propose the AFA-CD algorithm (with bounded delay) designed specifically for cross-silo AFL, which achieves a better convergence bound.

---

> > ### Author Response · Authors · 2021-11-23
> > **Response to Reviewer 8n7V (2/3)**
> >
> > > **Your Comment:** 3. I think one motivation for asynchronous training is that it leads to faster convergence (empirically) in the presence of stragglers. This point should be validated empirically (by simulating systems heterogeneity, for example).
> >
> > **Our Response:** Yes, your understanding is correct. One of the advantage of our AFL is the flexible client participation can alleviate the straggler problem. To empirically validate this advantage, we can run simulation by some pre-defined function about the parameter for systems heterogeneity the same as that in [FedBuff](https://arxiv.org/pdf/2106.06639.pdf) and [asynchronous FL](https://arxiv.org/pdf/1903.03934.pdf). In our paper, we have focused more on exploring the impacts of asynchrony, heterogeneous computing, worker’s arrival process, and non-i.i.d. datasets on the number of communication rounds to verify our theoretical results. This approach also allows a fair comparison with conventional FL algorithms.
> >
> > > **Your Comment:** 4. In practice, the delays may be unbounded (e.g., when each device only participates once or twice during the training); so the two assumptions can be too strong.
> >
> > **Our Response:** For cross-silo AFL, it makes sense to consider the bounded delay assumption since the number of clients is relatively small and both the computation and communication are relatively stable (e.g., [see this paper](https://arxiv.org/pdf/1912.04977.pdf).) For cross-device AFL, the delays may indeed be unbounded in practice. However, the unbounded delay setting may be viewed as an extension of the setting where the server abandons/reject clients with large return delay (e.g., by setting a maximum time delay threshold). Under this setting, the maximum delay in the "reduced" system can be viewed as bounded. We have added the discussions in this revision.
> >
> > > **Your Comment:** 5. Why does the lower bound also hold for FedAvg (described in Remark 1)? Under proper step-sizes, there is usually no constant error term.
> >
> > **Our Response:** We would like to clarify that FedAvg's property of  having a vanishing error term with proper steps-sizes only holds under ideal scenarios, such as with full worker participation or uniformly distributed partial worker participation. In these settings, FedAvg as well as many other FL algorithms converge to a stationary point with no constant error term. However, such ideal worker participation assumption rarely hold for FL systems in practice. Even if the server can sample the clients uniformly at random in each round, it remains hard to guarantee uniformly distributed worker participation due to other uncontrollable system factors in real-world FL systems. For example, it has been shown that more than 30% of devices never participate in the federated learning process and top 30% of devices contribute 81% of the total computation, which is due to unstable device availability and large failure proportion in real-world FL systems such as [Google FL deployment](https://arxiv.org/pdf/1902.01046.pdf) and  [practical FL simulation platform](https://arxiv.org/pdf/2006.06983.pdf).
> >
> > Therefore, in our paper, we consider the more challenging case that allows general worker participation (i.e., no assumptions being made on the worker participation model). Toward this end, in this paper, we explore the impact of worker participation on convergence and consider a more practical worker participation scenario, which is referred to as general worker information arrival processes in our paper. For FL systems with general worker information arrivals, Theorem 1 characterizes the impact of worker participation in FL/AFL and shows a non-vanishing lower bound for the *worst-case* scenario when no assumption on the worker participation is made. To our knowledge, this is a new insight in the FL literature.

---

> > > ### Author Response · Authors · 2021-11-23
> > > **Response to Reviewer 8n7V (3/3)**
> > >
> > > > **Your Comment:** 6. The last paragraph in Section 3 (1) spends much text highlighting that AFL is a general computing architecture. But this (using any local solver with different hyperparameters, allowing for device-specific local steps) is not a unique property of AFL, and lots of previous federated optimization methods also support this.
> > >
> > > **Our Response:** We agree with the reviewer that features such as using local solvers with different hyper parameters and allowing for device-specific local steps are not necessarily unique in AFL, and then have appeared in some existing FL algorithms (e.g., FedAvg, FedProx, and SCAFFOLD). The unique feature that renders AFL a general computing architecture is the fact that AFL allows loosely-coupled and concurrent computing processes on both server and worker sides, which does not exist in any conventional FL systems. Thanks to this unique feature and those algorithmic design freedoms (i.e., local solvers with different hyper parameters and allowing for device-specific local steps), AFL subsumes the conventional FL and asynchronous distributed optimization as special cases. We have added the above discussions in the revision to clarify the uniqueness of AFL.
> > >
> > > > **Your Comment:** 7. At the end of the first paragraph of related work (Sec 2), I don't think calling server-centric synchronous algorithms are not easy to implement is accurate or justified
> > >
> > > **Our Response:** Thanks for the comment. We would like to clarify that, by saying "...not easy to implement," we meant server-centric synchronous FL algorithms entail various implementation issues as discussed in Section 1 (e.g., interference between workers, periodic traffic spikes, high-complexity in maintaining a network-wide common clock). In this revision, we have modified the corresponding sentences to clarify.
> > >
> > > > **Your Comment:** 8. The phrase 'general worker information arrival process' appears for the first time in the abstract, which is a bit abrupt and unclear---what is a worker information arrival process? general in what sense?
> > >
> > > **Our Response:** Thanks for the suggestion. This sentence in the abstract may indeed introduce some clarify issues. We have added description and comparison to the widely-used worker participation assumptions to clarify what we mean by general worker information arrival process.
> > >
> > > > **Your Comment:** 9. The empirical evaluation does not describe how the stragglers are simulated precisely. For example, the starting points of the clients uniformly sampled (or sampled in a biased way) correspond to which time stamp? Does the 'bias sampling' scheme (described in the appendix) correspond to the bounded delay case?
> > >
> > > **Our response:** In this revision, we have added further explanations on how to simulate the effect of asynchrony, heterogeneous computing and worker’s arrival processes in Appendix E1. Specifically, the clients uniformly sampled (or sampled in a biased way) are simulated by randomly choosing model from the most recent five models, which corresponds to bounded delay $\tau = 5$. We have also added and modified some explanations and references in the experiment part (in Appendix).

---

### Official Review · Reviewer_Ubxj · 2021-10-31

**Correctness:** 4
**Technical Novelty And Significance:** 2
**Empirical Novelty And Significance:** 3
**Recommendation:** 5
**Confidence:** 4

**Main Review:**

This work is well positioned with the attractive properties of clients being "autonomous" in deciding whether to join the FL in each round. But, fundamentally, the studied FL problem is equivalent to asynchronous FL with random client selection (by the server), because the randomness due to the "free will" of the clients is statistically equivalent to being randomly selected by the server. Then, some additional degrees of freedom are considered, such as having different local steps and different delays. Nevertheless, these freedoms eventually are "masked" by the worse-case representations in Theorem 1, suggesting that it is equivalent to have these participating clients perform $K$ local update steps and delay at $\tau$. Furthermore, these worst cases are over both clients and time, suggesting that they are indeed the "worst" of the entire FL process. This result is quite discouraging. For example, throughout the whole FL process of say 500 rounds, if there is one and only one instance of a client having a large step size or large delay, Theorem 1 says this will dominate the entire convergence, which is counter-intuitive.

I'm willing to give the authors a chance to address these concerns, and I'll be open to adjusting the score accordingly. To me, the work itself is very interesting, but the aforementioned two issues should be addressed.

Some other comments:
(1) The requirement of having to wait for $m$ local updates is quite restrictive. Since the clients have the freedom to choose not to participate, there is always the chance that less than $m$ clients actually participate. Plus, waiting for $m$ local updates naturally leads to the straggler problem.

(2) On page 6, the authors claimed that only when considering the special case of uniformly distributed delay would we have the case of sampling $m$ clients from $[M]$ without replacement. As stated in my main concern, I do not believe this is the case. Rather, your model, even with non-uniformly distributed delay, is still equivalent to sampling $m$ clients from $[M]$ without replacement.

(3) Following up on (2), Theorem 3 is not well explained. It is unclear why with an additional assumption of uniformly distributed delay, one can improve Theorem 3, which relies only on the worst case delay.

(4) Theorem 2 has a dependence on $\alpha_L$, which scales with $\tau^2$. This term may becoming dominating if $\tau$ is large, as opposed to the $\sigma_G$ term. This is however hidden in Corollary 1, which is worth discussing.

(5) Typo: CS-AFL in page 2 should be AFL-CS.

**Summary Of The Paper:**

This paper proposes a general FL framework, called anarchic federated learning (AFL), that allows voluntary client participation of clients, with individual update steps and delay. Algorithms for both cross-device and cross-silo are presented. Convergence upper bounds and a lower bound are derived. Experimental results are reported to demonstrate the effectiveness of the proposed algorithm.

**Summary Of The Review:**

As mentioned in the main review, I believe this work has potential and is interesting, but difference to asynchronous FL with random client selection (by the server) and addressing the worst-case behavior that dominates the convergence bounds are the two concerns I have.

========

Post-rebuttal: I have read the authors' responses and some (responses 3 and 4) have clarified confusions I had, which is helpful. The other responses are largely in line with my understanding of the paper. Overall, my review of this paper does not change.

---

> ### Author Response · Authors · 2021-11-23
> **Response to Reviewer Ubxj (1/3)**
>
> Thank you very much for the review and the constructive comments. We believe the valuable suggestions from the reviewer have helped us significantly improve the quality of this paper. We believe that there is some misunderstanding regarding the results presented in this paper. In this revision, we have clarified the concerns raised by the reviewer and have carefully revised our paper based on reviewer's comments, questions, and suggestions. Please see the revised manuscript at the top of this page. The detailed point-by-point responses are as follows:
>
> > **Your Comment:** 1. This work is well positioned with the attractive properties of clients being "autonomous" in deciding whether to join the FL in each round. But, fundamentally, the studied FL problem is equivalent to asynchronous FL with random client selection (by the server), because the randomness due to the "free will" of the clients is statistically equivalent to being randomly selected by the server. Then, some additional degrees of freedom are considered, such as having different local steps and different delays. Nevertheless, these freedoms eventually are "masked" by the worse-case representations in Theorem 1, suggesting that it is equivalent to have these participating clients perform K local update steps and delay at $\tau$.
>
> **Our Response:** Thanks for your constructive comments. Your understanding is correct about the "free will" of the clients, i.e., it is statistically and approximately equivalent to being randomly selected by the server. However, we can not rule out other skewed scenarios due to device availability or unstable connection. One example is that 30% of devices never participate in the learning process in [FL systems in practice](https://arxiv.org/pdf/2006.06983.pdf) even if the server samples the workers uniformly at random. So we consider the worst-case scenarios (arbitrary workers' participation) by removing any workers' participation assumption, denoted as General Worker Information Arrival Processes. We believe that is one of the novelties in this paper, where we consider such a general worker participation model and no previous work has provided theoretical insight on the impact of arbitrary worker participation.
>
> We would like to specify two main differences between AFL and asynchronous FL with random client selection.
>
> 1) *System Flexibility:* This allows workers to participate in the training at will and at anytime. This worker-centric learning paradigm improves the dependence on systems heterogeneity and differentiates AFL from current FL. More specifically, AFL has concurrent and independent threads for each client and server, which is significantly different from the "sample-return-aggregation" sequential pattern in the current (asynchronous) FL.
> 2) *Optimization Flexibility:* AFL also enjoys some additional degrees of freedom in each client's optimization process such as local steps.
>
> Your comment on optimization freedom in local steps is correct. They eventually are "masked" by the worse-case representations. It is possible that the upper bound could be further improved, but we do not find other better results in the FL literature. The most relevant work we can find in asynchronous FL is that the dependence of delay is $\tau_{max}$ (cf. Theorem 1 and Corollary 1 [in this paper](https://arxiv.org/pdf/2106.06639.pdf)).

---

> > ### Author Response · Authors · 2021-11-23
> > **Response to Reviewer Ubxj (2/3)**
> >
> > > **Your Comment:** 2. Furthermore, these worst cases are over both clients and time, suggesting that they are indeed the "worst" of the entire FL process. This result is quite discouraging. For example, throughout the whole FL process of say 500 rounds, if there is one and only one instance of a client having a large step size or large delay, Theorem 1 says this will dominate the entire convergence, which is counter-intuitive.
> >
> > **Our Response:** Your understanding regarding the dominance of slow client is correct. However, this does not necessarily mean that Theorem 1 is too loose to be meaningful. Consider the following simple example: there are only two clients in the FL system with local loss functions $f_1(x) = (x+1)^2$ and $f_2(x) = (x-1)^2$. The first client always participates in the training ($\forall t \in [T]$), while the second client never participates; then the output would goes to the minimal point (i.e.,  -1) of $f_1$ instead of minimal point (i.e., 0) of $f(x) = \frac{1}{2} (f_1(x) + f_2(x))$. In addition, it has be found by [in FL platforms in practice](https://arxiv.org/pdf/2006.06983.pdf) that the slow-down due to such system heterogeneity (even without anarchic behavior) is not uncommon.
> >
> > Note that, in this paper, we consider the worst-case scenarios for arbitrary worker participation including the cases above. In this sense, our results are not overly conservative. Rather, this is an intrinsic property of FL systems with large data and system heterogeneity. On the other hand, our paper also sheds lights on FL systems, where flexible client's participation should be encouraged to prevent extreme cases from happening. In addition, once we have assumptions on the client's participation, e.g., randomness in Theorem 3 and bounded delay in Theorem 4, a stationary point convergence is guaranteed, which is the same in order sense as that in traditional distributed learning without data and/or system heterogeneity.
> >
> > > **Your Comment:** 3. The requirement of having to wait for $m$ local updates is quite restrictive. Since the clients have the freedom to choose not to participate, there is always the chance that less than $m$  clients actually participate. Plus, waiting for $m$ local updates naturally leads to the straggler problem.
> >
> > **Our Response:** We believe there are misunderstandings here. In AFL, each client and server are in independent and concurrent threads. Each client could just continue the local training by pulling the latest global model without waiting for the server's aggregation. Meanwhile, the server aggregates and updates the global model whenever there are $m$ new local updates, where $m$ is a tunable system parameter. That is, the operations of clients and the server are largely independent and concurrent, which would eliminate the straggler problem. This is one of the novelties for AFL that differentiates from previous works in conventional FL. In this revision, we have modified the statement in the AFL framework and algorithms to make it clearer.
> >
> > > **Your Comment:** 4. On page 6, the authors claimed that only when considering the special case of uniformly distributed delay would we have the case of sampling m clients from [M] without replacement. As stated in my main concern, I do not believe this is the case. Rather, your model, even with non-uniformly distributed delay, is still equivalent to sampling m clients from [M] without replacement.
> >
> > **Our Response:** There may be some confusion in there. We never meant to say our AFL model is equivalent to sampling m clients from [M] without placement only in the case of uniformly distributed delay. We only said that AFL could be viewed as  uniformly random client sampling with uniformly distributed delays. To avoid this consuion, in this revision, we specifically state the assumption, i.e., uniformly distributed worker information arrivals, equivalent to uniform worker sampling in partial worker participation in typical cross-device FL systems.

---

> > > ### Author Response · Authors · 2021-11-23
> > > **Response to Reviewer Ubxj (3/3)**
> > >
> > > > **Your Comment:** 5. Following up on (2), Theorem 3 is not well explained. It is unclear why with an additional assumption of uniformly distributed delay, one can improve Theorem 3, which relies only on the worst case delay.
> > >
> > > **Our Response:** We have added additional explanation and discussion on Page 29 in the Appendix. We also provide this additional explanation here as well: in each communication round, the surrogate objection function for partial worker participation in FL is $\tilde{f}(x) := \frac{1}{| \mathcal{M}_t |} \sum_{i \in \mathcal{M}_t} f_i(x)$. For uniformly distributed arrival process, the surrogate object function approximately equals to $f(x):= \frac{1}{M} \sum_{i=1}^{M} f_i(x)$ in expectation, i.e., $\mathbb{E}[\tilde{f}(x)] = f(x)$. However, the surrogate object function $\tilde{f}(x)$ may deviate from $f(x)$ when the worse-case should be considered, i.e., arbitrarily worker participation. We have also added these extra explanation and discussion below Theorem 3 in this revision.
> > >
> > > > **Your Comment:** 6. Theorem 2 has a dependence on $\alpha_L$, which scales with $\tau^2$. This term may becoming dominating if $\tau$ is large, as opposed to the $\sigma_G$ term. This is however hidden in Corollary 1, which is worth discussing.
> > >
> > > **Our Response:** Thanks for the comment. The dependence of delay $\tau$ indeed worths more discussion. It can be shown that the delay only increases the variance $\sigma_L$ and is independent of data heterogeneity $\sigma_G$. Compared with asynchronous FL without anarchic client behavior, the dependence of delay result is the same, i.e., $\tau_{max}^2$ as shown in Theorem 1 and Corollary 1 [in this paper](https://arxiv.org/pdf/2106.06639.pdf). We have added these discussions of $\tau$ and $K$ in the revision.
> > >
> > > > **Your Comment:**  7. Typo: CS-AFL in page 2 should be AFL-CS.
> > >
> > > **Our response:** Thanks for catching the typo. We have corrected it in the revision.

---

### Official Review · Reviewer_bjgp · 2021-11-02

**Correctness:** 3
**Technical Novelty And Significance:** 2
**Empirical Novelty And Significance:** 2
**Recommendation:** 3
**Confidence:** 4

**Main Review:**

## Convergence results

1) Why can't assumption 3 be relaxed to bound the heterogeneity only at any stationary point of $f$, i.e., $\|\|\nabla f_i(x^\star) - \nabla f(x^\star)\|\|^2 = \|\|\nabla f_i(x^\star)\|\|^2 \leq \sigma_G^2$, such as in [this work](https://arxiv.org/pdf/2006.04735.pdf)? This kind of an assumption better captures the heterogeneity of workers and is useful in understanding interpolation problems (c.f., [this work](https://arxiv.org/abs/2106.02720)). Regardless it should be clarified what are $x_k$'s in assumption 2 and 3, and assumption 3 needs an expectation because $x_k$'s are random variables(?).

2) There is no need to introduce $G$ in Theorem 1, instead just say that it is valid for any level of heterogeneity characterized by $\sigma_G$. Moreover, **the theorem is very vaguely stated and in a sense wrong**.

    - What do you mean by "any AFL algorithm"? Is SGD on each machine initialized at $0$ with a zero learning rate, not an AFL algorithm? That gets to the stationary point of the lower bound function exactly in zero steps! You need to give a lower bound for every optimization algorithm, for any setting of initialization and hyper-parameters (c.f., the [lower bounds here](https://arxiv.org/pdf/2006.04735.pdf) and in the references therein).
    - Also, you allow for an arbitrary number of local steps on each machine i.e., $K=\infty$, so that you can ignore the $\epsilon$ in the lower bound proof. This is not the assumption under which you analyze your algorithms, so it is incorrect to use this lower bound to claim that the results are tight in $\sigma_G$. In fact, if one assumes that worker step-size is small and $K, \tau < \infty$ then that would allow for a more optimistic lower bound, and a better baseline for your upper bounds. In that case, $\epsilon$ will depend on the $\eta, K, \tau$ like it should.
    - And what does it mean that the theorem holds for non-iid FL? The [lower bound here](https://arxiv.org/pdf/2006.04735.pdf) is already smaller than $\sigma_G^2$ (albeit for the convex setting, which is fine because your function is convex as well)!

3) In corollary 1, I do not buy that there should be a $\sigma_G^2$ term. This is because I don't believe the lower bound in Theorem 1 for the reasons stated above. With a small enough client step-size, it should be possible to have a decaying term w.r.t. $\sigma_G$.

4) You discuss in detail how it is important to improve the dependence on systems heterogeneity in the current federated learning guarantees. As an idea, anarchic federated learning sounds very enticing, but the result in theorem 3 is not AFL, it is just uniform partial client participation. Thus the theoretical result falls short of giving any novel insights.
    - We already have results, in the setting where the local updates are non-uniform but bounded, and when the gradient updates are delayed (asynchronous updates). How is your result not just a combination of both of them? I was hoping your analysis doesn't require these boundedness assumptions. What is worse is that the dependence is somewhat worse here. For instance, the effect of delay has already been captured in [this paper](https://arxiv.org/pdf/1909.05350.pdf) (it has a $\tau$ dependence vs your $\tau^2$), and it has even been improved to depend on $\tau_{avg}$ instead of depending on the worst delay in [this paper](https://arxiv.org/pdf/2106.11879.pdf). The latter is very significant, especially in actual anarchic federated learning, where some workers might participate very intermittently, but others might not. If the dependence on $\tau$ can't be avoided, it should at least have been reduced to $\tau_{avg}$.
    - Finally, how is this rate equivalent to the rate provided in the Scaffold paper in the non-iid FL setting? Their leading term is $1/\sqrt{mKT}$ which is better than your $1/\sqrt{mT}$ term. It is both **incorrect and misleading to assume that $K$ is a constant** and not a lever we'd like to move to see how the rate changes!

5) I think it is indeed interesting to see a speedup in terms of $M$ instead of $m$ in corollary 3. In my opinion, this was the most interesting result in the paper. Overall, I think the paper can improve its writing to remove the focus from Theorem 1, 2 and instead direct it to the improved results for partial client sampling with asynchrony in theorem 3 and 4 (of course while improving upon the dependencies I point out above).

## Experiments

It is unclear to me what to make of the experiments provided in the main paper. How was the anarchic behavior of the machines emulated in figure 1? It would make sense to bring some of this discussion from the appendix to the main paper.


***
## Minor comments

1) Linear speed-up is an abused term in FL literature. It would be good to specify somewhere that (i) this is just convergence to a first-order stationary point, which might not completely capture the performance while training neural networks, and (ii) the linear speedup is conditional (even theoretically), i.e., it holds only till the number of machines is not too large (c.f., [this paper](https://arxiv.org/abs/1811.03600)).

> We count each global model update as one communication round.

2) We are not in the intermittent communication setting anymore, what does a communication round even mean? Devices are asynchronously pulling and pushing in this model. What does this mean then?

3) Don't hide the dependence on $L$ and $\sigma_L^2$ in the corollaries. It makes it difficult to compare different terms in convergence with known convergence results. Moreover, it would make more sense to put the relevant baselines in a table to show a direct comparison with non-iid FL rates. It might also save some space.







**Summary Of The Paper:**

This paper analyses a variant of generalized federated averaging ([Wang et al.](https://arxiv.org/abs/2107.06917)) with partial worker participation and asynchrony in the stateful (i.e., the worker specific data can be saved on the server) and stateless settings, which characterize cross-silo and cross-device federated learning ([Kairouz et al.](https://arxiv.org/abs/1912.04977?utm_source=feedburner&utm_medium=feed&utm_campaign=Feed%253A+arxiv%252FQSXk+%2528ExcitingAds%2521+cs+updates+on+arXiv.org%2529)) respectively. Specifically for each global update, the server uses fresh gradients from $m$ out of $M$ machines, where each machine can make some local SGD updates starting from a stale global iterate (i.e., the gradients are delayed). This setting is termed *anarchic federated learning* (AFl) when the workers can have an arbitrary number of local steps and gradient delays.

The authors first provide a lower bound for convergence to a first-order stationary point in the AFL setting. Then they upper bound the convergence of their algorithms in the stateful and stateless setting when the local updates and delay are bounded. Under some regimes, and assumptions on the worker sampling and delay distribution, a linear convergence speed-up can be shown w.r.t. the number of machines ($m$ for the stateless setting and $M$ for the stateful setting). Some experiments are provided to measure the effect of the anarchic worker behavior.

**Summary Of The Review:**

I don't believe that the provided results or the techniques used in this paper are surprising. The convergence results provided here are not tight in all the constants, and some of the assumptions made to provide those are too restrictive and don't morally allow anarchic behavior. As a result, I am not in favor of accepting this paper in its current state.

---

> ### Author Response · Authors · 2021-11-23
> **Response to Reviewer bjgp (1/3)**
>
> Thank you very much for the review and the constructive comments. We believe the valuable suggestions from the reviewer have helped us significantly improve the quality of this paper. We believe that there is some misunderstanding regarding the results presented in this paper. In this revision, we have clarified the concerns raised by the reviewer and have carefully revised our paper based on reviewer's comments, questions, and suggestions. Please see the revised manuscript at the top of this page. The detailed point-by-point responses are as follows:
>
> > **Your Comment:** 1. Why can't assumption 3 be relaxed to bound the heterogeneity only at any stationary point of , i.e., $\| \nabla f_i(x^*) - \nabla f(x^*) \|^2 \leq \sigma_{G}^2$, such as in this work? This kind of an assumption better captures the heterogeneity of workers and is useful in understanding interpolation problems (c.f., this work). Regardless it should be clarified what are $x_k$ 's in assumption 2 and 3, and assumption 3 needs an expectation because
> 's are random variables(?).
>
> **Our Response:** Thanks for pointing out the alternatively relaxed assumption for Assumption 3. When studying non-convex FL problems, Assumption 3 is commonly-used in previous works including [your referred paper](https://arxiv.org/pdf/2006.04735.pdf) (Eq. (12) and Theorem 3) and other papers in FL (listed in our paper below Assumption 3 in Page 4). To our best knowledge, there is no existing work in non-convex setting adopting this relaxed assumption. In fact, even in [your referred paper](https://arxiv.org/pdf/2006.04735.pdf) (Page 7), the authors also used the same assumption as our Assumption 3 (cf. Eq. (12) on Page 7) and mentioned that "the difference between the local and global objectives gradient everywhere" can precisely capture homogeneity and see a smooth transition from the heterogeneous to homogeneous set in non-convex setting.
>
> For the clarity of $x_k$, we have changed the $x_k$ to $x$ in the assumptions, where $x$ is a parameter vector rather than a random variable. Thus, no expectation in Assumption 3 is needed. This is actually the same as Eq. (12) in [your referred paper](https://arxiv.org/pdf/2006.04735.pdf).
> But we thank the reviewer for pointing out these notation clarification issues, and we have added this clarification in the assumptions in this revision.
>
> > **Your Comment:** 2. There is no need to introduce $G$ in Theorem 1, instead just say that it is valid for any level of heterogeneity characterized by $\sigma_G$. Moreover, the theorem is very vaguely stated and in a sense wrong...
>
> **Our Response:**
> Thanks for your insightful comments. We believe there exists some misunderstanding for Theorem 1. In this revision, we provide further clarifications as follows:
>
> * In this revision, we have revised the statement of Theorem 1 in the form of any level of heterogeneity $\sigma_G$.
>   In addition, we have modified Theorem 1 by saying any potentially randomized FL algorithm instead of any FL algorithm to make it easier to understandable.
>
> * The lower bound in Theorem 1 is different from these in previous works, where we focus on worse-case scenario for the arbitrary worker participation case caused by system heterogeneity. Previous lower bounds in FL, including Theorem 2 in [yourreferred paper](https://arxiv.org/pdf/2006.04735.pdf) and Theorem 2 in [SCAFFOLD](https://arxiv.org/pdf/1910.06378.pdf), were built upon ideal worker participation assumptions, i.e., full worker participation (all machines participate in every round) or partial worker participation (participated machines are i.i.d. sampled in each round or some fixed probability). In other words, Theorem 1 in our paper captures the worse-case scenario for arbitrary system heterogeneity, while previous works studied the lower complexity in the optimization process with ideal system heterogeneity.
>
> * In the lower bound proof, the reason we ignored the $\epsilon$ parameter is that we assume the algorithm can find a stationary point under sufficient training time $T$. This is irrelevant to the number of local steps. A concrete example is as follows: there are only two workers in the FL system with local loss functions $f_1(x) = (x+1)^2$ and $f_2(x) = (x-1)^2$, but the first worker always participates in the training ($\forall t \in [T]$) while the second worker never participates; then the output would go to the minimum point (i.e.,  -1) of $f_1$ instead of minimum point (i.e., 0) of $f(x) = \frac{1}{2} (f_1(x) + f_2(x))$. It thus can be seen that the result is largely independent of the optimization process. This result is applicable to non-IID FL as well and is orthogonal to other lower bounds such as Theorem 2 in [your referred paper](https://arxiv.org/pdf/2006.04735.pdf) and Theorem 2 in [SCAFFOLD](https://arxiv.org/pdf/1910.06378.pdf). To our knowledge, our work is the first to provide a lower bound for the analysis of the impact of worker participation in FL/AFL.

---

> > ### Author Response · Authors · 2021-11-23
> > **Response to Reviewer bjgp (2/3)**
> >
> > > **Your Comment:** 3. In corollary 1, I do not buy that there should be a $\sigma_G$ term. This is because I don't believe the lower bound in Theorem 1 for the reasons stated above. With a small enough client step-size, it should be possible to have a decaying term w.r.t. $\sigma_G$.
> >
> > **Our Response:** As stated above, Theorem 1 in our paper captures the worse-case scenario for arbitrary system heterogeneity, while previous works studied the case with more ideal system heterogeneity. In Corollary 1, there exists a $\sigma_G$ term due to arbitrary system heterogeneity (denoted as General Worker Information Arrival Processes). Even with proper optimization hyper-parameters, it is impossible to decay the $\sigma_G$ term, which is caused by arbitrary system heterogeneity as shown in the example above. In this sense, the bound in corollary 1 is tight in terms of the error $\sigma_G$ term.
> >
> >
> > > **Your Comment:** 4. You discuss in detail how it is important to improve the dependence on systems heterogeneity in the current federated learning guarantees. As an idea, anarchic federated learning sounds very enticing, but the result in theorem 3 is not AFL, it is just uniform partial client participation. Thus the theoretical result falls short of giving any novel insights...
> >
> > **Our Response:** The novelty of AFL is to allow workers to participate in the training at will and at anytime. This worker-centric learning paradigm improves the dependence on systems heterogeneity and differentiates AFL from current FL. However, it does not necessarily mean that our AFL algorithms can tolerance arbitrary client behaviors. In order to guarantee convergence, our AFL algorithms can also have some assumptions or conditions, such as uniformly distributed arrivals in Theorem 3. If workers are of similar types, such assumption is a good statistical approximation for large-scale cross-device AFL.
> >
> > Technically, Theorem 3 shows the convergence under the complex couplings among asynchrony, different local step and partial worker participation, which generalizes previous results. Note the two referred papers by the reviewer are both delayed SGD (Section 4 [in your referred paper](https://arxiv.org/pdf/1909.05350.pdf) and Section 3 [in your referred paper](https://arxiv.org/pdf/2106.11879.pdf)) without any data or systems heterogeneity. Thus, it is unfair to direct compare our result with theirs. We have double-checked all asynchronous federated learning literatures as we can, and do not find any better dependence of $\tau$. To our knowledge, the most recent result for dependence of delay for asynchronous federated learning without anarchic behavior is $\tau_{max}^2$ as shown in Theorem 1 and Corollary 1 [in this paper](https://arxiv.org/pdf/2106.06639.pdf).
> >
> > Also, we would like to clarify that we do not compare with the convergence rate of SCAFFOLD algorithm but instead compare with the FedAvg analysis in the SCAFFOLD paper. Specifically, the bound of FedAvg for partial worker participation is $\mathcal{O} (\frac{M}{\sqrt{RKS}}) = \mathcal{O} (\frac{1}{\sqrt{RS}})$ where $M^2 \sim K$, which is detailed on Page 19 of [the SCAFFOLD paper](https://arxiv.org/pdf/1910.06378.pdf) (notations are adopted as originals). Hence, it exactly matches our rate $\frac{1}{\sqrt{mT}}$. We added the name of specific algorithm for comparison to avoid confusions in Remark 4 and have also made it clear in the revision.
> >
> > > **Your Comment:** 5. I think it is indeed interesting to see a speedup in terms of $M$ instead of $m$ in corollary 3. In my opinion, this was the most interesting result in the paper. Overall, I think the paper can improve its writing to remove the focus from Theorem 1, 2 and instead direct it to the improved results for partial client sampling with asynchrony in theorem 3 and 4 (of course while improving upon the dependencies I point out above).
> >
> > **Our Response:** Thanks for the positive feedback. We believe there are misunderstandings for Theorem 1 and 2. We hope the above explanation would address your concerns. Theorem 3 actually follows the line of Theorems 1 and 2. One advantage of Theorem 3 is the speedup of $M$ instead of $m$ as you mentioned. Another advantage is that we only need a bounded delay condition to guarantee a stationary point convergence. Such a bounded delay assumption is more relaxed and practical than widely-adopted assumptions, such as full worker participation or uniformly sampled partial worker participation.

---

> > > ### Author Response · Authors · 2021-11-23
> > > **Response to Reviewer bjgp (3/3)**
> > >
> > > > **Your Comment:** 6. Linear speed-up is an abused term in FL literature. It would be good to specify somewhere that (i) this is just convergence to a first-order stationary point, which might not completely capture the performance while training neural networks, and (ii) the linear speedup is conditional (even theoretically), i.e., it holds only till the number of machines is not too large (c.f., this paper).
> > >
> > > **Our Response:** We have introduced the notion of linear speedup in the footnote of Page 2, which is a widely adopted terminology in the federated learning literature. Here, we would like to futher clarify. When we show the convergence rate order, we state "for a sufficiently large $T$ ... with bounded ..." in the following remarks of all corollaries. So whenever we state convergence rate results, it implies that the result are asymptotic and hold for a sufficiently large $T$. We also specifically point out the linear speedup to stationary point in Corollary 2. In this revision, we have revised our statements. Hopefully, these explanations would clarify the misunderstanding.
> > >
> > > > **Your Comment:** 7. "We count each global model update as one communication round." We are not in the intermittent communication setting anymore, what does a communication round even mean? Devices are asynchronously pulling and pushing in this model. What does this mean then?
> > >
> > > **Our Response:** Your understanding is correct that the system is working in a continuous-time fashion in terms of communication as devices are asynchronously pulling and pushing in this model. But even in AFL, it remains possible to define the notion of communication round as follows: in AFL, a communication round is defined as the period between two successive global model updates on the server's side, which is comparable to a communication round in conventional FL in terms of model updates.

---

> > > > ### Comment · Reviewer_bjgp · 2021-11-30
> > > > **Response to author feedback**
> > > >
> > > > Thanks for your response, and apologies for the late response.
> > > >
> > > > I am fine with your response to my first point.
> > > >
> > > > I am still not satisfied with the response to my second point, I don't think it is correct to assume the algorithm can always find a stationary point, w.l.o.g. These are non-asymptotic lower bounds and should have an explicit dependence on the number of local steps, etc. even if you consider worst-case participation. I understand the anarchic nature of your lower bound, and how it is different from previous work, I just don't understand how you can assume that $\epsilon$ can be arbitrarily small within finite time. Due to the same reason, I don't believe your response to my third point.
> > > >
> > > > > Thus, it is unfair to direct compare our result with theirs. We have double-checked all asynchronous federated learning literatures as we can, and do not find any....
> > > >
> > > > I agree that the references I mentioned use delayed gradients, and do not directly deal with heterogeneity. However, the delay is a natural way to model systems heterogeneity. And if the upper bounds only depend on average delay, one would hope that for anarchic federated learning, where there is some structure to how the workers participate (like in your results), a better comparison can be made to the delayed gradient literature. It is incorrect to claim your setting is AFL, when you do indeed make assumptions about worker participation.
> > > >
> > > > > Another advantage is that we only need a bounded delay condition to guarantee a stationary point convergence.
> > > >
> > > > How is that better than bounded average delay as in the paper I mentioned. This underlines the point I am trying to make, your actual theoretical result is indeed related to the delayed updates literature, even if the general case AFL is not.
> > > >
> > > > > in AFL, a communication round is defined as the period between two successive global model updates on the server's side, which is comparable to a communication round in conventional FL in terms of model updates.
> > > >
> > > > I would rather plot it in terms of model updates then, instead of calling it communication, and include wall clock time experiments.
> > > >
> > > > Overall, I am still dissatisfied with how the authors present the lower bound and the theoretical novelty of the work. I appreciate the authors clarifying their writing, but would retain my score.

---

### Decision · Program_Chairs · 2022-01-20

**Decision:**

Reject

**Comment:**

This paper on 'anarchic' federated learning (FL) envisions an FL framework where edge clients can act independently instead of their participation being controlled by a central server. The idea is certainly promising, however, the reviewers pointed out the following main issues:
1) Technical gaps in the theoretical analysis need to be addressed
2) Bounded delay assumptions are too strong and are mismatching with the 'anarchic' goal of the framework
3) The linear speed-up claim should be better explained and justified.
The paper generated lots of post-rebuttal discussions. However, the concerns about the theoretical analysis still remain, and therefore I recommend rejection. I hope the authors will take these constructive comments into account when revising the paper.